# High-resolution drought simulations and comparison to soil moisture observations in Germany

Friedrich Boeing[1], Oldrich Rakovec[1,2], Rohini Kumar[1], Luis Samaniego[1], Martin Schrön[3], Anke Hildebrandt[1], Corinna Rebmann[1], Stephan Thober[1], Sebastian Müller[1], Steffen Zacharias[3], Heye Bogena[4], Katrin Schneider[5], Ralf Kiese[5], Sabine Attinger[1], and Andreas Marx[1]

[1]Helmholtz Centre for Environmental Research – UFZ, Department Computational Hydrosystems, Permoserstraße 15, 04318 Leipzig, Germany
[2]Faculty of Environmental Sciences, Czech University of Life Sciences Prague, Praha-Suchdol 16500, Czech Republic
[3]Helmholtz Centre for Environmental Research – UFZ, Department Monitoring and Exploration Technologies, Permoserstraße 15, 04318 Leipzig, Germany
[4]Forschungszentrum Jülich GmbH, Agrosphere Institute (IBG-3), Germany
[5]Karlsruhe Institute of Technology, IMK-IFU, Ecosystem Matter Fluxes, Kreuzeckbahnstr. 19, 82467 Garmisch-Partenkirchen, Germany

**Correspondence:** Friedrich Boeing (friedrich.boeing@ufz.de)

**Abstract.** Germany's 2018-2020 consecutive drought events resulted in impacts in multiple sectors including agriculture, forestry, water management, energy production, and transport. High-resolution information systems are key to preparedness for such extreme drought events. This study evaluates the new setup of the one-kilometre German drought monitor (GDM), which is based on daily soil moisture (SM) simulations from the mesoscale hydrological model (mHM). The simulated SM is compared against a set of diverse observations from single profile measurements, spatially distributed sensor networks, cosmic-ray neutron stations and lysimeters at 40 sites in Germany. Our results show that the agreement of simulated and observed SM dynamics in the upper soil (0-25 cm) is especially high in the vegetative active period (0.84 median correlation R) and lower in winter (0.59 median R). The lower agreement in winter results from methodological uncertainties in both simulations and observations. Moderate but significant improvements between the coarser 4km resolution setup and the ≈1.2km resolution GDM in the agreement to observed SM dynamics is observed in autumn (+0.07 median R) and winter (+0.12 median R). Both model setups display similar correlations to observations in the dry anomaly spectrum, with higher overall agreement of simulations to observations with a larger spatial footprint. The higher resolution of the second GDM version allows for a more detailed representation of the spatial variability of SM, which is particularly beneficial for local risk assessments. Furthermore, the results underline that nationwide drought information systems depend both on appropriate simulations of the water cycle and a broad, high-quality observational soil moisture database.

## 1 Introduction

The extreme drought events since 2018 in Germany led to multi-sectoral impacts (Madruga de Brito et al., 2020; Orth et al., 2022) and increased stakeholder awareness. Moreover, recent studies emphasized that extreme SM drought events will be

more likely and more severe in Central Europe under future warming scenarios (Samaniego et al., 2018; Grillakis, 2019).

The singularity of the 2018/19 drought within observational records in terms of consecutive multiyear water deficits has been confirmed for Germany and Central Europe (Boergens et al., 2020; Hari et al., 2020; Rakovec et al., 2022). With these prospects comes an increased need for state-of-the-art information on droughts as a basis for precise assessment of the uniqueness and potential impacts of drought events from local to continental scales.

In recent years, several national and international drought monitoring systems have been developed. The German Drought
Monitor (GDM) was first introduced in 2014 as an information platform for agricultural droughts in Germany under www.ufz. de/droughtmonitor and is operationally updated daily (Zink et al., 2016). Core to the GDM is simulated SM using the open-source mesoscale hydrological model (mHM; Samaniego et al., 2010; Kumar et al., 2013). The GDM provides a near real-time status of SM and drought in Germany with a time lag of one day due to the meteorological data availability. Information on the drought status is provided for the uppermost soil layer (25 cm) and total soil column (varying depth depending on the soil map)
by calculating the Soil Moisture Index (SMI; Samaniego et al., 2013) and Plant Available Water (PAW). With around 2200 media contributions in the year 2020 and more than four million website views since 2018 alongside its use in national and federal state agencies, it proved its important role as a drought information tool in Germany. The feedback and requests received show that the GDM is used by interested public and practitioners as well as media and politics to obtain up-to-date drought information. A crucial aspect for optimal use of scientific environmental data, from a practitioners point of view, is applicability
to local purposes. Data from targeted stakeholder interviews within the EDgE project (http://edge.climate.copernicus.eu) and in Climalert (http://climalert.eu/) with a core stakeholder group of 15 farmers in Central Germany supported this need. So far, hydrological models applied at national or international level in operational drought services were mostly run on relatively low spatial resolutions, e.g. with grid cell size $5 \times 5\,\mathrm{km^2}$ in the European drought observatory (EDO) (Sepulcre-Canto et al., 2012) or $4 \times 4\,\mathrm{km^2}$ in the GDM (Zink et al., 2017). The spatial resolution is mainly restricted due to input data availability, such as
the soil map BUEK1000 (spatial resolution 1:1,000,000) for Germany.

Recently, an updated version of the nationwide German soil database (BGR, 2020) was published with 25 times higher res-olution enabling hydrological modelling at much higher spatial resolution ($\approx 1.2 \times 1.2\,\mathrm{km^2}$, an $\approx 11$ fold increase to the prior GDM version). Nevertheless, it was not clear how the quality of the SM simulations would change at higher spatial resolution. In contrast to other environmental variables it is very challenging to aggregate SM to a larger scale due to its highly hetero-
geneous nature and measurement uncertainties (Western et al., 2004; Bogena et al., 2010; Rosenbaum et al., 2012). Simulated SM derived from hydrological models is the prime alternative to observed SM and is widely employed for SM estimation on regional to global scales (Keyantash and Dracup, 2002). Nevertheless, simulations also face methodological uncertainties, especially under transient conditions such as those caused by climate change (Marx et al., 2018; O et al., 2020). Cammalleri et al. (2015) investigated the use of hydrological models for drought monitoring in Europe using SM anomalies and drought
classification metrics and found that including multiple hydrological models improved overall performance. Furthermore, hy-drological models are typically calibrated based on streamflow, which represents the integral hydrological catchment response. Besides validating the modelled streamflow, there is a clear need to thoroughly evaluate other water cycle components that are not used for constraining the model parameters. Ideally, such validations require observations of the variable of interest that

(a) cover the same spatial scales as the model and extend over different climate regimes within the study area and (b) extend over long temporal scales, which would allow them to be termed "representative". Although large-scale meteorology-driven SM variations can display seasonal varying length scales up to 500 km (Koster et al., 2019), small-scale SM variability largely depends on local site characteristics such as soil properties, topography and land use. Therefore, optimal drought monitoring systems over large areas should make use of the best available observation data in combination with a smart simulation system.

Enormous efforts have been and are being made to construct environmental observation networks from regional to global scales. Within global environmental monitoring networks such as FLUXNET which focuses on measuring ecosystem carbon fluxes (Baldocchi et al., 2001), SM is sometimes measured in multiple depths at single profiles. However, extensive validations of simulated SM from hydrological models are hampered by the limited spatial representativeness of point-scale sensors and hence require novel measurement approaches to bridge the scale gap between local observations and model resolutions. Measurements that capture the spatial structure of SM at larger scales are expensive and time-consuming, and for this reason are rare and only applicable in comparatively small catchments of a few tens of hectares (Bogena et al., 2010). In Germany, the infrastructure of Terrestrial Environmental Observatories (TERENO) has been established since 2008 to build up a nationwide long-term monitoring network in which one of the focuses is on hydrological variables (Zacharias et al., 2011; Bogena, 2016). Many of those sites were equipped with spatially distributed measurements of SM networks (SDM, Bogena et al., 2010) and Cosmic-Ray Neutron Sensors (CRNS, Zreda et al., 2012; Andreasen et al., 2017; Schrön et al., 2018)). CRNS detectors count neutrons of the natural cosmic-ray background radiation as a proxy for soil water content (Desilets et al., 2010; Köhli et al., 2021). The integral measurement footprint covers areas of 300–600 m diameter and depths of 15–70 cm, both increasing for dryer conditions (Köhli et al., 2015; Schrön et al., 2017). The CRNS method has emerged as a reliable technique to continuously monitor root-zone SM at the field scale (Bogena et al., 2015; Andreasen et al., 2017) and has been used recently for the validation of land surface and hydrological models (Han et al., 2016; Iwema et al., 2017; Dimitrova-Petrova et al., 2020).

Satellite-based SM data benefits from spatial coverage at the kilometre scale, but the shallow penetration of the signal in the upper few centimeters of the soil is a significant constraint. While those signals also depend on the surface condition, vegetation density and microwave frequencies, these products themselves require ground-based SM observations for validation (Peng et al., 2021). The time series of SDM and CRNS observations at the TERENO sites appear to be better suited for evaluation of the drought monitor model in terms of long-term continuity and root-zone representation. In particular, the data covers recent wet (e.g. 2017) and dry (e.g. 2015, 2018–2020) years, including extreme drought conditions.

Here, we evaluate SM simulations from mHM at the one and four kilometre scale simulated against an unprecedented compilation of SM observations from 40 locations across Germany. A wide range of climatic conditions and vegetation types is covered. Specifically, the study aims to answer two questions. Firstly, how well do the high-resolution German-wide SM simulations capture the dynamics in observed SM? Emphasis is given on the comparison of different SM measurement techniques due to their relevance for interpreting the evaluation results. Secondly, can SM simulations at higher spatial modelling resolution including refined spatial-resolution soil input data be provided with a consistent quality? Higher resolution does not necessarily improve the model performance, and may even worsen the quality of the simulation results. To assess this, the low-resolution model setup GDM-v1-2016 as well as the one kilometer setup GDM-v2-2021 are compared against multi-method

SM observations. Furthermore, drought characteristics estimated with both model setups are compared using annual drought intensities over the last 69 years (1952–2020).

## 2 Methods and Datasets

### 2.1 The mesoscale Hydrological Model (mHM)

The mesoscale Hydrological Model is a grid-based spatially distributed hydrological model driven by daily precipitation, temperature and potential evapotranspiration PET. It accounts for major hydrological processes such as snow generation and snowmelt, canopy interception, soil infiltration, ET, deep percolation, baseflow generation, and surface runoff routing. The open-source model code repository is available and is under active development and maintenance (https://git.ufz.de/mhm/mhm). The model uses three distinct levels to organize the modelling procedures: level 0 (L0) for input data of the sub-grid physical basin characteristics, level 1 (L1) for the realization of the integrated hydrological processes and level 2 (L2) for specification of meteorological forcing inputs. An unique component of mHM is the Multiscale Parameter Regionalization (MPR) technique (Samaniego et al., 2010) that allows inferring spatial variability of the required model parameters seamlessly on various modelling scales. One of the distinguishing aspects of the MPR approach compared to other regionalization techniques is to deliver a quasi scale-invariant model performance across modelling scales and improve the transferability of model parameters to ungauged basins (Kumar et al., 2013; Rakovec et al., 2016; Samaniego et al., 2017). The model was applied and evaluated in multiple climatological regions including Europe (Thober et al., 2015; Rakovec et al., 2016), West Africa (Dembélé et al., 2020), India (Saha et al., 2021) and the conterminous United States (Livneh et al., 2015; Rakovec et al., 2019). Within the MPR technique, the subgrid physical basin characteristics at L0 are linked to model parameters through transfer functions and a set of global parameters and subsequently upscaled to generate effective parameters at L1. The aggregation is based on a set of upscaling rules (e.g. arithmetic or harmonic mean) following flux conservation schemes (Samaniego et al., 2010).

A general overview on the model processes and parameterization can be obtained from Samaniego et al. (2010) and Kumar et al. (2013). Only the SM component of mHM is described here, due to its relevance for this study. The incoming precipitation and snowmelt are partitioned into root-zone SM and runoff components, depending on the degree of soil saturation, using a power function similar to the well-known HBV model (Samaniego et al., 2010). The degree of non-linearity depends on the underlying vegetation and soil characteristics following the MPR framework (Samaniego et al., 2010; Kumar et al., 2013). The evapotranspiration from soil layers is estimated as a fraction of the potential evapotranspiration depending on the SM stress and the fraction of vegetation roots present in each layer (Samaniego et al., 2010). The moisture stress function depends on the specification of soil-water content at a permanent wilting point, critical and saturation levels, which are determined using a set of pedo-transfer functions estimated within the MPR framework (Livneh et al., 2015; Zacharias and Wessolek, 2007).

## 2.2 Model setups at $4 \times 4$ km$^2$ and $1.2 \times 1.2$ km$^2$ spatial resolution

The new setup GDM-v2-2021, as used in the GDM version 2, includes several changes to the previous model setup GDM-
v1-2016. The main features of the two mHM setups that are used in the analysis are described in Table 1. While the GDM-
v1-2016 uses mHM version 5.6, in GDM-v2-2021 mHM was updated to version 5.10 (see https://github.com/mhm-ufz). The
implemented changes in mHM did not change the hydrological process representations related to SM that were used in the
simulations here. Between the setups GDM-v1-2016 and GDM-v2-2021, the projection system was changed from the projected
coordinate system Gauss-Krueger 4 (EPSG:31468) to the World Geodetic coordinate system (EPSG:4326). While the size of
the grid cells in the GDM-v1-2016 setup was fixed at $4 \times 4$km$^2$ (L1 level), the grid cell size in the GDM-v2-2021 setup is
measured in degree. As such, the grid cell size varies with latitude, with grid cell width in east/west direction decreasing from
1.23 km at 47.25° N latitude (south of Germany) to 0.98 km at 55.5° N latitude (north of Germany) and a constant grid cell
length of 1.7 km in north/south direction.

**Table 1.** Main features of the model setups GDM-v1-2016 and GDM-v2-2021. Core to the setups is the mesoscale hydrological model mHM. Vertical discretization of soil layers in the hydrological model mHM and projection system are denoted. In the spatial model resolution the Level 0 (L0) describes the subgrid variability of relevant basins characteristics. Level 1 (L1) and Level 2 (L2) describe the dominant hydrological processes and meteorological forcings, respectively. Datasets used as model inputs for soil as well as land use and geology on L0 model resolution are stated.

| Setup | spatial model resolution | soil dataset | vertical soil discretization | projection | land use dataset | hydro geology dataset |
|---|---|---|---|---|---|---|
| GDM-v2-2021 | L1 and L2: 0.01562° ×0.01562° eq. ~1.2×1.2km$^2$ L0: 0.001953125° × 0.001953125° | BUEK200 | 4 layers: 0–5 cm 5–25 cm 25–60 cm 60–variable cm | Latlon (EPSG:4326) | GLOBCOVER | GLIM |
| GDM-v1-2016 | L1 and L2: 4×4km$^2$ L0: 100×100m$^2$ | BUEK1000 | 3 layers: 0–5 cm 5–25 cm 25–variable cm | Gauss Krüger-4 (EPSG:31468) | CORINE | HUEK200 |

Soil texture (sand and clay fraction) and mineral bulk density are derived from national digital soil maps provided by the BGR (Federal Institute for Geosciences and Natural Resources). The BUEK200 dataset (BGR, 2020) used in the GDM-v2-2021 setup substantially increased the mapping resolution compared to the BUEK1000 dataset (BGR, 1998) used in the GDM-v1-2016 setup (scale 1:1 000 000 to 1:200 000). At the time of the creation of this study, the database version of BUEK200 was v0.5. Figure 1 shows the depth averaged clay contents for an exemplary region in Central Germany where SM observations that were used in the analysis are located. The soil map used for the study (BUEK), is the standardized basic soil mapping for Germany. It shows the distribution and association of soils and their properties in Germany. The map content is classified according to soil regions and soil landscapes. For each map unit a soil series is given composed of an index soil (dominant soil) and accompanying soils. For modeling, the soil properties of the index soil within the spatial mapping unit were used to derive the model parameters.

The soil depths in mHM are discretized into an upper soil layer at depth 0–25 cm, including a top layer at depth 0–5 cm, and the remaining depth of the soil profile. In the GDM-v2-2021 setup an additional layer at 25–60 cm was added due to stakeholder feedback, mainly from the agricultural sector. The tillage depth is set to 30 cm in both model setups. The land use datasets used in the model setups GDM-v1-2016 and GDM-v2-2021 were CORINE (EEA, 2009) and GLOBCOVER (ESA, 2009), respectively. Hydrogeological input data that define the aquifer properties and govern the baseflow recession rates, was derived from the HUEK200 database for GDM-v1-2016 (BGR, 2009) and the GLIM database for GDM-v2-2021 (Hartmann, Jörg and Moosdorf, Nils, 2012). Digital elevation models were derived from BKG (2010) and USGS (2017) respectively.

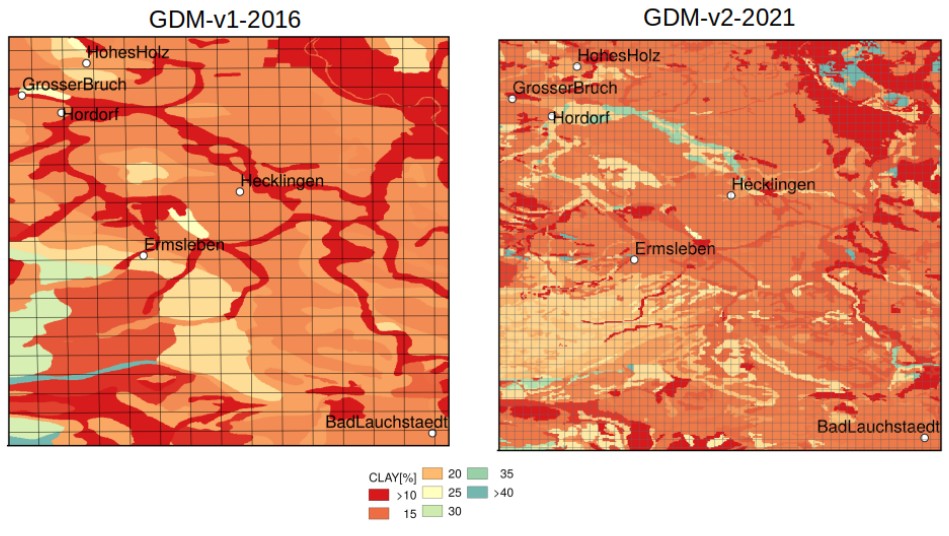

**Figure 1.** Average derived clay [%] over the soil column from the BUEK1000 soil dataset used in the GDM- v1-2016 model setup (left panel) versus the BUEK200 soil dataset used in the GDM-v2-2021 model setup (right panel). The grid shows the respective modelling resolution L1 at which the hydrological processes are simulated (see table 1). Both setups are projected in WGS 84 (EPSG:4326).

### 2.2.1 Meteorological input data

Precipitation, as well as minimum, maximum and average air temperature, are interpolated on a daily timescale based on meteorological station data from the German Weather Service (DWD) using external drift kriging (EDK) with elevation as the drift variable. The meteorological station data is subject to extensive quality controls (Kaspar et al., 2013). Additionally, quality controls such as checking the plausible variable range are implemented in the preprocessing steps of the interpolation routine. Theoretical variograms are estimated based on all available station data to derive seamless fields of hydro-meteorological fluxes and states for entire Germany (Zink et al., 2017). An exponential model is used for precipitation and spherical models for the temperature variables. The interpolation method and variogram parameter estimation for Germany are described and evaluated in detail in Zink et al. (2017) including cross-validation metrics and comparison to the comparable REGNIE gridded dataset by the German Weather Service (Rauthe et al., 2013). PET is calculated using the Hargreaves-Samani Method (Hargreaves and Samani, 1985) that is based on the interpolated temperature fields (average, minimum, and maximum) and (potential) extraterrestrial radiation, which is computed depending on the latitude of the location and day of the year.

### 2.2.2 Multibasin model calibrations

The unknown parameters of the mHM setup GDM-v2-2021 were calibrated against observed discharge using the Kling-Gupta efficiency (KGE; Gupta et al., 2009) as the objective function. The parameter optimization was conducted using the Dynamically Dimensioned Search (DDS; Tolson and Shoemaker, 2007) algorithm with 1 000 iterations, which underwent detailed scrutiny, as follows. In a first step, 200 parameter sets were obtained using a multi-basin/domain-wide joint basin calibration strategy, in which a subset of six basins was randomly selected (out of 201 total basins) and then jointly calibrated during a common period of 1990–2005 (see Table S1 in the Supplements). Subsequently, all 200 parameter sets were evaluated against the full ensemble of 201 basins during an extended period of 1986–2005 (with a warming period of 5 years). The parameter set with the best performance in terms of the median daily KGE over 201 basins was selected and used for the consequent analysis (See Table S2 in the Supplements). This updated approach is based on the earlier calibrations of the GDM-v1-2016 setup (Zink et al., 2017) in which the Nash–Sutcliffe efficiency instead of the KGE was applied and individual single-basin instead of the multi-basin calibrations were carried out as input to the model cross-evaluation at locations that were not used for model calibration. Previous works also focused on multibasin calibrations of mHM in other regions, such as Mizukami et al. (2017); Rakovec et al. (2019). The model performance of the best cross-evaluated parameters of the GDM-v2-2021 based on daily streamflow from 201 catchments in Germany yielded a median performance of 0.761 KGE (see Fig. A1).

### 2.3 Soil moisture observations

The SM observations used to conduct the model evaluations were gathered from the environmental observation networks TERENO (Zacharias et al., 2011) and FLUXNET (FLUXNET2015 Dataset; Pastorello et al., 2020) as well as from the Cunnersdorf site operated by the DWD. In total, SM data from 40 locations were compiled and processed for the analysis (see Fig. 1). Although it is not feasible to establish an evenly distributed grid of SM measurements on a national level (Vereecken

et al., 2008), the available locations cover a wide range of climatic and vegetation conditions in Germany.

In total, we analysed 46 measurements from 24 grassland sites, nine crop sites, six forest sites and one site containing a forest clearing. Four of the sites have multiple measurement methods available, which allowed comparing the evaluations between the measurement methods at single sites. The elevation ranges from 4 to 1 252 meters a.m.s.l., and the long-term yearly precipitation sums range from below 500 mm to more than 1 500 mm. Time series lengths of the observations are between 2.8 and 17.8 years with a median (mean) of 6.5 (6.7) years. A detailed overview of the location characteristics is shown in Table 3.

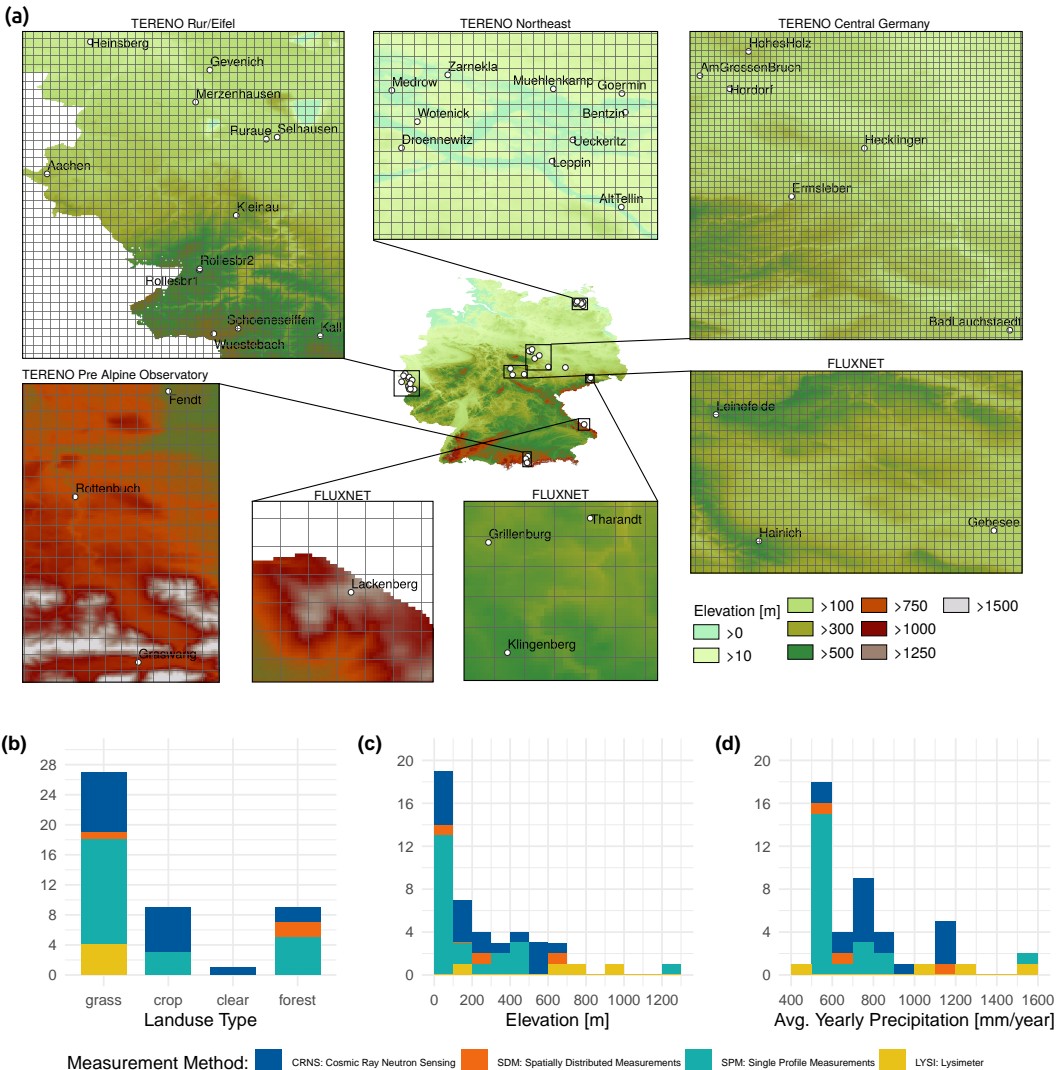

**Figure 2.** SM observations of 40 locations distributed over Germany were used in the SM evaluations of the GDM-v2-2021 and GDM-v1-2016 model setups. The subplots display the experimental sites in greater detail, representing different climate gradients in Germany. The maps show the digital elevation model on the hydrological subgrid variability resolution L0 of mHM in the GDM-v2-2021 setup ($0.001953125° \times 0.001953125°$). The grid corresponds to the modelling resolution L1 in the GDM-v2-2021 setup ($0.01562° \times 0.01562°$, which equals $\approx 1.2 \times 1.2 \, \text{km}^2$) at which the hydrological processes are simulated. The lower panel shows the distribution of different SM observations depending on land use type, elevation and average yearly precipitation. Note that some of the 40 locations have multiple SM data sources ($n = 46$).

185    The data is comprised of four different SM measurement methods. SM observations from seven FLUXNET and 16 TERENO sites in Germany based on several vertically distributed sensors within one soil profile were used (in the following abbrevi-

ated SPM). The sensor depths are described in Table 3. SPM sites used from TERENO-Northeast Observatory are further described in Itzerott et al. (2018a) and Itzerott et al. (2018b). SM data from lysimeters are available for four sites from the TERENO-SOILCan lysimeter network (Pütz et al., 2016) at the Bad Lauchstädt experimental site and the TERENO Pre-Alpine Observatory (PAO) (Kiese et al., 2018). Lysimeters are large vessels containing an undisturbed soil column to allow gravimetric measurements. Since the lysimeter vessels are closed at the bottom, water tension is adjusted to reference measurements at the same depth in the undisturbed soil close to the lysimeter (Pütz et al., 2016; Kiese et al., 2018). In the lysimeters, SM is measured by single sensors in multiple depths. At each SOILCan-site multiple lysimeters are organized in hexagons (Pütz et al., 2016). The number of lysimeters per site are described in Table 3.

For three of the 40 sites (Am Grossen Bruch, Hohes Holz and Wüstebach), spatially distributed measurements (SPM) of SM are available. Multiple sensors are installed in a spatial grid at different depths covering an area of some hundreds of square metres. For the locations Hohes Holz and Am Grossen Bruch 39 and 20 profile with sensors at multiple depths were used respectively (depths varied slightly between profiles depending on soil property changes). Therefore, they are not denoted explicitly. For the Wüstebach site, 51 profiles with two sensors each at 5 cm, 20 cm, and 50 cm depth were used (Wiekenkamp et al., 2019). The Wüstebach SM measurement network is described in detail in Bogena et al. (2018).

SM observations derived from Cosmic Ray Neutron Sensing (CRNS) stations were used from 17 sites (see Table 3) of the TERENO observatories (Bogena et al., 2022). The soil albedo component of cosmic-ray neutrons is particularly prone to changes of SM (Desilets et al., 2010; Köhli et al., 2021). However, since neutrons are sensitive to all pools of hydrogen, the measured neutron signal is also affected by biomass (Baatz et al., 2015), intercepted water (Bogena et al., 2013; Schrön et al., 2017), and snow (Schattan et al., 2017) and therefore requires a correction of the measured signal in this respect. In this study, periods of snow cover have been excluded from the CRNS data. SM from CRNS data has been calculated by standard methods (Desilets et al., 2010; Zreda et al., 2012) and aggregated to daily time steps. This leads to typical statistical uncertainties of less than 3 vol.% (Schrön et al., 2018).

All SM data was checked according to their flagging conventions for doubtful or low-quality values. In some cases, doubtful data was removed manually after personal communication from site maintainers (e.g. some sites from the TERENO PAO lysimeter sites showed doubtful data after frost in early 2017). The available SM data in the respective depths, as noted in Table 3, was aggregated to weighted vertical averages according to the soil discretization depths in mHM (0–25 cm, 25–60 cm and 0–60 cm). Highest weights were allocated when the sensor depth is located in the center of the soil depth range and weights linearly decrease towards the edges of the soil depth range. The spatial mean values were calculated for the SDM measurements based on the available sensors.

## 2.4 Soil moisture data preparation and evaluation metrics

Since the computation of SM drought indices, including the estimation of SM probability distributions by kernel density estimates, is hampered for the available observed data due to the limited length of observed SM data (<10 years for most locations), the analysis here is based on a comparison of observed to simulated SM (e.g., Samaniego et al., 2013). It is widely known that absolute SM values cannot be adequately determined by a regional model (partly due to the spatial heterogeneity),

yet the hydrological model typically captures the temporal dynamics well (Koster et al., 2009). As drought is defined by the deviation from normal conditions, SM anomalies were calculated. To preserve the units of volumetric SM [mm/mm] and the original range of SM dynamics, standardization by dividing standard deviation was not undertaken in this study. The anomalies are calculated in two ways. First, the mean of all values in each SM time series is subtracted.

$$\theta(anom)_{i,k} = \theta_{i,k} - \overline{\theta} \tag{1}$$

Secondly, the multi-year mean for each day of the year is subtracted to deseasonalize the anomalies. The removal of the annual average cycle of SM is necessary for the subsequent drought classification based on percentile thresholds as described in the next section.

$$\theta(deseas-anom)_{i,k} = \theta_{i,k} - \overline{\theta}_i \tag{2}$$

where $i$ is the calendar day of the year (DOY 1, ..., 365) and $k$ is the year. To reduce uncertainty in the mean resulting from heterogeneous and small sample sizes, for each $i$ a moving window with 15 days on each side of the day was used to increase the sample size and the multiyear mean of each $i$ was calculated based on the mean of randomly drawing 500 bootstrap samples. Since there are some data gaps in the observed data (see Table 3 for data availability), the simulated data was masked to the available observed data to allow a comparable calculation of SM seasonality. Leap days were removed before calculating the deseasonalized anomalies.

The evaluation of observed against simulated SM is based on the Spearman rank correlation coefficient (R). The Spearman rank correlation coefficient is a non-parametric measure to quantify the strength of the monotonic relationship between two variables. The correlations are calculated on whole data records as well as on sub-periods (months, seasons and vegetative active period) to investigate the seasonal variability in the performance metrics. Paired Wilcoxon signed-rank tests were conducted to identify significant changes between the model setups.

### 2.4.1 Soil moisture index computation and analysis

Simulated SM by the two model setups is used to compute a Soil Moisture Index (SMI) following Samaniego et al. (2013) and Zink et al. (2016) enabling a SM drought analysis based on long term SM data. The SMI for a given cell and day is estimated as

$$\mathrm{SMI}_t = \hat{F}_T(x_t) \tag{3}$$

and it represents the quantile at the SM fraction value $x$ (normalized against the respective saturated soil water content). $x_t$ denotes the simulated monthly SM fraction at a time $t$ and $\hat{F}_T$ is the empirical distribution function estimated using non-parametric kernel density estimates. The optimal bandwidths are estimated by minimizing a cross-validation error estimate. Details regarding the computation of the SMI can be found in Samaniego et al. (2013).

The SMI drought threshold concept used in the German Drought Monitor is based on the D0-D4 classification system for droughts from the US-Drought monitor (Svoboda et al., 2002) that related drought categories to potential impact types. The

drought thresholds reflect the occurrence of similar SM conditions in the past and hence indicate the potential impacts of these conditions (Zink et al., 2016). A cell at time $t$ is under drought when $\text{SMI}_t < \tau$. Here, $\tau$ denotes that the soil water content in a cell is less than the values occurring $\tau \times 100\%$ of the time. The 20th percentile used as $\tau$ in this study is defined as moderate drought conditions, that indicate conditions of "possible damages to crops and pastures". Extreme drought conditions are defined as the 5th percentile indicating "high probability of major losses in crops and pastures". The resulting impact of SM drought conditions need to be identified for each specific impact type based on the timing within the year and duration of drought conditions. For example, Peichl et al. (2018, 2021) identified specific monthly damage functions between the SMI and different crops using varying statistical methods. The work showed that dry SM anomalies in some months can reduce yield (e.g., August, September for maize), while in other months it may increase crop yield (e.g., May for maize). Impacts of SM droughts can affect a broad range of sectors besides agriculture. Especially, the considered soil depth of the SMI is relevant for different sectors. While the drought conditions in the upper soil (0-25 cm and 0-60 cm) are more relevant to agriculture, drought in the total soil column (up to 2 meters) indicate potential impacts on water resources and the forestry sector.

SMI based drought statistics are calculated for the years 1952–2020 on fixed temporal (annual and vegetative active period from April to October) and spatial scales (per grid cell and aggregated for Germany). When calculating the cumulative density functions of SM, a common statistical basis of 1951-2015 was used for both model setups. The drought intensities (DI) per year are calculated by

$$DI = \frac{1}{d * A} \sum_{t_0}^{t_1} \int_A [\tau - SMI_i(t)]_+ \tag{4}$$

with the area of interest $A$ (here Germany) and duration $d$ $(t_1 - t_0)$ in days (annual $t_0$ Jan 1st to $t_1$ Dec 31st and vegetative active period $t_0$ Apr 1st to $t_1$ Oct 31st ). The drought intensities take into account the degree of negative departure from drought conditions (hence, the extremer the drought conditions, the higher the intensities) as well the temporal aggregation length and the spatial aggregation area. The area under drought is calculated as the percentage of grid cells where SMI $< 0.2$ averaged over the respective temporal periods.

## 3   Results and Discussion

In the following sections, the comparisons of the multi-method SM observations with two hydrological model simulations are presented and discussed to investigate the proposed research objectives. In section 3.1, a comparison of SM observations to the simulations from the high-resolution operational model setup GDM-v2-2021 is shown. The setup allows a comparison of observations to the 0–25 cm layer as well as to the additional, deeper soil layer of 25–60 cm. In section 3.2, the differences between the two simulation setups are shown for annual drought intensities during 1952–2020 and compared to SM observations. The two mHM simulations are used in their operational setups, meaning that only data and information available for the whole of Germany was used. Additional available information on soils or meteorological measurements at the observation sites was not incorporated in the simulations.

## 3.1 Comparison of high resolution simulations against observed SM dynamics

Here, $1.2 \times 1.2$ km$^2$ simulations in two soil layers (GDM-v2-2021) are compared to SM observations using four different measurement methods: Cosmic Ray Neutron Sensing (CRNS), spatially distributed measurements (SDM), single profile measurements (SPM) and lysimeter (LYSI). SM anomalies as well as deseasonalized SM anomalies are used.

Figure 3 shows the results for three selected locations that contain both CRNS and SDM measurements for the six-year period 2014–2019. In general, the SM anomalies and deseasonalized data agree well, with a small reduction of correlations for the deseasonalized data. Furthermore, observations and simulations agree well both in the uppermost soil layer (0–25 cm and in the deeper layer (25–60 cm depth). The correlation strength between simulations and observations from different measurement techniques is similar for the sites Am Grossen Bruch and Hohes Holz, but deviates more for the Wüstebach site. It is worth noting that different spatial scales are mapped by those measurements. While the SPM (not included here) represents point information, the SDM and CRNS cover an area less than 0.1 km$^2$ and the mHM simulations cover an area of $\approx 1.44$ km$^2$. In general, the day-to-day variability is lower in simulations than in observations. At the forest sites Wüstebach and Hohes Holz, the day-to-day variability in the CRNS data is higher than in SDM. Several environmental factors other than SM can influence the CRNS signals (see Methods). While the changing biomass might have a low impact on the signal, it can introduce a (constant) systematic bias. Since only anomalies are analysed here, the impact of such bias on this (comparative) anomaly analysis should be minimal. Intercepted water on leaves and in the litter layer can be particularly challenging to quantify, especially in forested stations such as Hohes Holz or Wüstebach (Bogena et al., 2013; Schrön et al., 2017). It might lead to stronger dynamics in the CRNS signal during and shortly after rain events, in comparison to the model output or other observation methods. Additionally, partial deforestation in 2013 at the Wüstebach site modified SM flows resulting in stronger response to rainfall (Wiekenkamp et al., 2019). Nevertheless, there is no general tendency for lower correlations at forest sites than crop and grassland sites (see Figure 4). Crop sites show slightly lower correlations than grassland sites, which is expected since anthropogenic activities (e.g., crop rotation) are not represented in mHM. Correlations display no clear tendency across the range of elevation and precipitation regimes. In general, Figure 4 reveals that the model performance does not systematically depend on site conditions. Moreover, no systematic relationship between correlations and the length of the time series can be found (see Figure 4 d)).

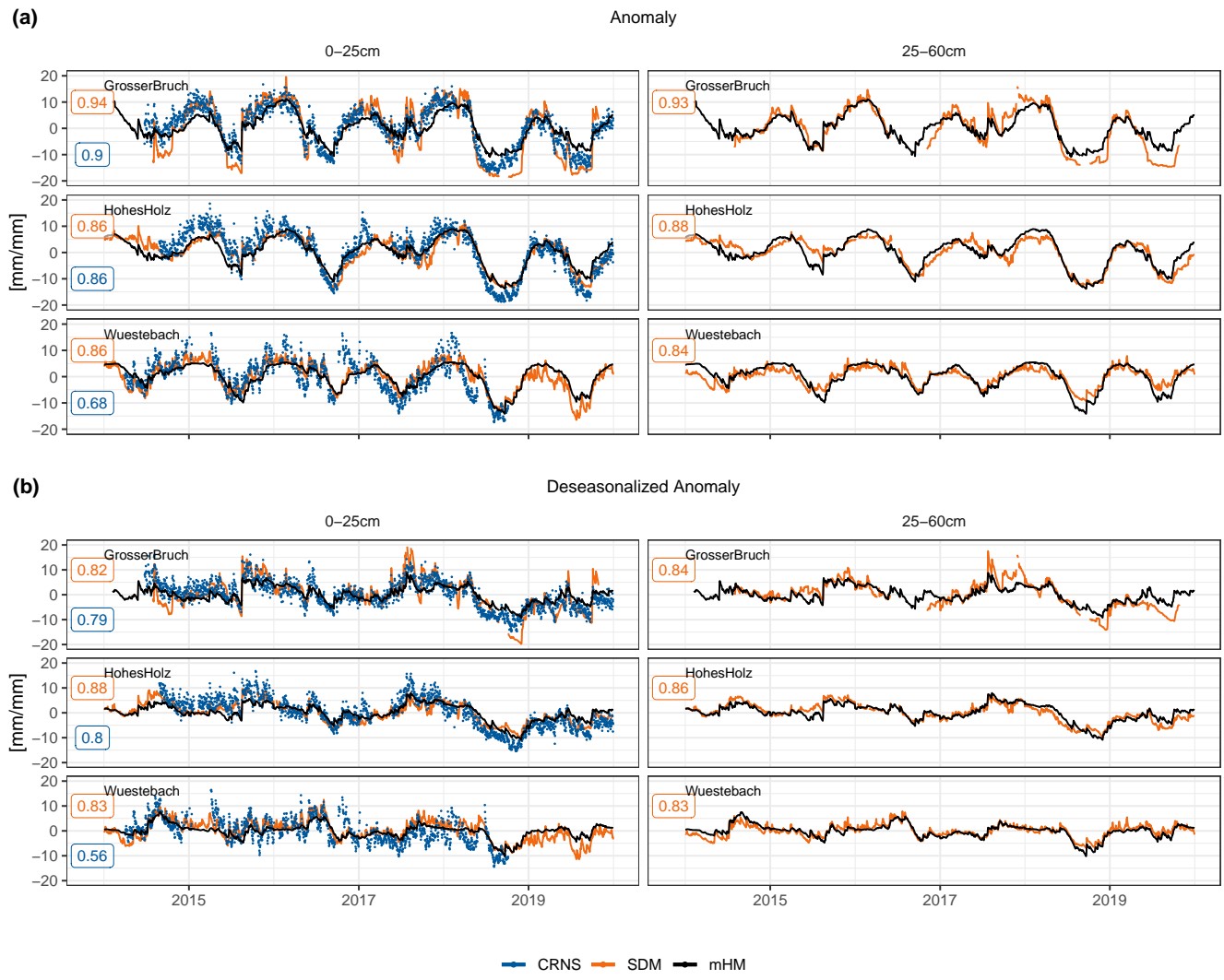

**Figure 3.** SM time series for 2014–2019 for the selected locations Am Grossen Bruch, Hohes Holz and Wüstebach, showing SDM and CRNS data against simulated data from mHM in 0–25 cm and 25–60 cm depth in the GDM-v2-2021 setup. The Hordorf site also contains both CRNS and SDM measurements, but with much shorter time series length. The stations with longer time series were selected for visualization. Spearman rank correlation coefficients are denoted at the left side of each time series. Panel (a) shows SM anomalies including seasonality and panel (b) shows deseasonalized SM anomalies.

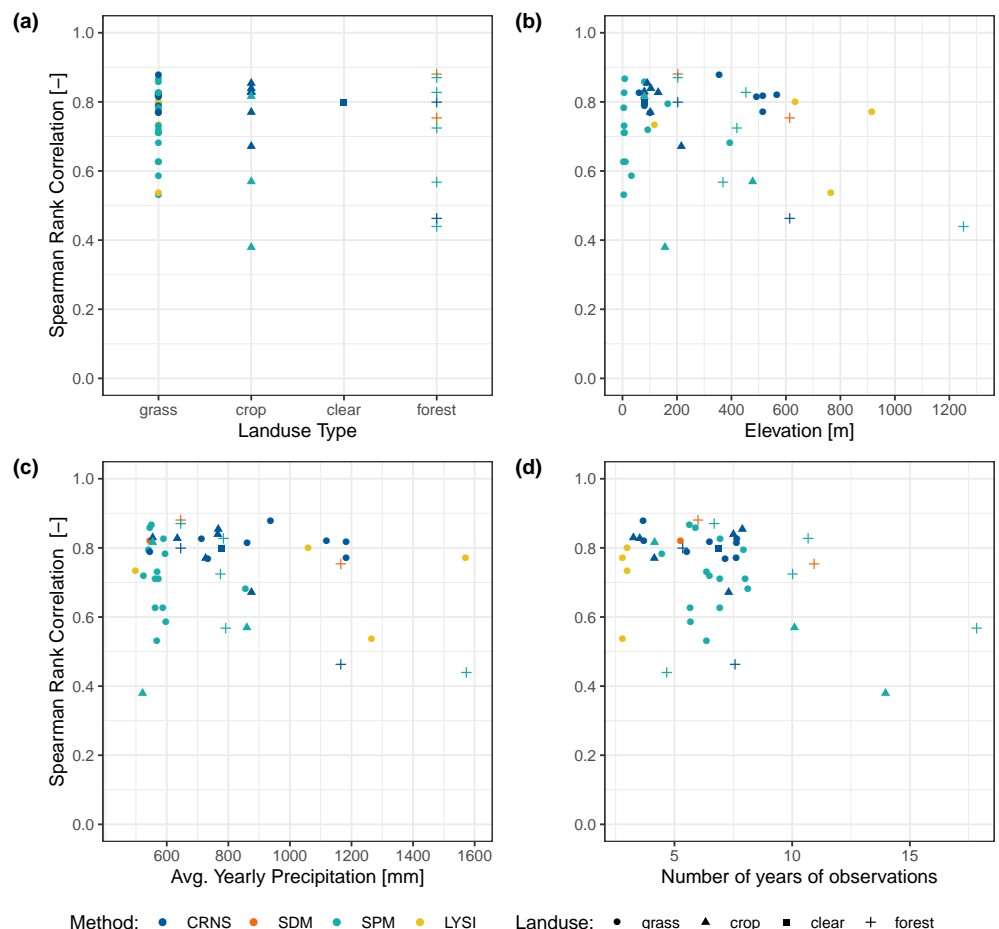

**Figure 4.** Spearman rank correlation coefficients of the simulated versus observed deseasonalized SM anomalies against site characteristics: (a) land use, (b) elevation and (c) average yearly precipitation and (d) length of the time series. See Table 3 for detailed overview per location. Colors denote the SM data method (Cosmic Ray Neutron Sensing (CRNS), spatially distributed measurements (SDM), single profile measurements (SPM) and lysimeter (LYSI)) and shapes the land use types reported at the locations (abbreviated as following: grass = grassland, clear= forest clearing, LYSI=Lysimeter).

Monthly Spearman correlation coefficients for all locations and measurement methods at 0–25 cm depth are shown in Fig. 5. The correlation coefficients show an apparent clear seasonal variation, with the highest values in summer/autumn months and the lowest values in winter. The highest median correlation is detected in August (0.87), while the lowest median correlation is found in January (0.37). The spread of the correlation coefficients within the different locations is largest in winter months, with some locations have correlations close to 1.0. While in February and March, some locations with CRNS, SPM and LYSI measurements show correlations below zero. The intra-annual variation of performance metrics was similar to the findings of Xia et al. (2014), that extensively evaluated simulated SM from four different hydrological models (Noah, Mosaic, SAC, VIC) in the North American Land Data Assimilation System phase 2 (NLDAS-2) dataset, which is used for drought monitoring in

the United States and similarly observed generally higher correlations in summer and lower correlations in winter. The lower correlations observed in winter could be related to higher uncertainties in simulations and observations with respect to frozen soils and snow cover. The sensor quality of SDM, SPM and LYSI in winter can be reduced during frost days. Especially, SPM and LYSI measurements can be affected by sensor failures as they rely only on few sensors compared to the spatially distributed measurements (SDM) with a larger number of sensors. Annex Figure A3 shows correlations between simulated SM, CRNS and SDM. Low correlations between simulation and observations are accompanied by low correlations between the measurement methods, especially in winter. In a climate impact study investigating low flows over Europe, it could be shown that uncertainty due to the selection of the hydrological model dominates the overall uncertainty including the meteorological drivers in snow dominated areas (Marx et al., 2018). Furthermore, the mHM does not contain a full energy balance model, which limits the description of soil frost depths.

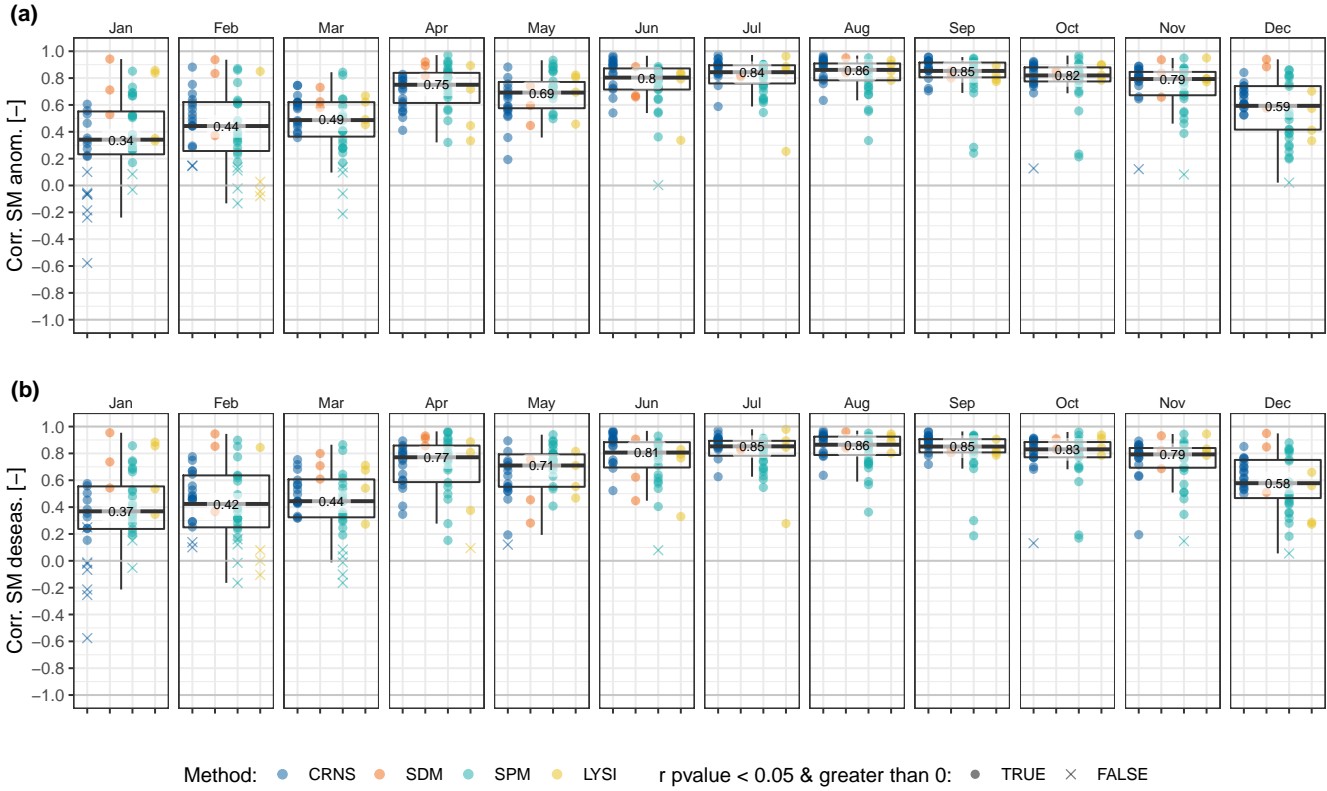

**Figure 5.** Comparison of the simulated mHM SM (0–25 cm) in the GDM-v2-2021 setup to observed SM anomalies without (a) and with (b) subtraction of the mean seasonal SM cycle for each month depicted with boxplots. SM anomalies are plotted for each location and coloured according to the SM measurement menthod used. Note that sample sizes between measurement methods differ (CRNS: n=17, SDM: n=3,SPM: n=23; LYSI: n=4). Data points that are not both significantly (p-value < 0.05) and positively correlated are marked with x. See Figure A2 for detailed comparisons.

Observations using different SM measurement methods display considerably different correlations. The SPM generally vary more, with large variation in winter and the presence of low-performance outliers in summer months. CRNS measurements show a consistently high performance in summer months but notably low correlations in winter (especially January). As snow days were removed from the time series in the CRNS measurements, the anomaly calculation from the remaining data was impacted by a smaller sample size. Another reason for the lower correlations observed might be due to the variable penetration depth of CRNS, which ranges between 15 to 70 cm depending on SM (Köhli et al., 2015; Schrön et al., 2017). This could introduce systematic and temporally variable errors and affect the correlation between observed and simulated soil water content (Baroni et al., 2018). Comparison to the mHM top soil (0–25 cm) layer is assumed to remain a good compromise, since the soil water distribution is rather homogeneous between 0 and 25 cm under wet conditions. Under dry conditions the footprint is deeper and more heterogenous, but the highest sensitivity is in the upper soil layers (exponential sensitivity). The SDM measurements show the most consistent performance across all months with the exception of May, as illustrated in Figure A2 a). All measurement methods at these sites show a drop in correlations to the SM simulations in May and June, while the observations have higher correlations between each other (see also Figure A3). This points to deficiencies in the model, which may be related to the static vegetation module in mHM which does not include processes such as possible early onset of the growing season and consequent earlier depletion of the soil water storage. Moreover, lower correlations of deseasonalized anomalies in May are detected especially at forest locations (median of 0.63 over all forest locations). The timing of leaf unfolding in trees, usually between late April to May (Chen et al., 2018), is subject to annual fluctuations and affects evaporation from the soil and therefore SM dynamics. Figure A2 b) depicts the SPM data from FLUXNET and TERENO separately, which cover the time periods 1997–2014 and 2011–2019, respectively. The seasonal variation of the correlations is in good agreement among both monitoring networks and time periods. The performance at TERENO sites is generally higher than at the FLUXNET sites, possibly due to a larger number of sensors installed along the soil depth in the TERENO sites which may improve the vertical averaging of SM (see Table 3).

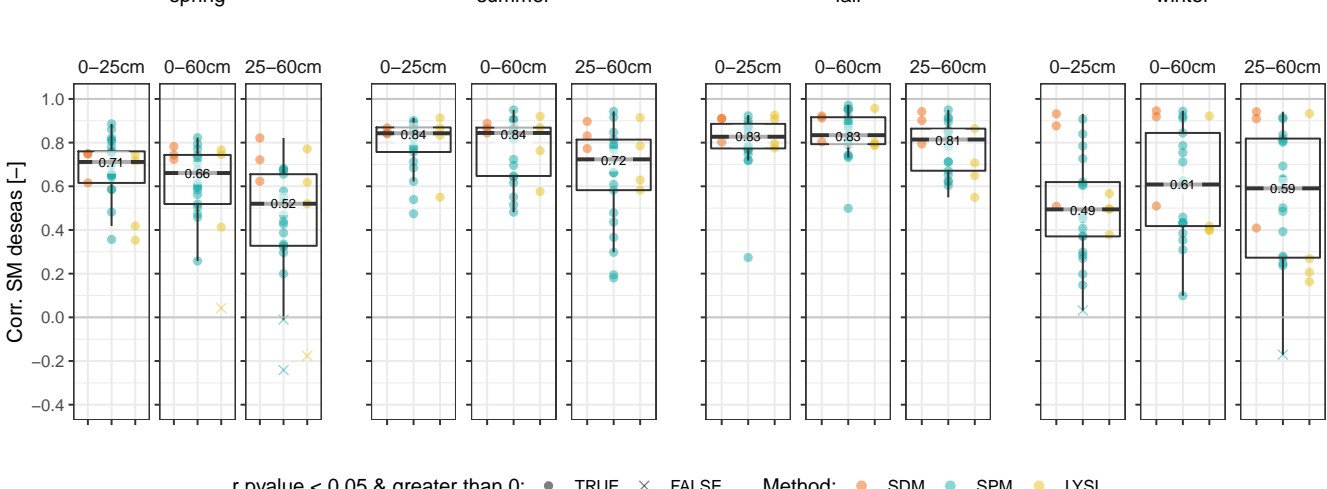

**Figure 6.** Spearman correlation coefficients of the simulated mHM SM in the GDM-v2-2021 against observed deseasonalized SM anomalies depicted with boxplots for three depths (0–25 cm, 0–60 cm, 25–60 cm). Values of each location are plotted and colours denote the measurement method of the SM data (SDM: spatially distributed measurements; SPM: single profile measurements; LYSI: Lysimeter). Here only locations with measurements at 25–60 cm depth are taken into account (SDM: n=3,SPM: n=19; LYSI: n=4). Data points that are not both significantly (p-value < 0.05) and positively correlated are marked with x.

The Spearman correlation coefficients for each season and soil depth is depicted in Figure 6. Note that here only locations that have SM data in all depths were considered and CRNS data was excluded as its varying penetration depth does not allow a consistent depth-wise evaluation. This leads to a smaller sample sizes of locations (n=26). Figure 6 shows that the median correlation is lower for the deeper SM simulations for all seasons, except for winter. In spring, the lower depth also shows the strongest negative difference to the upper depth in comparison to summer and autumn (Spring $\Delta - 0.19$, Summer $\Delta - 0.12$, Fall $\Delta - 0.02$). The correlations vary more in the 25-60 cm depth between locations in all seasons, with more outliers of very low correlations observed. Since the mHM was conceptualized for dominant processes at the large scale (mesoscale), not all processes that are important at the local scale are currently accounted for (e.g. species specific root water uptake, lateral flow or groundwater soil water interaction). For instance, Rosenbaum et al. (2012) showed for the distributed SM measurements at the Wüstebach catchment that SM dynamics in the topsoil (5–50 cm depth) are influenced by groundwater. Processes of capillary rise are not modelled in mHM, hence it is expected that agreement to simulated SM by mHM at sites with groundwater-influence is lower compared to groundwater-distant sites. This effect should increase with depth due to increasing groundwater influence that could explain the lower correlations in the 25–60 cm depth. SPM sites might be more affected by this than SDM in which these effects can be averaged out. Identification of groundwater characteristics at each measurement site was, however, out of scope for this study. The SDM overperform SPM and LYSI, with higher than average correlation values especially for the 25–60 cm depth, which underlines the assumption that local characteristics of single sensors e.g. groundwater influence and the resulting spatial variability of SM (Famiglietti et al., 2008) are averaged out over a larger area and generally supports

365 the closer scale match of SDM measurements and the $1.2{\times}1.2$ km$^2$ simulation grid cells. It has to be noted that the SPM at the same sites as the SDM also show comparable high correlation values (for an overview of the locations, see Table 3).

## 3.2 Comparison of different mHM model setups

In the following, the comparison between observations and the two model setups GDM-v1-2016 and GDM-v2-2021 (i.e., GDM version 1 and 2), as well as drought metrics between the two simulation setups are shown and discussed. Table 2 shows the me-
370 dian values of the Spearman correlation coefficients for selected sub-periods (seasons, vegetative active period April–October) and for the full year. Considering the observed SM data from all locations and measurement methods, the median correlations between the two simulation setups slightly increase by +0.05 in GDM-v2-2021. The results on an seasonal scale show a small decrease in the correlations in spring ($\Delta - 0.03$) and summer ($\Delta - 0.01$), but a significant increase of correlations in fall ($\Delta + 0.07$) and winter ($\Delta + 0.12$) in the new model setup. Several of the changes in the model setup may provide explanations
375 for the improved model agreement to observed SM dynamics in fall and winter. The higher modelling resolution of the 1 km runs may better resolve the sub-grid variability of cold season related processes such as snow accumulation that improves the simulated SM dynamics. As well, the finer spatial soil texture representation possibly contribute to an improved model representation of soil wetting/drying e.g. especially during saturated conditions in the cold season. When analysing the metric over the vegetative and non-vegetative active period (defined from April–October and November-March, respectively), the increase
380 in median correlations is +0.03 and +0.10 respectively. Median correlations using CRNS and SPM measurements support the overall findings. In general, the results show that the CRNS yield higher median correlation than the SPM measurements for both model setups, except for spring. While the median correlation in winter increased by $+0.17$ between GDM-v1-2016 to GDM-v2-2021 for CRNS, there is only a small increase in correlation of $+0.03$ for SPM. Similar results showing an overall increase in simulation performance were found by Albergel et al. (2012). In their study, the EMCWF operational and re-analysis
385 SM product using the hydrological model H-TESSEL was improved due to changes in the soil hydrology in the model and an increase of model resolution. They concluded that a better representation of soil texture might obtain further improvements. Furthermore, De Lannoy et al. (2014) found moderate improvements in the agreement of SM simulations compared to observations through implementing updated soil texture information.

**Table 2.** Median Spearman rank correlation coefficients R of simulated (GDM-v1-2016, GDM-v2-2021) versus observed deseasonalized SM anomalies at depth 0–25 cm. The correlation coefficients are calculated annually, seasonally (spring=March, April, May; summer=June, July, August; fall=September, October, November; winter=December, January, February) and over the vegetative and non-vegetative active period (defined from April–October and non-veg November–March respectively). Note that some of the 40 locations have multiple SM data sources available, resulting in $n = 46$; see Table 3. Stars denote significant differences (p-value $< 0.05$) in correlations between the model setups according to paired Wilcoxon signed-rank test.

| metric | method | setup | annual | spring | summer | fall | winter | non-veg | veg |
|--------|--------|-------|--------|--------|--------|------|--------|---------|-----|
| | ALL | GDM-v2-2021 | 0.78 | 0.65 | 0.85 | 0.84 | 0.49 | 0.59 | 0.84 |
| | (n=46) | GDM-v1-2016 | 0.73 | 0.68 | 0.86 | 0.77 | 0.37 | 0.49 | 0.81 |
| | | Δ | +0.05 | -0.03 | -0.01 | +0.07 (*) | +0.12 (*) | +0.10 (*) | +0.03 |
| R [-] | CRNS | GDM-v2-2021 | 0.81 | 0.63 | 0.88 | 0.86 | 0.60 | 0.65 | 0.86 |
| | (n=17) | GDM-v1-2016 | 0.79 | 0.67 | 0.88 | 0.80 | 0.46 | 0.48 | 0.84 |
| | | Δ | +0.02 | -0.04 | 0.0 | +0.06 (*) | +0.14 (*) | +0.17 (*) | +0.02 |
| | SPM | GDM-v2-2021 | 0.72 | 0.67 | 0.79 | 0.78 | 0.39 | 0.45 | 0.82 |
| | (n=23) | GDM-v1-2016 | 0.71 | 0.69 | 0.80 | 0.76 | 0.34 | 0.42 | 0.77 |
| | | Δ | +0.01 | -0.02 | -0.01 | +0.02 | +0.05 | +0.03 | +0.05 |

Spearman rank correlations between simulated and observed deseasonalized SM anomalies that fall below the 20th percentile in the observed SM time series are shown in Figure 7 to specifically analyse the dry anomaly spectrum. It is important to emphasize that we do not aim to estimate drought periods here, as its solid calculation requires a much longer time series. The drought estimation is performed using histograms for every grid cell and day of the year (see method section 2.4.1). Consequently, estimating robust percentiles requires time series lengths of minimum 30 years – this means that the time series length of the observational data is considered insufficient. Figure 7 a) shows a median correlation of 0.61 over all observations in the GDM-v2-2021 setup. The performance in the two model setups remains similar. However, the comparison separated between the measurements with larger spatial footprint (SDM, CRNS) and point scale measurements (SPM, LYSI) shows that the agreement between simulations and the larger footprint observations increased towards the high resolution setup, but the median agreement to the point scale SM measurement decreased. In general, the measurements with larger spatial footprint display higher agreement to the simulations. Due to the varying day-to-day variability of SM between the SM observation types and the simulations, in Figure 7 b) additionally a statistical smoothing was applied by calculating a running 30 day mean on the daily SM time series before subtraction of the seasonal SM cycle. This approach is similar to the SM preprocessing for the SMI as proposed in Zink et al. (2016). Figure 7 b) shows when smoothing is applied, the agreement between observations and simulations during dry periods can be substantially improved to a median correlation of 0.7 over all observations in the GDM-v2-2021 setup ($\approx$ 1-km resolution). Especially the agreement between the point scale measurements and simulations is increased to a median correlation of 0.63 in both model setups.

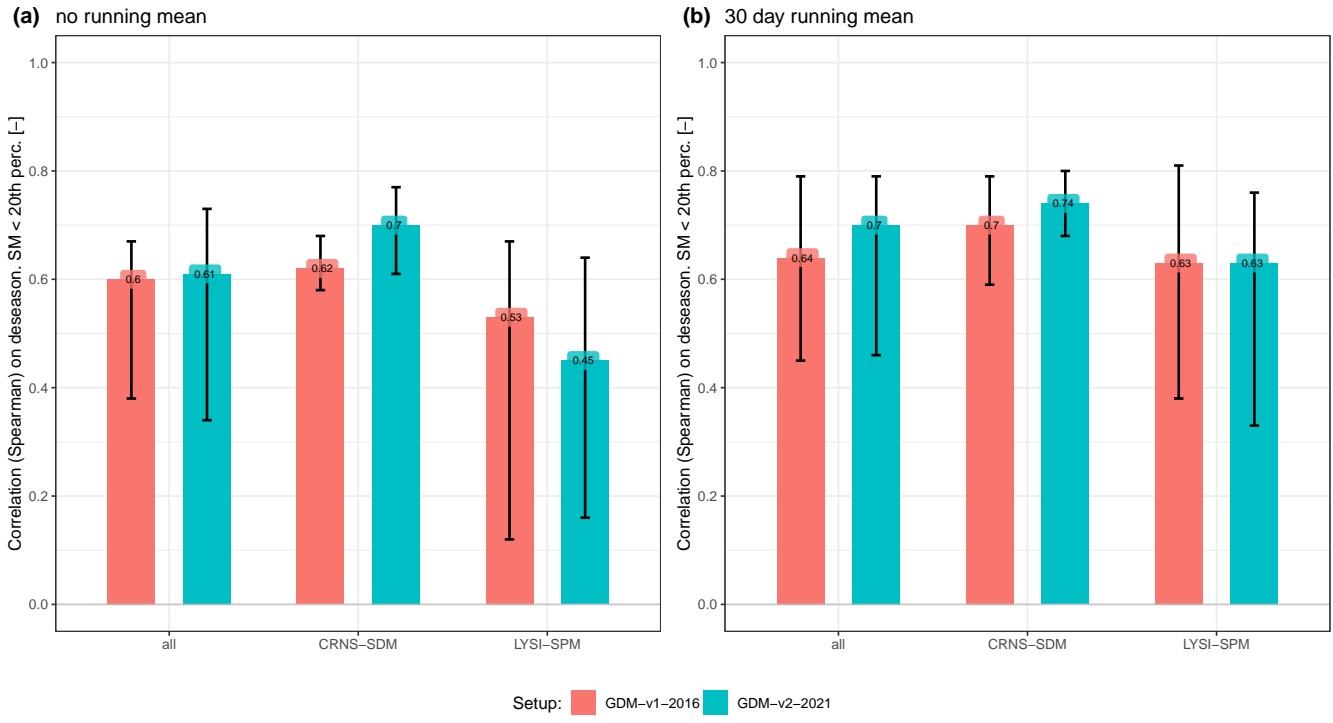

**Figure 7.** Correlations of deseasonalized daily SM below the 20th percentile (based on the observed SM time series) between simulations and observations (a). In (b) additionally a statistical smoothing was applied by calculating a running 30 day mean on the daily SM time series before subtraction of the seasonal cycle. The correlations are shown for all observations (n=46) and separated between observations with larger spatial footprint (n=20) including Cosmic Ray Neutron Sensing (CRNS) and spatially distributed measurements (SDM) as well as point measurements (n=26) including single profile measurements (SPM) and lysimeters (LYSI) .

Next, we contrast the drought characteristics based on the two model setups to assess the differences in drought ranking and the spatial structure of drought events. Annual drought intensities aggregated over Germany based on the daily SMI using simulated SM from 1952–2020 are presented in Fig. 8 and grid-based for the last decade in Fig. 9. Fig. 8 shows only marginal differences between the model setups, which are slightly more prominent in the top soil compared to the total soil column. 410 The model setups largely agree on the three years with the most intensive droughts. The ranking in the top soil during the vegetative active period differs slightly due to the similar drought intensities in the years 1959, 1976 and 2003. The drought years are more pronounced with respect to drought intensities in the GDM-v2-2021 setup in the top soil, but in contrast, the average drought area is estimated larger in the GDM-v1-2016 setup in those years. Generally, the classification of drought years aggregated over Germany results in similar estimates using the different operational drought monitor setups.

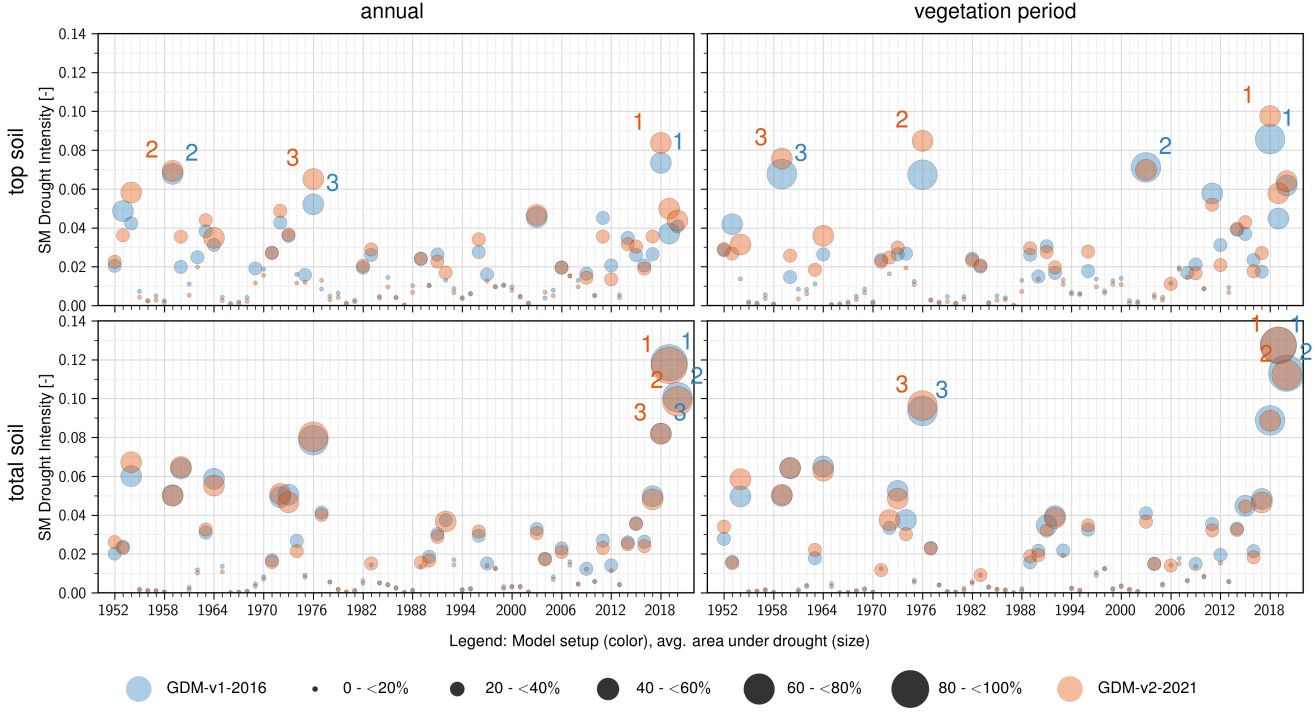

**Figure 8.** SM drought intensities spatially aggregated over Germany during the vegetative active period (April - October) in the top soil (5–25 cm) and total soil column (up to 2 m). The size of the circles represents the average area under drought. The three largest drought events are numbered in each panel. Colours represent the two model setups.

To assess regional differences in drought characteristics between the model setups, Figure 9 shows the drought intensity maps in the vegetative active period for 2011–2020. Drought intensities are more spatially diverse in the GDM-v2-2021 setup stemming from the higher granularity of the GDM-v2-2021 setup including higher-resolution soil information and less smooth patterns than the GDM-v1-2016. Nevertheless, the general patterns are similar between the two setups. Regionally large differences can be seen (e.g., the drought intensities in the Swabian and Franconian Jura region are more pronounced concerning

the neighbouring areas in the GDM-v1-2016 setup – see years 2017 and 2019 for the total soil). Additionally, the differences in drought intensities are more pronounced in the total soil column in the last decade, which can be explained by multi-annual, cumulative effects. The current total soil drought lasts in many regions for at least three years. In Figure 10 the variance between grid cells for drought intensities during the vegetative active period are shown as semi-variograms. In general, the spatial variance is larger in the total soil than top soil. The GDM-v2-2021 setup shows a general larger spatial variance between grid

cells in the top soil and larger increase with distance (see Figure 10 (a)). The spatial variance in the total soil is lower at smaller distances in the GDM-v1-2016 setup, but slightly higher at larger distances. Figure 10 (b) showing semi-variance normalized by distance demonstrates that in the GDM-v2-2021 setup the distance-normalized variance of drought intensities is increased

especially at small spatial scale in both the top and total soil, indicating larger local differences in response to drought intensities. These findings are in line with Livneh et al. (2015) who investigated the influence of different soil databases on resulting

hydrologic fluxes. They reported that the higher variability of soil properties in the finer soil database generally resulted in simulations with more variability in (extreme) hydrologic responses.

We would like to highlight that in our study several changes between the operational model setups were implemented besides changing the underlying soil dataset as described in section 2.2. The changes such as the land use and geology datasets influence the hydrological simulations, yet they play a minor role for the SM simulations compared to the change in the soil

dataset. The SM simulations are not influenced by the geological dataset, because no direct feedback from the saturated aquifer to the SM reservoir is implemented in mHM. To demonstrate the different role of the change in SM dynamics related to the specific soil and land use datasets temporal correlations between SM from separated model runs fixing all model settings (L1 $\approx 1.2 \times 1.2 \, \text{km}^2$ resolution, default mHM parameters) only changing the soil dataset (BUEK200 – BUEK1000) and in a separate step only changing the land use dataset (CORINE – GLOBCOVER) are shown in Figure A4. The change of the soil

dataset has a much larger impact on the SM simulations compared to the change of the land use dataset in these specific model setups. The CORINE and GLOBCOVER land use datasets both have already high horizontal resolutions ($\approx$100 m and 300 m, respectively). The differences between the land use datasets mostly lie in the subgrid scale of the mHM hydrological modelling resolution and have a minor effect on the upscaled hydrological response at the L1 level (here $\approx 1.2 \times 1.2 \, \text{km}^2$).

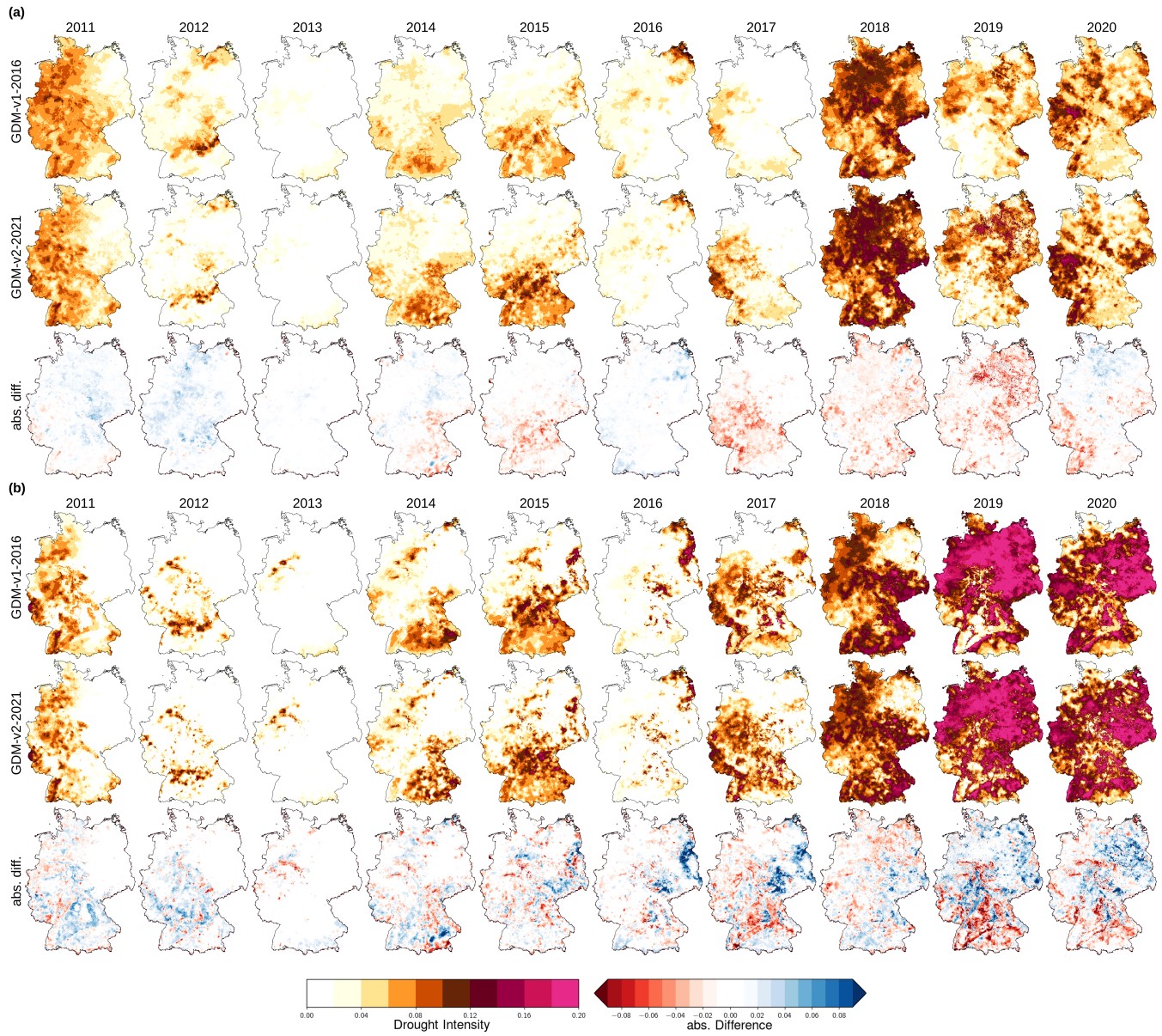

**Figure 9.** SM drought intensities (DI) per grid cell for (a) upper soil (5–25 cm) and (b) total soil column (up to 2 m) during the vegetative active period (April -October) in the last decade (2011–2020) for the model setups GDM-v1-2016 and GDM-v2-2021 and for the absolute differences between the setups (GDM-v1-2016 - GDM-v2-2021). The GDM-v1-2016 data was remapped to the GDM-v2-2021 grid for the difference calculation. Graphs including of years from 1952 on can be found at https://www.ufz.de/index.php?de=47252.

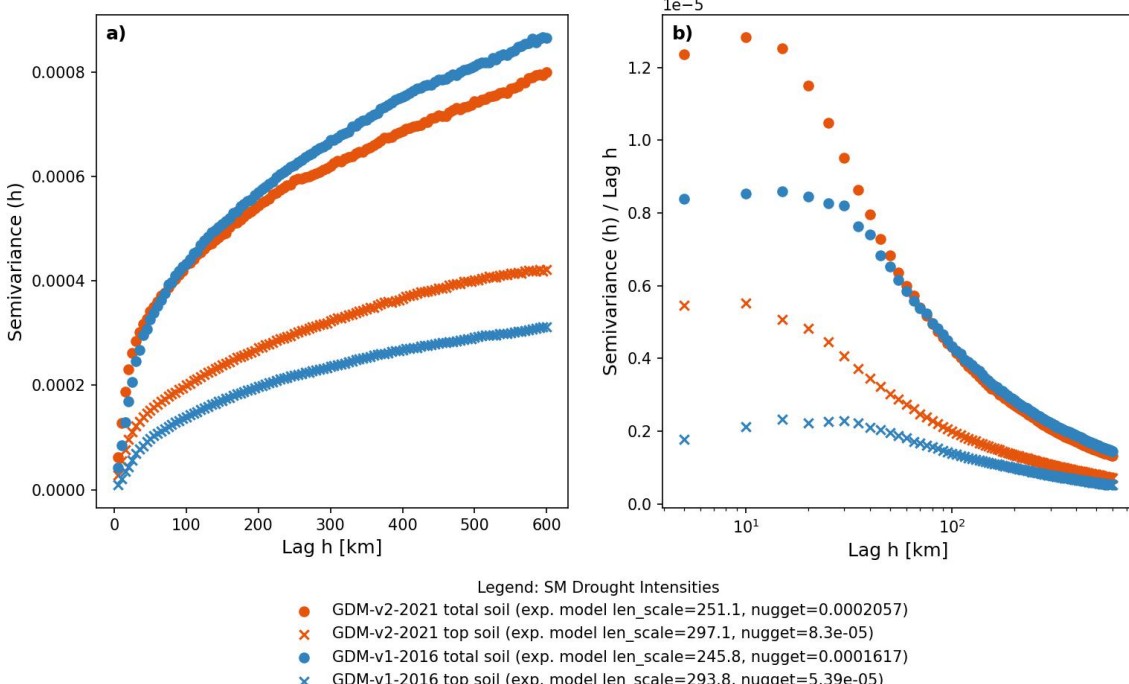

**Figure 10.** Empirical semi-variograms for drought intensities during the vegetative active period in upper soil for the GDM-v1-2016 and the GDM-v2-2021 setup. The bin size was set to 5 km that corresponds to the nearest larger even kilometre bin size relative to the GDM-v2-2016 modelling resolution. The length scale and nugget of the fitted exponential theoretical semi-variograms are noted in the legend. Subplot (b) shows the semivariance normalized by distance and the x axis is log scaled.

**Table 3.** Overview of SM measurement sites. Method denotes the different data sources: Cosmic Ray Neutron Sensing (CRNS), spatially distributed measurements (SDM), single profile measurements (SPM) and lysimeter (LYSI). Network denotes the environmental observation network name (for TERENO: GC =Central Germany; Rur/E = Rur/Eifel; NE = Northeast, PAO = Pre Alpine Observatory) and land use describes the site characteristics (grass= grassland, crop = cropland, DBF = Deciduous Broadleaved Forest, ENF = Evergreen Needled Forest, clear=clearing). For FLUXNET, the original site name is included in parentheses. Sensor depths and numbers are denoted. R Spearman correlation coefficients of simulated versus observed deseasonalized SM anomalies in the GDM-v2-2021 setup shown are based on the whole period at 0–25 cm and 25–60 cm depth.

| network | site | method | land use | begin | end | availability | data $n$ | elevation | precipitation | sensor $n$ | sensor depth | | R | R |
|---|---|---|---|---|---|---|---|---|---|---|---|---|---|---|
| | | | | | | | | | | | 0–25 cm | 25–60 cm | 0–25 cm | 25–60 cm |
| TERENO CG | BadLauchstädt | LYSI | crop | 2016-01-01 | 2018-12-31 | 100 % | 1091 | 118 m | 498 mm | 3 | 10 | 30, 50 | 0.73 | 0.71 |
| | Ermsleben | SPM | grass | 2012-01-25 | 2019-12-31 | 81 % | 2343 | 167 m | 541 mm | 1 | 10, 20 | 30, 40, 50, 60 | 0.80 | 0.68 |
| | Am Grossen Bruch | CRNS | grass | 2014-06-24 | 2019-11-28 | 95 % | 1892 | 81 m | 545 mm | | | | 0.79 | - |
| | | SDM | | 2014-07-30 | 2019-11-18 | 84 % | 1618 | | | 20 | var. | var. | 0.82 | 0.84 |
| | | SPM | | 2014-02-07 | 2019-12-31 | 98 % | 2114 | | | 1 | 10, 20 | 30, 40, 50 | 0.86 | 0.85 |
| | Hecklingen | SPM | grass | 2013-07-05 | 2019-12-31 | 94 % | 2223 | 93 m | 525 mm | 1 | 10, 20 | 30, 40, 50 | 0.72 | 0.71 |
| | HohesHolz | CRNS | DBF | 2014-08-27 | 2019-11-28 | 91 % | 1745 | 203 m | 645 mm | | | | 0.80 | - |
| | | SDM | | 2012-07-20 | 2019-12-30 | 96 % | 2613 | | | 39 | var. | var. | 0.88 | 0.86 |
| | | SPM | | 2013-04-25 | 2019-12-31 | 97 % | 2358 | | | 2 | 10, 20 | 30, 40, 50 | 0.87 | 0.88 |
| | Hordorf | CRNS | crop | 2016-09-29 | 2019-11-28 | 88 % | 1022 | 80 m | 554 mm | | | | 0.83 | - |
| | | SPM | | 2015-11-06 | 2019-12-31 | 98 % | 1493 | | | 1 | 10, 20 | 30, 40, 50 | 0.82 | 0.74 |
| TERENO Rur/E | Aachen | CRNS | crop | 2012-01-13 | 2019-05-01 | 91 % | 2437 | 216 m | 875 mm | | | | 0.67 | - |
| | Gevenich | CRNS | crop | 2011-07-06 | 2019-01-04 | 91 % | 2496 | 104 m | 766 mm | | | | 0.84 | - |
| | Heinsberg | CRNS | grass | 2011-09-08 | 2019-05-01 | 94 % | 2628 | 61 m | 712 mm | | | | 0.83 | - |
| | Kall | CRNS | grass | 2011-09-14 | 2019-05-01 | 78 % | 2185 | 492 m | 861 mm | | | | 0.82 | - |
| | Kleinau | CRNS | grass | 2015-08-25 | 2019-04-26 | 87 % | 1169 | 355 m | 937 mm | | | | 0.88 | - |
| | Merzenhausen | CRNS | crop | 2011-05-18 | 2019-04-03 | 90 % | 2597 | 91 m | 767 mm | | | | 0.85 | - |
| | Rollesbr1 | CRNS | crop | 2011-05-18 | 2018-12-31 | 87 % | 2409 | 516 m | 1183 mm | | | | 0.77 | - |
| | Rollesbr2 | CRNS | grass | 2012-06-30 | 2018-12-25 | 86 % | 2045 | 516 m | 1183 mm | | | | 0.82 | - |
| | Ruraue | CRNS | grass | 2011-11-08 | 2019-01-01 | 90 % | 2340 | 102 m | 734 mm | | | | 0.77 | - |
| | Schoeneseiffen | CRNS | grass | 2015-08-13 | 2019-04-25 | 83 % | 1120 | 567 m | 1119 mm | | | | 0.82 | - |
| | Selhausen | CRNS | crop | 2015-03-06 | 2019-04-26 | 95 % | 1441 | 102 m | 726 mm | | | | 0.77 | - |
| | Wildenrath | CRNS | clear | 2012-05-11 | 2019-03-23 | 91 % | 2273 | 79 m | 776 mm | | | | 0.80 | - |
| | Wüstebach | CRNS | ENF | 2011-03-12 | 2018-10-05 | 79 % | 2173 | 614 m | 1165 mm | | | | 0.44 | - |
| | | SDM | | 2009-01-27 | 2019-12-31 | 100 % | 3989 | | | 150 | 10, 20 (2x) | 50 | 0.75 | 0.74 |
| TERENO NE | AltTellin | SPM | grass | 2014-05-10 | 2019-12-30 | 90 % | 1859 | 9 m | 551 mm | 1 | 10, 20 | 30, 40, 50 | 0.87 | 0.26 |
| | Bentzin | SPM | grass | 2013-08-19 | 2019-12-30 | 98 % | 2281 | 5 m | 568 mm | 1 | 10, 20 | 30, 40, 50, 60 | 0.53 | 0.77 |
| | Droennewitz | SPM | grass | 2014-04-12 | 2019-12-30 | 73 % | 1519 | 33 m | 598 mm | 1 | 10, 20 | 30, 40, 50, 60 | 0.59 | 0.66 |
| | Goermin | SPM | grass | 2013-08-19 | 2019-12-30 | 95 % | 2207 | 7 m | 569 mm | 1 | 10, 20 | 30, 40, 50, 60 | 0.73 | 0.47 |
| | Leppin | SPM | grass | 2013-01-28 | 2019-12-30 | 96 % | 2420 | 6 m | 563 mm | 1 | 10, 20 | 30, 40, 50, 60 | 0.71 | 0.53 |
| | Medrow | SPM | grass | 2015-07-13 | 2019-12-30 | 100 % | 1627 | 5 m | 595 mm | 1 | 10, 20 | 30, 40, 50, 60 | 0.78 | 0.76 |
| | Muehlenkamp | SPM | grass | 2012-01-01 | 2019-12-30 | 82 % | 2388 | 8 m | 574 mm | 1 | 10, 20 | 30, 40, 50, 60 | 0.71 | 0.37 |
| | Ueckeritz | SPM | grass | 2013-01-28 | 2019-12-30 | 93 % | 2349 | 4 m | 562 mm | 1 | 10, 20 | 30, 40, 50, 60 | 0.63 | 0.61 |
| | Wotenick | SPM | grass | 2014-04-30 | 2019-12-30 | 95 % | 1964 | 11 m | 588 mm | 1 | 10, 20 | 30, 40, 50, 60 | 0.63 | 0.51 |
| | Zarnekla | SPM | grass | 2013-01-23 | 2019-12-30 | 95 % | 2404 | 6 m | 590 mm | 1 | 10, 20 | 30, 40, 50, 60 | 0.83 | 0.77 |
| TERENO PAO | Fendt | LYSI | grass | 2017-01-01 | 2019-12-31 | 100 % | 1090 | 634 m | 1059 mm | 18 | 10 | 30, 50 | 0.80 | 0.7 |
| | Graswang | LYSI | grass | 2017-03-17 | 2019-12-31 | 93 % | 948 | 916 m | 1570 mm | 6 | 10 | 30, 50 | 0.77 | 0.66 |
| | Rottenbuch | LYSI | grass | 2017-03-17 | 2019-12-31 | 93 % | 948 | 765 m | 1265 mm | 12 | 10 | 30, 50 | 0.54 | 0.53 |
| FLUXNET | Gebesee (DE-Geb) | SPM | crop | 2001-01-16 | 2014-12-31 | 91 % | 4657 | 156 m | 522 mm | 1 | 8, 16 | 32 | 0.38 | 0.33 |
| | Grillenburg (DE-Gri) | SPM | grass | 2006-11-21 | 2014-12-31 | 98 % | 2891 | 394 m | 856 mm | 1 | 10 | - | 0.68 | - |
| | Hainich (DE-Hai) | SPM | DBF | 2002-12-27 | 2012-12-31 | 98 % | 3570 | 420 m | 774 mm | 1 | 8, 16 | 32 | 0.72 | 0.57 |
| | Klingenberg (DE-Kli) | SPM | crop | 2004-11-27 | 2014-12-31 | 87 % | 3208 | 478 m | 860 mm | 1 | 10 | - | 0.57 | - |
| | Lackenberg (DE-Lkb) | SPM | ENF | 2009-05-01 | 2013-12-31 | 90 % | 1533 | 1252 m | 1573 mm | 1 | 4 | - | 0.44 | - |
| | Leinefelde (DE-Lnf) | SPM | DBF | 2002-05-01 | 2012-12-31 | 72 % | 2796 | 453 m | 784 mm | 1 | 8, 16 | 32 | 0.83 | 0.77 |
| | Tharandt (DE-Tha) | SPM | ENF | 1997-03-06 | 2014-12-31 | 95 % | 6189 | 369 m | 791 mm | 1 | 10 | - | 0.57 | - |
| DWD | Cunnersdorf | CRNS | crop | 2016-06-23 | 2019-12-31 | 95 % | 1217 | 131 m | 634 mm | | | | 0.83 | 0.68 |

## 4 Summary and conclusions

This study evaluates soil moisture (SM) dynamics from two mHM simulations used as operational model setups in the German Drought Monitor (GDM). The increase of hydrological modelling resolution between the model setups from $4{\times}4\,\mathrm{km}^2$ in the GDM-v1-2016 setup to $\approx 1.2{\times}1.2\,\mathrm{km}^2$ in the GDM-v2-2021 setup was motivated by the implementation of a higher-resolution input soil data (BUEK1000 to BUEK200). The comparisons between observed and simulated SM were conducted using various ground-based SM observations with multiple measurement methods and different climate gradients. The agreement between

simulated and observed SM dynamics is especially high in the vegetative active period (median R 0.84 in GDM-v2-2021) and lower in winter (median R 0.59 in GDM-v2-2021). It was shown that the $\approx$1.2km resolution GDM not only produces simulated SM in similar quality as the lower resolution model setup, but partly enhances the model ability to simulate observed SM dynamics. We identified significant improvements between the first and second GDM versions in terms of agreement to observed SM, with enhanced correlations during fall (+0.07 median) and winter (+0.12 median). However, the overall

improvements were relatively small, partly because the lower resolution model setup (4x4km grid cells) was already capturing the observed SM dynamics well. Both model setups display similar correlations to observations in the dry anomaly spectrum, with higher overall agreement of simulations to observations with a larger spatial footprint. Although several changes were made between the operational model setups as changing landuse and hydrogeology datasets in addition to the change in the underlying soil dataset (see section 2.2), it was demonstrated that the soil dataset played the dominant role in the changes in

simulated SM dynamics. Annual drought statistics and ranking based on drought intensities and average area under drought computed on the time frame 1952–2020 were robust between the model setups, with only minor differences on the scale of Germany. The spatial structures in the higher-resolution GDM-v2-2021 setup, including an updated soil map, display larger granularity and spatially more diverse responses to drought, allowing a more refined representation of spatial SM heterogeneity. The higher spatial resolution achieved is of great relevance, especially concerning local risk assessments.

The results underline the importance of long-term measurement series for developing and optimising data products such as the GDM. Good coverage of relevant environmental gradients with suitable measurement networks is essential due to rapidly changing environmental conditions. The direct comparison of the different measurement methods for recording SM showed the importance of measurement methods such as CRNS or SDM, which allow better estimates of mean SM conditions across larger areas. However, the temporal and spatial availability still limits the studies, such as the one presented here, in terms of

statistical robustness. Furthermore, we did not analyse deeper soil depths ($> 60\,\mathrm{cm}$), as most measurement sites do not have SM data at those depths. Continuous improvements of the SM observational database will be beneficial for future hydrological model evaluations. For future studies, a solution to the variable penetration depth of CRNS could be to compare observed and simulated neutron counts directly by using the COSMIC forward model (Shuttleworth et al., 2013), which is designed to account for irregular SM profiles in all modelled depth layers. While COSMIC has been already implemented in mHM,

its proper parameterization would require dedicated research and is outside of the scope of this study. Regarding the SDM measurements, a source of uncertainty remains in the calculation the spatial average. The mean calculation is challenging due

to the varying number of available sensors in the measurement grids over time. A robust mean calculation with advanced sensor weighting is currently a subject of active research.

We compared the model simulations in terms of SM dynamics for their relevance for SM droughts, which are defined as
a negative deviation from normal SM conditions. The integration of observed SM data in the model calibration itself could improve the absolute estimations of simulated SM and the model internal flux partitioning. This approach has been successfully demonstrated using CRNS data in the Rur catchment in Germany (Baatz et al., 2017) and remotely sensed SM in the Danube catchment (Wanders et al., 2014). An extended model validation of the SM component of mHM forced with onsite precipitation and local soil maps with soil physical property information on even higher resolution (e.g. BUEK25 or BUEK50) would help
to further understand current limitations of mHM in modelling SM dynamics and separate the analyses from the limited data availability at the scale of Germany.

Several other aspects are relevant to further improve SM states' simulations with mHM on a national scale in Germany (or larger towards a continental scale). A decisive input that influences hydrological model performance is precipitation (Mo et al., 2012). Model performance of mHM was related to rain gauge density on a European scale by Rakovec et al. (2016).
While Germany has a very dense meteorological station network, local precipitation can still differ significantly from the interpolated products. Although, Samaniego et al. (2013) showed that the interpolation results on daily precipitation data here compared to the high resolution German Weather Service reanalysis product REGNIE (Rauthe et al., 2013) only differ marginally, the difference of local precipitation from interpolated values is expected to have a large influence on SM dynamics. Thus, improvements in the interpolated precipitation may result in increased model performance. Additionally, a more precise
estimation of potential evapotranspiration may be achieved by implementing the Penman-Monteith methods.

Finally, we conclude that the resolution of $\approx 1.2 \times 1.2 \, \text{km}^2$ is currently the best compromise between the need for increased model resolution (user perspective) and the current data availability and process representation in mHM (scientific perspective). We emphasise the need for continuous dialogues between stakeholders and the scientific community to improve the underlying model system alongside the provision of user-tailored drought information.

 **Appendix A: Appendix**

**A1**

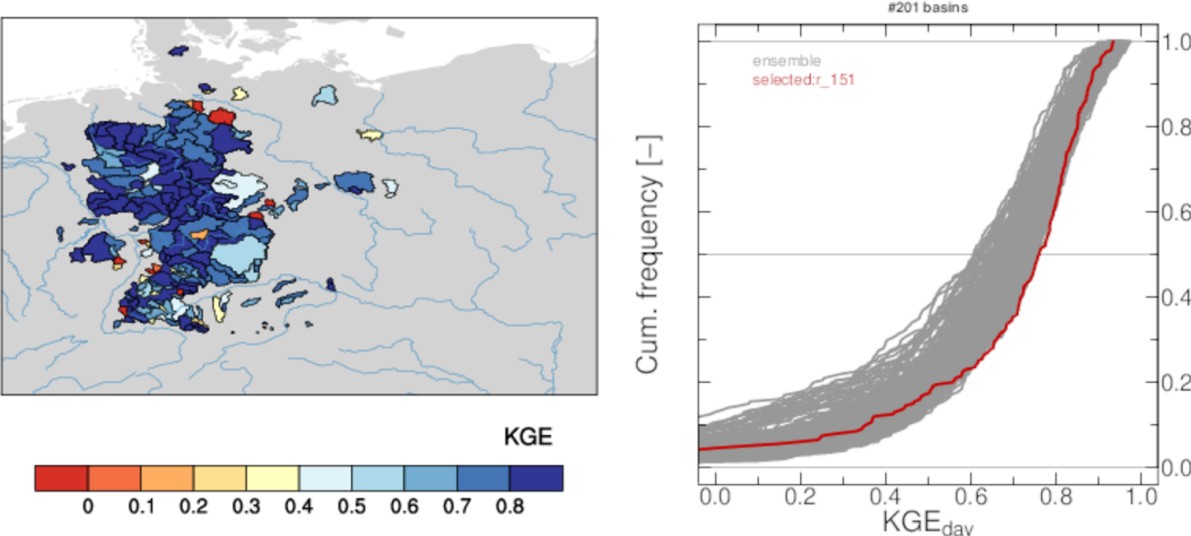

**Figure A1.** Results of mHM multbasin model calibration based on streamflow data from 201 catchments. Left: spatial map of KGE for each basin. Right: KGE Cumulative density function of setup 200 parameter sets, generated by random sampling of the basins. Bold red marks the selected parameter set.

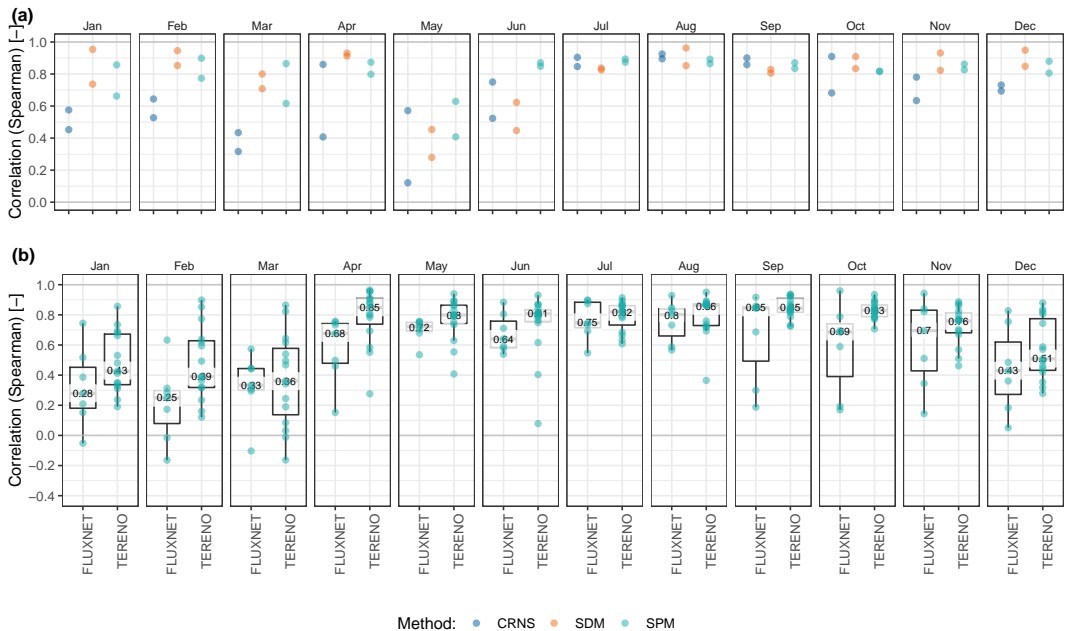

**Figure A2.** Spearman correlation coefficients of simulated soil moisture by mHM in the GDM-v2-2021 setup versus observed deseasonalized soil moisture anomalies for each month as a supplement to Fig. 5 by (a) comparing the locations Hohes Holz and Am Grossen Bruch equipped with CRNS, SDM and SPM soil moisture measurements and (b) comparing FLUXNET (n=7) and TERENO (n=20) SPM data.

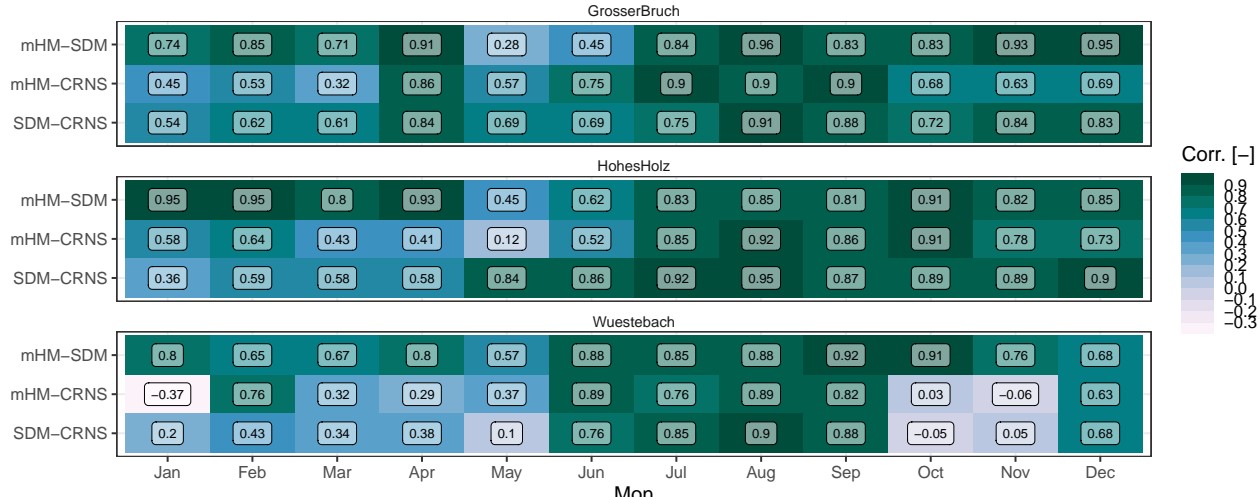

**Figure A3.** Monthly Spearman rank correlation coefficients for 2014 - 2019 between deseasonalized SM anomalies simulated by mHM and SM observations from CRNS and SDM measurements and between the CRNS and SDM observations for the three locations Am Grossen Bruch, Hohes Holz and Wüstebach.

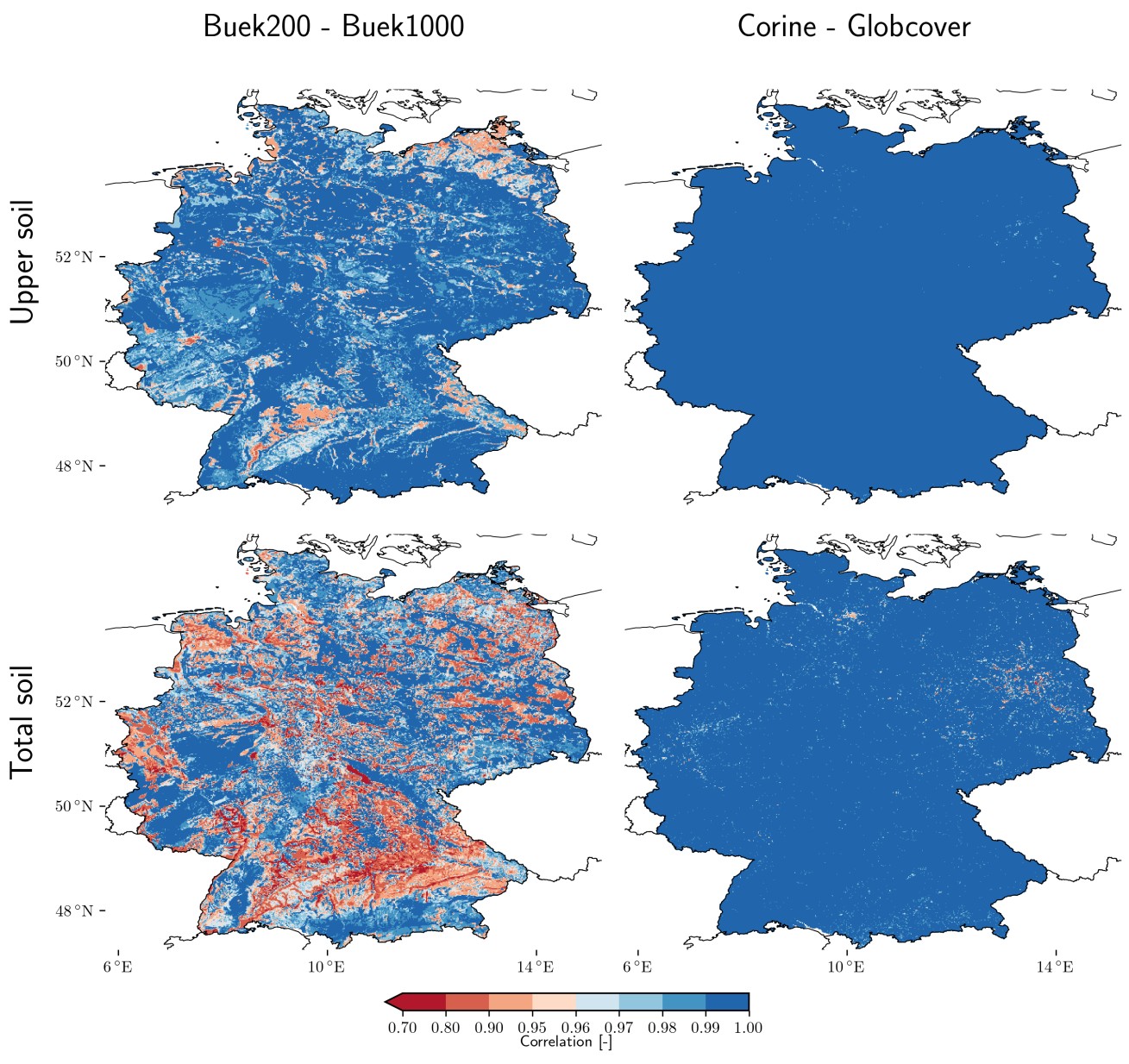

**Figure A4.** Correlations between simulated daily SM in the upper soil (5–25 cm) and the total soil column (up to 2 m) using model runs for the time period 1991–2019 keeping the model settings identical (L1 1.2km$^2$ resolution, default mHM parameters) except changing soil dataset BUEK200 versus BUEK1000 (left) and secondly changing landcover dataset CORINE versus GLOBCOVER (right).

*Code and data availability.* Observational and simulated SM data (https://www.doi.org/10.48758/ufz.12541) as well as simulated SMI drought characteristics (DOI: https://www.doi.org/10.48758/ufz.12534) that were used in the study are available in UFZ Data Investigation Portal. Open-source mHM code is available at https://github.com/mhm-ufz and SMI code https://doi.org/10.5281/zenodo.5842486. TERENO and

FLUXNET soil moisture data can be obtained at https://ddp.tereno.net/ddp/ and https://fluxnet.org/, respectively.

*Author contributions.* F.B., L.S., A.H., C.R., M.S., A.M., R.Kumar developed the concept for the manuscript; F.B. conducted simulations, SM data compilation, data analyses, first manuscript drafts; O.R.,F.B. L.S, R.Kumar, A.M. contributed to develop the GDM-v2-2021 setup; O.R. calibrated the GDM-v2-2021 setup; R.Kumar., M.S.,O.R.,A.M., A.H., L.S.,S.T. helped to improve the analyses and manuscript; S.M. and S.T. supported mHM model development and technical maintenance of the GDM; M.S., C.R., R.Kiese, K.S., H.B. provided SM ob-

servation data and helped with interpreting the data/improve the manuscript; S.Z. assisted answering questions related to soil processes and improve the manuscript.

*Competing interests.* The authors declare that they have no conflict of interest.

*Acknowledgements.* This work is partly funded by Helmholtz-Climate-Initiative (HI-CAM) in the Helmholtz Associations Initiative and Networking Fund. The authors are responsible for the content of this publication. We acknowledge the TERENO community and FLUXNET

community for providing soil moisture observational data. Special acknowledgment goes to Hans-Jörg Vogel (UFZ), René Zahl (UFZ), Christian Hohmann (GFZ), Ingo Heinrich (GFZ) and Falk Böttcher (DWD) for providing soil moisture observations. We kindly acknowledge the German Weather Service (DWD), the European Environmental Agency (EEA), the Federal Institute for Geosciences and Natural Resources (BGR), the Federal Agency for Cartography and Geodesy (BKG), the European Space Agency (ESA), the U.S. Geological Survey (USGS), the Global Runoff Data Centre (GRDC) as data providers. The scientific results have (in part) been computed at the High-Performance

Computing (HPC) Cluster EVE, a joint effort of both the Helmholtz Centre for Environmental Research - UFZ (http://www.ufz.de/) and the German Centre for Integrative Biodiversity Research (iDiv) Halle-Jena-Leipzig (http://www.idiv-biodiversity.de/). We would like to thank the administration and support staff of EVE who keep the system running and support us with our scientific computing needs: Thomas Schnicke, Ben Langenberg, Guido Schramm, Toni Harzendorf, Tom Strempel and Lisa Schurack from the UFZ, and Christian Krause from iDiv. We thank Lily-belle Sweet for her help to improve the language of the manuscript. Finally, we want to thank two anonymous reviewers,

the reviewer René Orth and the Editor for their constructive comments, which improved the quality of this study.

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
