# Peer review of "High-resolution drought simulations and comparison to soil moisture observations in Germany"

_Hydrology and Earth System Sciences, 2021_

## Referee Comment (RC1)

**Review of High-resolution drought simulations and comparison to soil moisture observations in Germany by Boeing et al.**

**Manuscript Number hess-2021-402**

A recommendation: moderate revision

Using observed SM data collected from 40 locations and in 4 different measurement methods in Germany, the authors evaluate the performance of second generation operational German Drought Monitor in simulating soil moisture (SM). Two major research questions within this paper have been adequately addressed and can be summarized as follows: 1. how well the GDM capture the SM dynamics; 2. will GDM with higher spatial resolution produce SM estimates with higher quality compared with the GDM of former edition. Through the research, it was found that 1. SM dynamics simulations could be moderately improved; 2. higher resolution drought information at the one-kilometer scale can be met.

This research is a report of the improvement in the model performance which is evaluated in the perspective of comparisons between the model simulations and the observations. The article conforms to the journal-specific instructions and is relevant to HESS. The work is appropriate to be published in this journal after some revisions. In the following part, I will state the major arguments in detail.

The comparison between the observations and the model simulations are mainly indicated in the form of Spearman rank correlation coefficient. Nevertheless, the results of the comparison are in fact not so ideal in the perspective of solely the Spearman correlation coefficient, not to mention that significance values have been neglected when the observations and model results are compared. For example, when the observations from all the sites are included, the coefficients are normally lower. In addition, p-value has only been mentioned when both versions of the mHM and observations are compared (table 2). No mentioning of significance values also makes the comparison results not so validated. The significance value should be mentioned in the research to make results more reliable.

In addition, the analyses are focused on the explanations for the correlation coefficients, few further knowledge regarding the inner uncertainty within the models such as misrepresentations or no involvement of certain natural factors are included to be accounted for the discrepancy between the observations and model results. In this way, the detailed explanations in the conclusion and discussion part of this paper are needed to clarify the differences between observations and model results.

The four kinds of observations seem not to be standardized. In this way, some values of correlation coefficient which concern a specific observation technique, are not guaranteed to be valid in indicating the performance of GDM in simulating SM dynamics. A standardization method should be carried out to ensure a consistent comparison between observations and model simulations.

In order to facilitate a better standardization, not only the differences in the methods of the

observations, the observation sites' conditions should be considered in explaining the research results, such as the landuse, elevation, precipitation of different sites. Additionally, the change of the landuse dataset and geology dataset in the two versions of the models has not been verified to have a positive effect on the simulation of soil moisture as the improvement in the second version of the model is not so evident (especially in spring and summer the change of Spearman rank correlation coefficient is negative). Further clarification for the effects of the model setting is needed in this research or in the future studies to be conducted by the authors' team.

Moreover, the limited length of observed soil moisture data (< 10 years for most locations) causes some uncertainty in the comparison between observations and model results in the whole. More observations are needed to facilitate a more reliable model evaluation using the observation datasets as the existing observations within this paper are only validated in representing some regions of the Germany.

The authors can try making better simulations of the water cycle including soil moisture, ground water, and precipitation, while a higher-quality observational soil moisture dataset is applied in the future study. Other methods to indicate the correlations between the observations and model results can be used and also other indices apart from SMI can be utilized in indicating the severity of soil moisture drought.

Last but not least, it would be better to have some discussions on the applicability of the model to other regions other than Germany. Are there some future plans on extending the regional applicability of the model? Besides, some discussions on what can we learn from a small-scale modelling to improve large-scale modelling can be stated.

Some specific revisions regarding some parts of the manuscript are listed as follows:

1. The abstract is complete and correctly summarize the content of the paper, but it may need to be reduced a little to be more concise.
2. The process of parameter calibration and optimization needs to be in more detail to facilitate a reproducibility in the future study.
3. Some of the conclusions are overstated. The explanation should be in more detail regarding these issues.

   (1) Figure 3 shows the time series of both the observations and model simulations. It seems that the coefficient is much higher than those when all the sites were selected. How the sites are selected may need to be mentioned if there are other sites that contain both Cosmic Ray Neutron Sensing (CRNS) and Spatially Distributed Measurements (SDM).

   (2) There is lower agreement between observations and simulations in winter.

   (3) There is improvement of second version of the model in representing the upper soil but stagnation in representing the whole soil.

(4) The values of the Spearman rank correlation coefficient are not high enough to conclude a definite improvement of the first version of the mHM (Table 2).

4.  In general, the authors have given proper credit to related work and clearly indicate their own original contribution. The references are appropriate to the research, but it would be better if some more papers are referenced especially those in which multibasin model calibrations and the SMI were applied.

The suggestions regarding some minor flaws and typos are described as follows:

Page 5, line 125: delete the ".However".

Page 5, line 131: delete the "," between "1.23" and "km".

Page 6, line 138: move "that were used in the analysis" before "are located".

Page 9, line 212, 213: remove "as"

Page 9, line 212, 213: add "," before "including", "the estimating", and "is hampered".

Page 11, line 270: remove "," after the "both".

Figure 3: add (a) to (l) for each sub panel to facilitate a better reference to the figures in the text when making the explanations.

Page 22: change the subtitle to "Conclusions and Discussions".

Page 23, line 436: remove "that".

Page 23, line 438: change "constitute" to "conclude".

The research is sound and fundamental. Some language edits could be good for improving the paper's quality.

---

## Author Comment (AC1)

*Original reviewer comments are in italics,* **authors' response is in bold**.

**Referee #1:**

*SUMMARY:*
*A recommendation: moderate revision*
*Using observed SM data collected from 40 locations and in 4 different measurement methods in Germany, the authors evaluate the performance of second generation operational German Drought Monitor in simulating soil moisture (SM). Two major research questions within this paper have been adequately addressed and can be summarized as follows: 1. how well the GDM capture the SM dynamics; 2. will GDM with higher spatial resolution produce SM estimates with higher quality compared with the GDM of former edition. Through the research, it was found that 1. SM dynamics simulations could be moderately improved; 2. higher resolution drought information at the one-kilometer scale can be met.*
*This research is a report of the improvement in the model performance which is evaluated in the perspective of comparisons between the model simulations and the observations. The article conforms to the journal-specific instructions and is relevant to HESS. The work is appropriate to be published in this journal after some revisions. In the following part, I will state the major arguments in detail..*

**Authors' response #1: We thank the Reviewer for an overall positive assessments of our work. We paid detailed attention to all comments and have addressed all of them below accordingly.**

*The comparison between the observations and the model simulations are mainly indicated in the form of Spearman rank correlation coefficient. Nevertheless, the results of the comparison are in fact not so ideal in the perspective of solely the Spearman correlation coefficient, not to mention that significance values have been neglected when the observations and model results are compared. For example, when the observations from all the sites are included, the coefficients are normally lower. In addition, p-value has only been mentioned when both versions of the mHM and observations are compared (table 2). No mentioning of significance values also makes the comparison results not so validated. The significance value should be mentioned in the research to make results more reliable*

**Authors' response #2: Thank you for this comment. We agree with the reviewer that the significance value is the essential information. Therefore we will add the significance values into Figure 4 (see exemplary Figure R1 below) and Figure 5 by indicating locations with p-values $< 0.05$ and positive correlations.**

[Figure]

Figure R1: Same as Figure 4 in the main manuscript, but including information of correlation significance (p-value < 0.05) and positive correlation.

*In addition, the analyses are focused on the explanations for the correlation coefficients, few further knowledge regarding the inner uncertainty within the models such as misrepresentations or no involvement of certain natural factors are included to be accounted for the discrepancy between the observations and model results. In this way, the detailed explanations in the conclusion and discussion part of this paper are needed to clarify the differences between observations and model results.*

*The four kinds of observations seem not to be standardized. In this way, some values of correlation coefficient which concern a specific observation technique, are not guaranteed to be valid in indicating the performance of GDM in simulating SM dynamics. A standardization method should be carried out to ensure a consistent comparison between observations and model simulations.*

**Authors' response #3: We thank the reviewer for the suggestions and agree that aspects regarding soil moisture measurements need to be considered carefully. Since there are differences between available soil moisture measurement techniques and to the spatial simulation scale, the approach of this paper was to include various measurement techniques in the model evaluation. Therefore, we believe that it is one of the strengths of the manuscript to show the comparison of mHM simulations to not only one type, but different soil moisture measurement techniques. As a consequence, differences that result from the measurement techniques are considered and discussed thoroughly in the manuscript.**

**The differences can be related e.g. to the site conditions, type of measurement devices and spatial or vertical scales that are represented. We were not completely sure to which of these differences the reviewer refers. Therefore, we aim to point out in the following how we addressed them in the main manuscript and how to improve it.**

**The SM sites are not homogeneously distributed over site conditions (e.g.**

lysimeters are placed in grasslands). In general, it could be shown that the model performances doe not systematically depend on site conditions that could result from different measurement techniques (see also answer #4). We suggest to move the Figure A2 to the section 3.1 to give it a more central focus.

Regarding the measurement devices, the single profile measurements (SPM), spatially distributed measurements (SDM) and lysimeters (LYSI) use either Time or Frequence Domain Reflectrometry (TDR, FDR) sensors that do not expect differences in measurement quality. Cosmic Ray Neutron Sensing (CRNS) use a different measurements technique but were validated by spatially distributed measurements using TDR and FDR sensors (lines 200-204). As well, due to methodological reasons days with snow were discarded from analyses for CRNS as further explained in the method section in lines 200-204 and possible implications in lines 304-307.

The measurements represent different horizontal spatial scales, with the area represented being smallest for SPM and LYSI ($\approx$ point scale), while SDM (resp. the corresponding mean) and CRNS represent much larger areas at 0.1 km$^2$ (see lines 275-276). It was shown that SDM shows slightly better correlations to the simulations that support the closer scale match to the simulations (see e.g. 342-345). Those measurements are however still rare (see line 67) and since the paper focuses on SM dynamics the value of including SPM and LYSI measurements increases substantially. Additionally, CRNS data was excluded from the comparison in depth 25 - 60 cm due to its varying vertical penetration depth that does not allow a consistent depth-wise evaluation (see lines 325-236).

*In order to facilitate a better standardization, not only the differences in the methods of the observations, the observation sites' conditions should be considered in explaining the research results, such as the landuse, elevation, precipitation of different sites.*

Authors' response #4: We acknowledge the reviewers recommendations. In the main manuscript it is mentioned that the German-wide available information (soil maps, land use) is considered (lines 263-265). No available site-specific information is used in setting-up the hydrological model, e.g. precipitation is taken from daily available, interpolated DWD-observations and existing observations in the locations of soil moisture measurements are neglected. Hence, a different model setup would be required to meet the proposed standardization.

Correlations against sites´ conditions (landuse as reported at the site, elevation and average precipitation from model) are plotted in the appendix of the main manuscript Figure A2 (see also Figure R2). As stated in the results section lines 285-286 "there is no general tendency for lower correlations at forest sites compared to crop and grassland sites (see Fig. A2)". Crop sites show slightly lower correlations than grassland sites, which is expected since anthropogenic activities (e.g., crop rotation) are not represented in mHM. Also, correlations show no clear tendency across the range of elevation and precipitation regimes.

[Figure]

Figure R2: Same as Figure A2 in preprint manuscript.

*Additionally, the change of the landuse dataset and geology dataset in the two versions of the models has not been verified to have a positive effect on the simulation of soil moisture as the improvement in the second version of the model is not so evident (especially in spring and summer the change of Spearman rank correlation coefficient is negative). Further clarification for the effects of the model setting is needed in this research or in the future studies to be conducted by the authors' team.*

**Authors' response #5: The increase of model resolution in the second version of the drought monitor was motivated both by the release of a new German-wide soil map [1] and increased user need to higher resolution simulations as extensively described in the main manuscript. This resulted in $\approx 1.2 \times 1.2\,\mathrm{km}^2$ model resolution in the GDM-v2-2021 setup as a compromise between scientific/model perspective (limited by data availability and process representation) and stakeholder/user perspective (see conclusion lines 438-441). Changes in landuse data and geology data (also the change of projection to WGS-84) in the new drought monitor version (GDM-v2-2021) on the other hand were driven by current efforts increasing the applicability and comparability of mHM to regions other than Germany and outside Europe. See also reviewer response #8 regarding the applicability of mHM outside of Germany. We acknowledge the critical remarks related with these changes but will point out in the following that the geology and landcover dataset changes have minor implications for SM drought simulations compared to the change in soil dataset. Currently, mHM takes relatively raw landuse classes. Species specific landcover is currently not accounted for. The difference in the resolution of GLOBCOVER and CORINE landuse dataset are in sub grid scale that influences the subgrid variability (GLOBCOVER resolution: 300 meters, CORINE < 100 m). Differences between the land cover datasets reduce if the land cover data is aggregated to the spatial resolution of the model. For example, at the spatial resolution of 1.2km, over 85 % of the grid cells both datasets agree on the dominant landcover. This shows that differences stem from differences at high spatial resolution and do not have a large impact on the simulation. We will include these aspects in the main manuscript to point out the limitations of the study. We propose to add the following sentence in the main manuscript in line 147: "The changes in landuse and geology dataset can influence the simu-**

lations, yet play a minor role for the soil moisture simulations compared to the change in the soil dataset because changes of landuse data are in subgrid scale (resolution GLOBCOVER 300m, CORINE <100m) and no direct feedback of from saturated "groundwater" storage to soil moisture storage is implemented in mHM."

*Moreover, the limited length of observed soil moisture data (< 10 years for most locations) causes some uncertainty in the comparison between observations and model results in the whole. More observations are needed to facilitate a more reliable model evaluation using the observation datasets as the existing observations within this paper are only validated in representing some regions of the Germany.*

Authors' response #6: We compiled an unprecedented sample of soil moisture observations for hydrological model evaluation in Germany from different state-of-the-art measurement techniques and monitoring networks (FLUXNET and TERENO). We acknowledge the limitations of the length of observational data. In order to investigate the consequences of different time series lengths, Figure R3 shows correlation against length of time series. No trend of deteriorating correlations with length of time series can be detected. We agree on the need to further broaden the observational database for future comparisons of model simulations to observations.

[Figure]

Figure R3: Correlations for the soil moisture observations against simulations (GDM-v2-2021 setup, 0-25cm depth) dependent of number of years of observations.

*The authors can try making better simulations of the water cycle including soil moisture, ground water, and precipitation, while a higher-quality observational soil moisture dataset is applied in the future study. Other methods to indicate the correlations between the observations and model results can be used and also other indices apart from SMI can be utilized in indicating the severity of soil moisture drought.*

**Authors' response #7: Thank you for your comment. We will include the recommendation in future studies.**

*Last but not least, it would be better to have some discussions on the applicability of the model to other regions other than Germany. Are there some future plans on extending the regional applicability of the model? Besides, some discussions on what can we learn from a small-scale modelling to improve large-scale modelling can be stated.*

**Authors' response #8: The applicability of the model to multiple scales and locations is a core element of mHM. The multiscale parameter regionalization (MPR) framework that is unique feature in mHM and now**

available as a standalone tool yields seamless model parameters at multiple scales and locations in an effective manner [13]. The effectiveness of MPR to transfer model parameters to scales and locations other than those used during calibration was demonstrated first in [10, 3]. The mHM was applied and evaluated in different climatological regions e.g. Europe [14, 7], West Africa [2], India [9] and US [4, 8]. [7] extensively evaluated mHM fluxes in Europe against evapotranspiration, soil moisture, runoff and total water storage (GRACE). [9] recently implemented a drought monitoring tool for South Asia based on mHM simulations and SMI. Global mHM simulations are currently conducted and evaluated within ULSYSSES project `https://www.ufz.de/index.php?en=47367`.

*Some specific revisions regarding some parts of the manuscript are listed as follows:*
*1. The abstract is complete and correctly summarize the content of the paper, but it may need to be reduced a little to be more concise.*

**Authors' response #9: We double checked that the abstract length meets the requirements of the journal. We could not identify potential for reducing the length without removing important information.**

*2. The process of parameter calibration and optimization needs to be in more detail to facilitate a reproducibility in the future study.*

**Authors' response #10: To improve reproducibility we suggest to extend the description of the calibration procedure with the following tables and suggest to put them in the supplements of the main manuscript.**

- **Table 1 showing the results of the 201 basins from the final selected parameter set.**

- **Table 2 showing the 200 sets of random multi-basin draws.**

*3. Some of the conclusions are overstated. The explanation should be in more detail regarding these issues.*
*(1) Figure 3 shows the time series of both the observations and model simulations. It seems that the coefficient is much higher than those when all the sites were selected. How the sites are selected may need to be mentioned if there are other sites that contain both Cosmic Ray Neutron Sensing (CRNS) and Spatially Distributed Measurements (SDM).*

**Authors' response #11: The sites have been selected because of the available time series length for different measurement methods. We propose to add relevant information regarding the selection to the manuscript by extending the caption of Figure 3 with the following sentence: "The Hordorf site also contains both CRNS and SDM measurements, but with much shorter time series length. For visualization the stations with longer time series were selected."**

*(2) There is lower agreement between observations and simulations in winter.*

**Authors' response #12: This is shown and discussed against other studies extensively in lines 289-302 of the main manuscript. The reasons lie in**

the variable importance of hydrological processes in different seasons.

*(3) There is improvement of second version of the model in representing the upper soil but stagnation in representing the whole soil.*

**Authors' response #13: We are not sure if we understand this comment correctly. Direct comparison of observed and simulated soil moisture is only possible for the upper soil due to observational data availability. For the total soil column, a comparison of drought intensities was performed showing similar results for both GDM versions. Assessing an improvement in soil moisture simulations between the GDM versions for the total soil column based on SM observations is not possible in the underlying study design.**

*(4) The values of the Spearman rank correlation coefficient are not high enough to conclude a definite improvement of the first version of the mHM (Table 2).*

**Authors' response #14: We clearly support this comment, but did not conclude a definite improvement. See line 397 in manuscript: „However, the overall improvements were relatively small, partly because the lower resolution model setup (4x4km grid cells) was already capturing the observed SM dynamics well."**

*4. In general, the authors have given proper credit to related work and clearly indicate their own original contribution. The references are appropriate to the research, but it would be better if some more papers are referenced especially those in which multibasin model calibrations and the SMI were applied.*

**Authors' response #15: We have extended references with previous works focusing on the multi-basin calibrations, such as [5, 8, 6]. SMI was applied a.o. in [11, 12].**

*The suggestions regarding some minor flaws and typos are described as follows:*
*Page 5, line 125: delete the ".However".*
*Page 5, line 131: delete the "," between "1.23" and "km".*
*Page 6, line 138: move "that were used in the analysis" before "are located".*
*Page 9, line 212, 213: remove "as"*
*Page 9, line 212, 213: add "," before "including", "the estimating", and "is hampered".*
*Page 11, line 270: remove "," after the "both".*
*Figure 3: add (a) to (l) for each sub panel to facilitate a better reference to the figures in the text when making the explanations.*
*Page 22: change the subtitle to "Conclusions and Discussions".*
*Page 23, line 436: remove "that".*
*Page 23, line 438: change "constitute" to "conclude".*
*The research is sound and fundamental. Some language edits could be good for improving the paper's quality.*

**Authors' response #16: We agree to the suggestions and will change accordingly.**

**References**

[1] BGR. Digital soil map of Germany 1 : 200,000 (BUEK 200) v0.5, 2020.

[2] M. Dembélé, N. Ceperley, S. J. Zwart, E. Salvadore, G. Mariethoz, and B. Schaefli. Potential of satellite and reanalysis evaporation datasets for hydrological modelling under various model calibration strategies. *Advances in Water Resources*, 143:103667, Sept. 2020.

[3] R. Kumar, L. Samaniego, and S. Attinger. Implications of distributed hydrologic model parameterization on water fluxes at multiple scales and locations: DISTRIBUTED HYDROLOGIC MODEL PARAMETERIZATIONS. *Water Resources Research*, 49(1):360–379, Jan. 2013.

[4] B. Livneh, R. Kumar, and L. Samaniego. Influence of soil textural properties on hydrologic fluxes in the Mississippi river basin: Influence of Soil Textural Properties on Hydrologic Fluxes. *Hydrological Processes*, 29(21):4638–4655, Oct. 2015.

[5] J. Mai, B. A. Tolson, H. Shen, [U+FFFD] Gaborit, V. Fortin, N. Gasset, H. Awoye, T. A. Stadnyk, L. M. Fry, E. A. Bradley, F. Seglenieks, A. G. T. Temgoua, D. G. Princz, S. Gharari, A. Haghnegahdar, M. E. Elshamy, S. Razavi, M. Gauch, J. Lin, X. Ni, Y. Yuan, M. McLeod, N. B. Basu, R. Kumar, O. Rakovec, L. Samaniego, S. Attinger, N. K. Shrestha, P. Daggupati, T. Roy, S. Wi, T. Hunter, J. R. Craig, and A. Pietroniro. Great Lakes Runoff Intercomparison Project Phase 3: Lake Erie (GRIP-E). *Journal of Hydrologic Engineering*, 26(9):05021020, Sept. 2021.

[6] N. Mizukami, M. P. Clark, A. J. Newman, A. W. Wood, E. D. Gutmann, B. Nijssen, O. Rakovec, and L. Samaniego. Towards seamless large-domain parameter estimation for hydrologic models: LARGE-DOMAIN MODEL PARAMETERS. *Water Resources Research*, 53(9):8020–8040, Sept. 2017.

[7] O. Rakovec, R. Kumar, J. Mai, M. Cuntz, S. Thober, M. Zink, S. Attinger, D. Schäfer, M. Schrön, and L. Samaniego. Multiscale and Multivariate Evaluation of Water Fluxes and States over European River Basins. *Journal of Hydrometeorology*, 2016.

[8] O. Rakovec, N. Mizukami, R. Kumar, A. J. Newman, S. Thober, A. W. Wood, M. P. Clark, and L. Samaniego. Diagnostic Evaluation of Large[U+2010]Domain Hydrologic Models Calibrated Across the Contiguous United States. *Journal of Geophysical Research: Atmospheres*, 124(24):13991–14007, Dec. 2019.

[9] T. R. Saha, P. K. Shrestha, O. Rakovec, S. Thober, and L. Samaniego. A drought monitoring tool for South Asia. *Environmental Research Letters*, 16(5):054014, May 2021.

[10] L. Samaniego, R. Kumar, and S. Attinger. Multiscale parameter regionalization of a grid-based hydrologic model at the mesoscale: MULTISCALE PARAMETER REGIONALIZATION. *Water Resources Research*, 46(5), May 2010.

[11] L. Samaniego, R. Kumar, and M. Zink. Implications of parameter uncertainty on soil moisture drought analysis in Germany. *Journal of Hydrometeorology*, 14(1):47–68, 2013. ISBN: 1525-755X.

[12] L. Samaniego, S. Thober, R. Kumar, N. Wanders, O. Rakovec, M. Pan, M. Zink, J. Sheffield, E. F. Wood, and A. Marx. Anthropogenic warming exacerbates European soil moisture droughts. *Nature Climate Change*, 8(5):421–426, May 2018.

[13] R. Schweppe, S. Thober, S. Müller, M. Kelbling, R. Kumar, S. Attinger, and L. Samaniego. MPR 1.0: a stand-alone multiscale parameter regionalization tool for improved parameter estimation of land surface models. *Geoscientific Model Development*, 15(2):859–882, Jan. 2022.

[14] S. Thober, R. Kumar, J. Sheffield, J. Mai, D. Schäfer, and L. Samaniego. Seasonal Soil Moisture Drought Prediction over Europe Using the North American Multi-Model Ensemble (NMME). *Journal of Hydrometeorology*, 16(6):2329–2344, Dec. 2015.

**Appendix**

Table 1: Calibration results of the final selected best mHM parameter set for the 201 basins in Germany. Daily and monthly NSE and KGE metrics are shown as well as their three components r (correlation), alpha (bias) and beta (variability). The best calibration parameter set was selected based on the median of daily KGE over all 201 catchments. The IDs are GRDC

| num | ID | nse mon | kge mon | r mon | alpha mon | beta mon | nse day | kge day | r day | alpha day | beta day |
|---|---|---|---|---|---|---|---|---|---|---|---|
| 1 | 6337200 | 0.938 | 0.942 | 0.972 | 1.050 | 1.010 | 0.795 | 0.816 | 0.924 | 1.167 | 1.010 |
| 2 | 6335304 | 0.925 | 0.847 | 0.979 | 1.139 | 1.060 | 0.836 | 0.786 | 0.950 | 1.200 | 1.061 |
| 3 | 6337515 | 0.931 | 0.913 | 0.971 | 1.080 | 1.019 | 0.790 | 0.780 | 0.931 | 1.208 | 1.019 |
| 4 | 6335240 | 0.936 | 0.900 | 0.975 | 1.090 | 1.035 | 0.825 | 0.819 | 0.937 | 1.166 | 1.035 |
| 5 | 6337517 | 0.923 | 0.896 | 0.970 | 1.091 | 1.039 | 0.801 | 0.785 | 0.935 | 1.201 | 1.039 |
| 6 | 6337518 | 0.932 | 0.933 | 0.970 | 1.056 | 1.019 | 0.831 | 0.831 | 0.938 | 1.156 | 1.020 |
| 7 | 6335302 | 0.925 | 0.857 | 0.977 | 1.130 | 1.055 | 0.783 | 0.787 | 0.926 | 1.192 | 1.056 |
| 8 | 6337100 | 0.937 | 0.951 | 0.971 | 1.039 | 1.004 | 0.822 | 0.845 | 0.930 | 1.139 | 1.004 |
| 9 | 6337514 | 0.936 | 0.922 | 0.973 | 1.073 | 1.004 | 0.805 | 0.823 | 0.927 | 1.161 | 1.004 |
| 10 | 6337516 | 0.953 | 0.951 | 0.978 | 1.044 | 0.996 | 0.850 | 0.875 | 0.938 | 1.109 | 0.996 |
| 11 | 6337250 | 0.900 | 0.883 | 0.961 | 1.100 | 1.047 | 0.647 | 0.741 | 0.878 | 1.224 | 1.048 |
| 12 | 6335500 | 0.942 | 0.919 | 0.978 | 1.046 | 0.937 | 0.768 | 0.832 | 0.906 | 1.125 | 0.937 |
| 13 | 6337519 | 0.957 | 0.963 | 0.980 | 1.025 | 0.981 | 0.845 | 0.885 | 0.933 | 1.092 | 0.981 |
| 14 | 6335301 | 0.952 | 0.927 | 0.981 | 1.054 | 0.955 | 0.791 | 0.846 | 0.914 | 1.120 | 0.955 |
| 15 | 6335600 | 0.949 | 0.917 | 0.979 | 1.078 | 1.018 | 0.908 | 0.932 | 0.958 | 1.049 | 1.019 |
| 16 | 6337400 | 0.957 | 0.972 | 0.979 | 0.996 | 0.983 | 0.847 | 0.909 | 0.929 | 1.054 | 0.983 |
| 17 | 6335303 | 0.947 | 0.895 | 0.982 | 1.093 | 0.954 | 0.767 | 0.824 | 0.908 | 1.143 | 0.954 |
| 18 | 6338140 | 0.977 | 0.964 | 0.989 | 0.978 | 0.973 | 0.923 | 0.933 | 0.961 | 0.953 | 0.972 |
| 19 | 6338100 | 0.969 | 0.927 | 0.988 | 0.958 | 0.942 | 0.914 | 0.906 | 0.958 | 0.939 | 0.942 |
| 20 | 6335601 | 0.931 | 0.911 | 0.973 | 1.064 | 1.056 | 0.891 | 0.895 | 0.954 | 1.076 | 1.056 |
| 21 | 6337501 | 0.870 | 0.787 | 0.966 | 1.195 | 1.078 | 0.735 | 0.745 | 0.916 | 1.228 | 1.078 |
| 22 | 6335530 | 0.762 | 0.635 | 0.962 | 1.362 | 0.976 | 0.558 | 0.588 | 0.899 | 1.399 | 0.975 |
| 23 | 6337510 | 0.877 | 0.919 | 0.942 | 1.014 | 1.054 | 0.614 | 0.751 | 0.856 | 1.195 | 1.055 |
| 24 | 6337507 | 0.934 | 0.922 | 0.972 | 1.069 | 1.023 | 0.768 | 0.863 | 0.897 | 1.087 | 1.023 |
| 25 | 6340200 | 0.757 | 0.809 | 0.910 | 1.143 | 1.089 | 0.144 | 0.482 | 0.785 | 1.462 | 1.090 |
| 26 | 6337513 | 0.935 | 0.966 | 0.968 | 1.010 | 1.004 | 0.788 | 0.853 | 0.911 | 1.118 | 1.004 |
| 27 | 6337509 | 0.885 | 0.912 | 0.948 | 1.012 | 1.070 | 0.675 | 0.757 | 0.884 | 1.201 | 1.071 |
| 28 | 6337511 | 0.933 | 0.939 | 0.970 | 1.025 | 1.047 | 0.766 | 0.813 | 0.911 | 1.158 | 1.047 |
| 29 | 6338110 | 0.957 | 0.866 | 0.987 | 0.881 | 0.940 | 0.888 | 0.822 | 0.951 | 0.840 | 0.940 |
| 30 | 6337502 | 0.921 | 0.867 | 0.979 | 1.015 | 0.870 | 0.795 | 0.826 | 0.921 | 1.085 | 0.870 |
| 31 | 6337512 | 0.939 | 0.958 | 0.969 | 0.977 | 1.016 | 0.764 | 0.856 | 0.897 | 1.099 | 1.017 |
| 32 | 6335800 | 0.929 | 0.802 | 0.982 | 0.819 | 0.923 | 0.771 | 0.791 | 0.881 | 0.846 | 0.924 |
| 33 | 6335083 | 0.852 | 0.759 | 0.973 | 1.087 | 1.223 | 0.824 | 0.749 | 0.945 | 1.100 | 1.224 |
| 34 | 6335030 | 0.950 | 0.900 | 0.983 | 1.017 | 1.097 | 0.851 | 0.874 | 0.933 | 1.047 | 1.096 |
| 35 | 6342502 | 0.676 | 0.728 | 0.927 | 1.163 | 1.205 | 0.138 | 0.468 | 0.798 | 1.448 | 1.206 |
| 36 | 6335115 | 0.948 | 0.902 | 0.977 | 0.941 | 1.075 | 0.855 | 0.797 | 0.932 | 0.825 | 1.076 |
| 37 | 6335602 | 0.911 | 0.870 | 0.964 | 0.897 | 0.929 | 0.893 | 0.891 | 0.948 | 0.935 | 0.930 |
| 38 | 6338130 | 0.905 | 0.735 | 0.988 | 0.770 | 0.869 | 0.825 | 0.683 | 0.944 | 0.717 | 0.869 |
| 39 | 6335350 | 0.964 | 0.899 | 0.986 | 0.902 | 0.982 | 0.865 | 0.803 | 0.938 | 0.814 | 0.982 |
| 40 | 6340700 | 0.818 | 0.847 | 0.928 | 1.127 | 0.956 | 0.737 | 0.794 | 0.903 | 1.176 | 0.958 |
| 41 | 6337506 | 0.967 | 0.969 | 0.983 | 0.975 | 0.992 | 0.807 | 0.852 | 0.899 | 0.892 | 0.992 |
| 42 | 6338160 | 0.943 | 0.859 | 0.984 | 1.135 | 0.965 | 0.879 | 0.852 | 0.956 | 1.137 | 0.964 |
| 43 | 6337500 | 0.851 | 0.825 | 0.929 | 0.861 | 1.080 | 0.722 | 0.809 | 0.854 | 0.907 | 1.080 |
| 44 | 6338120 | 0.900 | 0.774 | 0.972 | 0.800 | 0.900 | 0.763 | 0.652 | 0.906 | 0.681 | 0.899 |
| 45 | 6335045 | 0.981 | 0.949 | 0.992 | 0.958 | 1.027 | 0.873 | 0.871 | 0.936 | 0.892 | 1.027 |
| 46 | 6335116 | 0.962 | 0.952 | 0.982 | 0.978 | 1.038 | 0.873 | 0.838 | 0.939 | 0.855 | 1.038 |
| 47 | 6342970 | 0.691 | 0.755 | 0.888 | 0.966 | 1.215 | 0.460 | 0.660 | 0.804 | 1.174 | 1.216 |
| 48 | 6337508 | 0.974 | 0.965 | 0.987 | 0.969 | 0.988 | 0.843 | 0.866 | 0.919 | 0.894 | 0.988 |
| 49 | 6342521 | 0.817 | 0.833 | 0.934 | 1.098 | 1.118 | 0.292 | 0.557 | 0.803 | 1.378 | 1.119 |
| 50 | 6338161 | 0.969 | 0.928 | 0.989 | 1.046 | 0.946 | 0.925 | 0.931 | 0.966 | 1.025 | 0.946 |
| 51 | 6335081 | 0.907 | 0.940 | 0.956 | 1.029 | 0.973 | 0.696 | 0.761 | 0.894 | 1.212 | 0.973 |
| 52 | 6335031 | 0.849 | 0.755 | 0.970 | 1.094 | 1.225 | 0.788 | 0.724 | 0.934 | 1.146 | 1.225 |
| 53 | 6335290 | 0.937 | 0.825 | 0.982 | 0.838 | 0.936 | 0.824 | 0.713 | 0.931 | 0.729 | 0.936 |
| 54 | 6335681 | 0.946 | 0.943 | 0.975 | 1.000 | 1.051 | 0.844 | 0.905 | 0.922 | 0.985 | 1.052 |
| 55 | 6335351 | 0.960 | 0.910 | 0.983 | 0.913 | 0.982 | 0.871 | 0.803 | 0.942 | 0.813 | 0.981 |
| 56 | 6335520 | 0.775 | 0.789 | 0.923 | 1.151 | 1.125 | 0.726 | 0.782 | 0.894 | 1.142 | 1.126 |
| 57 | 6342130 | 0.915 | 0.796 | 0.973 | 0.803 | 0.955 | 0.757 | 0.730 | 0.878 | 0.764 | 0.956 |
| 58 | 6338150 | 0.953 | 0.948 | 0.978 | 0.972 | 0.962 | 0.839 | 0.813 | 0.921 | 0.835 | 0.962 |
| 59 | 6335660 | 0.854 | 0.788 | 0.964 | 1.165 | 1.128 | 0.722 | 0.684 | 0.928 | 1.279 | 1.129 |
| 60 | 6335046 | 0.968 | 0.901 | 0.988 | 0.906 | 0.969 | 0.850 | 0.822 | 0.926 | 0.842 | 0.969 |
| 61 | 6337505 | 0.641 | 0.817 | 0.837 | 1.082 | 0.989 | 0.031 | 0.564 | 0.603 | 1.179 | 0.989 |
| 62 | 6340050 | -0.892 | -0.287 | 0.949 | 2.286 | 0.979 | -0.440 | -0.008 | 0.892 | 2.002 | 0.977 |
| 63 | 6341500 | 0.459 | 0.587 | 0.918 | 0.863 | 0.619 | 0.413 | 0.581 | 0.856 | 0.903 | 0.619 |
| 64 | 6342520 | 0.880 | 0.863 | 0.941 | 0.882 | 0.963 | 0.468 | 0.742 | 0.752 | 1.062 | 0.963 |
| 65 | 6340365 | 0.701 | 0.808 | 0.885 | 1.107 | 0.891 | 0.563 | 0.761 | 0.815 | 1.104 | 0.891 |

| 66 | 6337050 | 0.945 | 0.880 | 0.982 | 1.106 | 1.055 | 0.905 | 0.876 | 0.964 | 1.105 | 1.055 |
|----|---------|-------|-------|-------|-------|-------|-------|-------|-------|-------|-------|
| 67 | 6337504 | 0.936 | 0.800 | 0.985 | 0.801 | 0.999 | 0.829 | 0.776 | 0.920 | 0.791 | 0.999 |
| 68 | 6337610 | 0.940 | 0.914 | 0.971 | 0.920 | 0.993 | 0.772 | 0.884 | 0.891 | 1.037 | 0.993 |
| 69 | 6335160 | 0.862 | 0.760 | 0.976 | 1.141 | 1.193 | 0.828 | 0.751 | 0.949 | 1.147 | 1.194 |
| 70 | 6335082 | 0.698 | 0.721 | 0.922 | 1.193 | 1.186 | 0.315 | 0.465 | 0.866 | 1.483 | 1.186 |
| 71 | 6337550 | 0.909 | 0.817 | 0.967 | 0.854 | 0.894 | 0.806 | 0.710 | 0.919 | 0.742 | 0.894 |
| 72 | 6335521 | 0.883 | 0.864 | 0.942 | 0.878 | 1.019 | 0.802 | 0.801 | 0.899 | 0.830 | 1.021 |
| 73 | 6337601 | 0.949 | 0.940 | 0.977 | 0.988 | 0.946 | 0.887 | 0.874 | 0.944 | 0.900 | 0.946 |
| 74 | 6342522 | 0.881 | 0.813 | 0.953 | 0.877 | 1.133 | 0.623 | 0.756 | 0.806 | 0.935 | 1.134 |
| 75 | 6337060 | 0.879 | 0.857 | 0.961 | 0.983 | 0.864 | 0.789 | 0.778 | 0.904 | 0.854 | 0.863 |
| 76 | 6335125 | 0.922 | 0.893 | 0.967 | 0.940 | 1.082 | 0.867 | 0.886 | 0.934 | 0.958 | 1.083 |
| 77 | 6342110 | 0.512 | 0.440 | 0.946 | 1.556 | 1.038 | 0.315 | 0.344 | 0.922 | 1.650 | 1.038 |
| 78 | 6335310 | 0.961 | 0.942 | 0.983 | 1.041 | 1.038 | 0.751 | 0.861 | 0.887 | 1.072 | 1.038 |
| 79 | 6340320 | 0.850 | 0.914 | 0.925 | 0.978 | 1.035 | 0.508 | 0.725 | 0.811 | 1.197 | 1.036 |
| 80 | 6337570 | 0.932 | 0.912 | 0.972 | 1.078 | 0.972 | 0.837 | 0.914 | 0.919 | 1.002 | 0.972 |
| 81 | 6335048 | 0.967 | 0.891 | 0.991 | 1.093 | 1.056 | 0.886 | 0.918 | 0.946 | 1.027 | 1.055 |
| 82 | 6335460 | 0.918 | 0.952 | 0.961 | 1.021 | 1.018 | 0.856 | 0.880 | 0.926 | 0.908 | 1.019 |
| 83 | 6342980 | 0.920 | 0.840 | 0.968 | 0.846 | 0.970 | 0.832 | 0.883 | 0.913 | 0.928 | 0.971 |
| 84 | 6357020 | -0.979 | 0.370 | 0.837 | 0.894 | 0.401 | -0.841 | 0.364 | 0.795 | 0.951 | 0.400 |
| 85 | 6335291 | 0.916 | 0.785 | 0.976 | 0.792 | 0.951 | 0.824 | 0.724 | 0.928 | 0.738 | 0.951 |
| 86 | 6342670 | 0.653 | 0.655 | 0.943 | 1.286 | 1.185 | 0.632 | 0.747 | 0.828 | 0.989 | 1.185 |
| 87 | 6336930 | 0.971 | 0.970 | 0.986 | 1.012 | 1.023 | 0.908 | 0.935 | 0.953 | 0.961 | 1.023 |
| 88 | 6335540 | 0.911 | 0.793 | 0.971 | 0.799 | 1.038 | 0.828 | 0.811 | 0.914 | 0.837 | 1.038 |
| 89 | 6335670 | 0.726 | 0.620 | 0.969 | 1.265 | 1.271 | 0.557 | 0.504 | 0.922 | 1.408 | 1.271 |
| 90 | 6335695 | 0.920 | 0.879 | 0.963 | 0.887 | 1.020 | 0.873 | 0.871 | 0.935 | 0.891 | 1.021 |
| 91 | 6335076 | 0.965 | 0.934 | 0.985 | 0.988 | 1.063 | 0.859 | 0.847 | 0.929 | 0.880 | 1.063 |
| 92 | 6335360 | 0.967 | 0.875 | 0.990 | 0.876 | 1.013 | 0.890 | 0.836 | 0.949 | 0.844 | 1.013 |
| 93 | 6335450 | 0.906 | 0.858 | 0.957 | 0.868 | 0.968 | 0.844 | 0.841 | 0.921 | 0.866 | 0.969 |
| 94 | 6340810 | -1.298 | 0.423 | 0.717 | 1.038 | 0.499 | -0.560 | 0.436 | 0.744 | 0.978 | 0.498 |
| 95 | 6340070 | 0.410 | 0.367 | 0.963 | 1.590 | 1.227 | 0.372 | 0.368 | 0.940 | 1.587 | 1.227 |
| 96 | 6338250 | 0.896 | 0.880 | 0.956 | 0.937 | 0.908 | 0.788 | 0.757 | 0.899 | 0.800 | 0.907 |
| 97 | 6335510 | 0.054 | 0.385 | 0.830 | 1.525 | 1.273 | -0.368 | 0.140 | 0.824 | 1.796 | 1.273 |
| 98 | 6335690 | 0.937 | 0.895 | 0.979 | 1.076 | 1.069 | 0.802 | 0.807 | 0.898 | 0.851 | 1.068 |
| 99 | 6337530 | 0.967 | 0.911 | 0.986 | 0.918 | 0.967 | 0.864 | 0.817 | 0.936 | 0.832 | 0.967 |
| 100 | 6337560 | 0.783 | 0.795 | 0.888 | 0.842 | 0.932 | 0.672 | 0.684 | 0.824 | 0.746 | 0.932 |
| 101 | 6335675 | 0.905 | 0.893 | 0.964 | 1.063 | 1.078 | 0.702 | 0.715 | 0.843 | 0.776 | 1.079 |
| 102 | 6342120 | -0.134 | 0.066 | 0.942 | 1.929 | 1.073 | 0.273 | 0.420 | 0.872 | 1.560 | 1.074 |
| 103 | 6337520 | 0.930 | 0.809 | 0.979 | 0.814 | 0.963 | 0.803 | 0.789 | 0.900 | 0.818 | 0.962 |
| 104 | 6338800 | 0.927 | 0.945 | 0.966 | 1.023 | 0.964 | 0.882 | 0.929 | 0.941 | 0.984 | 0.964 |
| 105 | 6335710 | 0.929 | 0.917 | 0.969 | 0.968 | 1.070 | 0.822 | 0.887 | 0.912 | 0.991 | 1.070 |
| 106 | 6335603 | 0.919 | 0.820 | 0.971 | 0.827 | 0.959 | 0.783 | 0.697 | 0.906 | 0.714 | 0.960 |
| 107 | 6342070 | 0.497 | 0.756 | 0.763 | 1.054 | 1.020 | -0.000 | 0.519 | 0.679 | 1.357 | 1.021 |
| 108 | 6335032 | 0.927 | 0.942 | 0.966 | 1.043 | 1.019 | 0.727 | 0.839 | 0.882 | 1.108 | 1.019 |
| 109 | 6337410 | 0.924 | 0.872 | 0.969 | 0.903 | 0.921 | 0.780 | 0.710 | 0.900 | 0.740 | 0.921 |
| 110 | 6335650 | 0.702 | 0.712 | 0.919 | 1.263 | 1.087 | 0.620 | 0.711 | 0.876 | 1.245 | 1.088 |
| 111 | 6340218 | -2.784 | -0.761 | 0.922 | 2.755 | 0.878 | -1.504 | -0.268 | 0.824 | 2.250 | 0.877 |
| 112 | 6342640 | 0.532 | 0.507 | 0.924 | 1.485 | 1.047 | 0.472 | 0.722 | 0.790 | 1.176 | 1.047 |
| 113 | 6342100 | 0.337 | 0.445 | 0.887 | 1.541 | 0.953 | 0.524 | 0.654 | 0.858 | 1.312 | 0.954 |
| 114 | 6342655 | -0.166 | 0.287 | 0.893 | 1.651 | 1.272 | 0.423 | 0.611 | 0.815 | 1.207 | 1.272 |
| 115 | 6335167 | 0.329 | 0.362 | 0.969 | 1.560 | 1.304 | 0.397 | 0.417 | 0.923 | 1.492 | 1.303 |
| 116 | 6340220 | 0.369 | 0.507 | 0.776 | 1.042 | 1.437 | 0.339 | 0.480 | 0.765 | 1.151 | 1.438 |
| 117 | 6335075 | 0.784 | 0.739 | 0.940 | 1.254 | 0.986 | 0.624 | 0.793 | 0.847 | 1.139 | 0.984 |
| 118 | 6340335 | -3.081 | -0.289 | 0.340 | 1.755 | 1.811 | -2.381 | -0.162 | 0.168 | 1.638 | 1.501 |
| 119 | 6335810 | 0.938 | 0.873 | 0.975 | 0.898 | 1.071 | 0.786 | 0.734 | 0.898 | 0.765 | 1.071 |
| 120 | 6335565 | 0.834 | 0.782 | 0.931 | 0.832 | 1.120 | 0.736 | 0.655 | 0.880 | 0.700 | 1.120 |
| 121 | 6338163 | 0.917 | 0.889 | 0.970 | 1.081 | 0.930 | 0.819 | 0.888 | 0.914 | 1.019 | 0.930 |
| 122 | 6337340 | 0.899 | 0.929 | 0.951 | 0.982 | 1.048 | 0.699 | 0.805 | 0.879 | 1.144 | 1.048 |
| 123 | 6340210 | 0.921 | 0.957 | 0.962 | 1.017 | 1.011 | 0.839 | 0.850 | 0.917 | 0.876 | 1.010 |
| 124 | 6337541 | 0.891 | 0.940 | 0.945 | 0.988 | 0.977 | 0.814 | 0.903 | 0.910 | 1.030 | 0.977 |
| 125 | 6335117 | 0.766 | 0.759 | 0.914 | 0.912 | 0.793 | 0.713 | 0.697 | 0.867 | 0.824 | 0.793 |
| 126 | 6342675 | 0.612 | 0.667 | 0.919 | 1.271 | 1.175 | 0.588 | 0.670 | 0.779 | 0.828 | 1.175 |
| 127 | 6340350 | 0.716 | 0.674 | 0.935 | 1.141 | 1.287 | 0.624 | 0.667 | 0.862 | 1.095 | 1.288 |
| 128 | 6335190 | -1.020 | -0.127 | 0.839 | 2.103 | 1.164 | -1.454 | -0.252 | 0.810 | 2.227 | 1.162 |
| 129 | 6342230 | 0.082 | 0.439 | 0.934 | 1.417 | 1.369 | 0.021 | 0.252 | 0.896 | 1.642 | 1.370 |
| 130 | 6337310 | 0.914 | 0.844 | 0.966 | 0.888 | 1.103 | 0.852 | 0.812 | 0.929 | 0.860 | 1.103 |
| 131 | 6340315 | 0.889 | 0.826 | 0.959 | 0.986 | 1.168 | 0.782 | 0.800 | 0.903 | 1.044 | 1.169 |
| 132 | 6335621 | -1.611 | -0.208 | -0.203 | 1.075 | 1.090 | -1.398 | -0.084 | -0.075 | 1.107 | 1.088 |
| 133 | 6340440 | 0.879 | 0.861 | 0.959 | 1.085 | 0.898 | 0.768 | 0.813 | 0.882 | 0.897 | 0.897 |
| 134 | 6340360 | 0.764 | 0.644 | 0.911 | 0.658 | 1.042 | 0.672 | 0.617 | 0.841 | 0.654 | 1.044 |
| 135 | 6340366 | 0.868 | 0.838 | 0.946 | 0.892 | 0.893 | 0.647 | 0.612 | 0.822 | 0.672 | 0.892 |
| 136 | 6337542 | 0.845 | 0.826 | 0.941 | 0.912 | 0.862 | 0.808 | 0.835 | 0.913 | 0.973 | 0.862 |
| 137 | 6335470 | 0.663 | 0.641 | 0.950 | 1.323 | 1.150 | 0.694 | 0.712 | 0.914 | 1.230 | 1.151 |
| 138 | 6336510 | 0.881 | 0.743 | 0.982 | 1.107 | 1.233 | 0.821 | 0.754 | 0.923 | 0.983 | 1.233 |
| 139 | 6342105 | 0.513 | 0.484 | 0.929 | 1.509 | 1.043 | 0.622 | 0.643 | 0.904 | 1.341 | 1.043 |
| 140 | 6335465 | 0.871 | 0.868 | 0.945 | 0.915 | 1.084 | 0.819 | 0.851 | 0.908 | 0.919 | 1.084 |
| 141 | 6335697 | 0.918 | 0.895 | 0.963 | 0.940 | 1.077 | 0.675 | 0.628 | 0.839 | 0.673 | 1.077 |

| 142 | 6335610 | -0.386 | 0.281 | 0.732 | 1.655 | 1.128 | -0.707 | 0.063 | 0.777 | 1.897 | 1.158 |
| 143 | 6342060 | 0.812 | 0.825 | 0.905 | 0.863 | 1.052 | 0.624 | 0.808 | 0.815 | 1.011 | 1.052 |
| 144 | 6335010 | 0.938 | 0.886 | 0.972 | 0.894 | 0.969 | 0.867 | 0.823 | 0.937 | 0.838 | 0.969 |
| 145 | 6337350 | 0.903 | 0.856 | 0.955 | 0.864 | 0.982 | 0.585 | 0.794 | 0.811 | 1.079 | 0.983 |
| 146 | 6340330 | 0.766 | 0.771 | 0.923 | 0.894 | 0.812 | 0.714 | 0.716 | 0.870 | 0.831 | 0.812 |
| 147 | 6340225 | -1.133 | -0.213 | 0.853 | 2.203 | 0.938 | -1.085 | -0.126 | 0.796 | 2.107 | 1.014 |
| 148 | 6337600 | 0.923 | 0.902 | 0.969 | 0.985 | 0.909 | 0.824 | 0.825 | 0.912 | 0.881 | 0.908 |
| 149 | 6335651 | 0.537 | 0.628 | 0.882 | 1.344 | 0.924 | 0.442 | 0.663 | 0.813 | 1.270 | 0.924 |
| 150 | 6335640 | -0.024 | 0.255 | 0.921 | 1.649 | 1.357 | 0.275 | 0.353 | 0.892 | 1.529 | 1.357 |
| 151 | 6337320 | 0.920 | 0.945 | 0.960 | 0.975 | 0.972 | 0.842 | 0.905 | 0.927 | 1.053 | 0.972 |
| 152 | 6342820 | 0.930 | 0.916 | 0.971 | 1.069 | 1.038 | 0.811 | 0.878 | 0.902 | 0.937 | 1.038 |
| 153 | 6340400 | 0.694 | 0.765 | 0.895 | 1.186 | 0.902 | 0.583 | 0.718 | 0.773 | 0.865 | 0.902 |
| 154 | 6335730 | 0.905 | 0.911 | 0.957 | 0.976 | 1.074 | 0.701 | 0.796 | 0.881 | 1.148 | 1.074 |
| 155 | 6335820 | 0.771 | 0.696 | 0.960 | 1.230 | 1.194 | 0.692 | 0.678 | 0.910 | 1.240 | 1.195 |
| 156 | 6342081 | 0.044 | 0.399 | 0.916 | 1.326 | 1.498 | 0.392 | 0.444 | 0.859 | 1.199 | 1.499 |
| 157 | 6335175 | 0.028 | 0.199 | 0.934 | 1.770 | 1.210 | 0.042 | 0.246 | 0.899 | 1.717 | 1.209 |
| 158 | 6335035 | 0.814 | 0.812 | 0.905 | 0.838 | 0.994 | 0.403 | 0.509 | 0.636 | 0.670 | 0.995 |
| 159 | 6338260 | -0.863 | -0.249 | 0.935 | 2.247 | 0.961 | -0.379 | 0.195 | 0.783 | 1.774 | 0.959 |
| 160 | 6335676 | 0.857 | 0.843 | 0.952 | 1.047 | 1.141 | 0.627 | 0.641 | 0.800 | 0.738 | 1.142 |
| 161 | 6335700 | -3.456 | -0.483 | 0.862 | 1.887 | 2.181 | -1.554 | -0.472 | 0.749 | 1.839 | 2.183 |
| 162 | 6335680 | 0.889 | 0.753 | 0.966 | 0.756 | 0.990 | 0.790 | 0.715 | 0.906 | 0.731 | 0.991 |
| 163 | 6357510 | 0.700 | 0.765 | 0.864 | 0.885 | 0.847 | 0.480 | 0.488 | 0.709 | 0.607 | 0.847 |
| 164 | 6335830 | -0.740 | -0.046 | 0.836 | 2.033 | 1.034 | -0.839 | -0.063 | 0.819 | 2.047 | 1.032 |
| 165 | 6335550 | 0.441 | 0.612 | 0.845 | 0.777 | 0.723 | 0.333 | 0.632 | 0.777 | 1.095 | 0.724 |
| 166 | 6340340 | 0.747 | 0.604 | 0.915 | 0.616 | 0.954 | 0.632 | 0.510 | 0.860 | 0.532 | 0.955 |
| 167 | 6335620 | 0.900 | 0.838 | 0.962 | 1.016 | 1.157 | 0.841 | 0.807 | 0.921 | 0.922 | 1.157 |
| 168 | 6342810 | 0.418 | 0.575 | 0.906 | 1.142 | 1.389 | -0.102 | 0.331 | 0.776 | 1.496 | 1.390 |
| 169 | 6337330 | 0.891 | 0.857 | 0.948 | 0.870 | 0.972 | 0.823 | 0.853 | 0.908 | 0.889 | 0.972 |
| 170 | 6342525 | 0.774 | 0.762 | 0.938 | 1.134 | 1.187 | 0.738 | 0.776 | 0.890 | 1.056 | 1.187 |
| 171 | 6335635 | 0.917 | 0.931 | 0.958 | 0.945 | 0.998 | 0.674 | 0.655 | 0.832 | 0.698 | 0.999 |
| 172 | 6335720 | 0.896 | 0.831 | 0.955 | 0.848 | 1.058 | 0.789 | 0.856 | 0.890 | 0.926 | 1.057 |
| 173 | 6342660 | 0.797 | 0.874 | 0.910 | 1.088 | 1.007 | 0.716 | 0.786 | 0.846 | 0.851 | 1.008 |
| 174 | 6335696 | 0.854 | 0.832 | 0.928 | 0.848 | 0.993 | 0.738 | 0.870 | 0.872 | 1.021 | 0.994 |
| 175 | 6335155 | -1.910 | -0.279 | 0.896 | 2.177 | 1.490 | -3.707 | -0.717 | 0.729 | 2.623 | 1.490 |
| 176 | 6335671 | 0.834 | 0.795 | 0.949 | 1.138 | 1.143 | 0.417 | 0.562 | 0.849 | 1.386 | 1.143 |
| 177 | 6342050 | 0.380 | 0.425 | 0.834 | 0.581 | 0.643 | 0.432 | 0.369 | 0.769 | 0.534 | 0.643 |
| 178 | 6342125 | 0.229 | 0.381 | 0.931 | 1.580 | 1.205 | 0.588 | 0.724 | 0.822 | 1.047 | 1.206 |
| 179 | 6342571 | -0.162 | 0.491 | 0.574 | 1.256 | 1.107 | -0.455 | 0.347 | 0.615 | 1.517 | 1.106 |
| 180 | 6335560 | 0.691 | 0.746 | 0.887 | 0.955 | 1.222 | 0.703 | 0.703 | 0.855 | 0.868 | 1.223 |
| 181 | 6335665 | 0.767 | 0.818 | 0.925 | 0.932 | 0.848 | 0.593 | 0.690 | 0.788 | 0.833 | 0.849 |
| 182 | 6335725 | 0.881 | 0.839 | 0.944 | 0.851 | 1.026 | 0.801 | 0.817 | 0.896 | 0.852 | 1.026 |
| 183 | 6337590 | 0.799 | 0.625 | 0.961 | 0.811 | 1.321 | 0.800 | 0.618 | 0.921 | 0.811 | 1.322 |
| 184 | 6335677 | 0.884 | 0.784 | 0.955 | 0.792 | 0.964 | 0.649 | 0.579 | 0.836 | 0.614 | 0.964 |
| 185 | 6342960 | 0.863 | 0.862 | 0.946 | 1.127 | 0.989 | 0.706 | 0.852 | 0.852 | 0.994 | 0.990 |
| 186 | 6335485 | -0.041 | 0.306 | 0.891 | 1.595 | 1.339 | 0.038 | 0.372 | 0.800 | 1.489 | 1.339 |
| 187 | 6335156 | 0.812 | 0.740 | 0.952 | 1.145 | 1.211 | 0.765 | 0.766 | 0.896 | 1.002 | 1.210 |
| 188 | 6335735 | 0.921 | 0.851 | 0.975 | 0.923 | 1.125 | 0.810 | 0.787 | 0.906 | 0.857 | 1.126 |
| 189 | 6337503 | 0.557 | 0.767 | 0.815 | 1.136 | 0.962 | -0.284 | 0.447 | 0.563 | 1.338 | 0.961 |
| 190 | 6343537 | 0.569 | 0.685 | 0.918 | 1.199 | 1.230 | 0.546 | 0.629 | 0.863 | 1.257 | 1.230 |
| 191 | 6335165 | -0.127 | 0.194 | 0.934 | 1.729 | 1.337 | 0.007 | 0.289 | 0.866 | 1.612 | 1.337 |
| 192 | 6340216 | -0.755 | 0.075 | 0.927 | 1.632 | 1.672 | -0.028 | 0.242 | 0.828 | 1.309 | 1.671 |
| 193 | 6337580 | 0.910 | 0.865 | 0.958 | 0.873 | 0.985 | 0.807 | 0.784 | 0.904 | 0.807 | 0.985 |
| 194 | 6335570 | 0.818 | 0.853 | 0.912 | 0.903 | 0.933 | 0.742 | 0.853 | 0.871 | 0.979 | 0.932 |
| 195 | 6342945 | 0.635 | 0.805 | 0.849 | 1.027 | 1.120 | 0.570 | 0.764 | 0.799 | 1.032 | 1.120 |
| 196 | 6343120 | 0.547 | 0.633 | 0.887 | 1.027 | 1.348 | 0.511 | 0.578 | 0.834 | 1.174 | 1.347 |
| 197 | 6343520 | 0.722 | 0.817 | 0.902 | 1.059 | 1.143 | 0.638 | 0.777 | 0.834 | 1.044 | 1.142 |
| 198 | 6342540 | -0.031 | 0.439 | 0.818 | 1.352 | 1.397 | 0.306 | 0.504 | 0.755 | 1.169 | 1.397 |
| 199 | 6342940 | -7.264 | -1.095 | 0.607 | 3.034 | 1.311 | -15.740 | -2.599 | 0.661 | 4.569 | 1.311 |
| 200 | 6337300 | 0.900 | 0.831 | 0.964 | 0.903 | 1.134 | 0.846 | 0.799 | 0.930 | 0.870 | 1.136 |
| 201 | 6342947 | 0.778 | 0.845 | 0.904 | 0.981 | 1.120 | 0.679 | 0.748 | 0.828 | 0.861 | 1.121 |

Table 2: 200 sets of random multi-basin draws used for multi-basin calibrations.

| | basin 1 | basin 2 | basin 3 | basin 4 | basin 5 | basin 6 |
|---|---|---|---|---|---|---|
| N01 | 6336930 | 6338140 | 6337502 | 6340360 | 6335725 | 6335116 |
| N02 | 6335083 | 6342070 | 6338800 | 6335510 | 6337505 | 6335303 |
| N03 | 6335602 | 6335310 | 6338161 | 6335450 | 6335676 | 6342081 |
| N04 | 6338130 | 6337580 | 6337410 | 6335720 | 6335075 | 6335650 |
| N05 | 6335710 | 6340220 | 6335610 | 6335650 | 6342525 | 6335190 |
| N06 | 6335650 | 6336510 | 6338140 | 6341500 | 6335800 | 6335830 |
| N07 | 6335660 | 6340315 | 6335460 | 6335665 | 6340335 | 6335730 |
| N08 | 6342571 | 6337501 | 6335700 | 6340216 | 6335820 | 6335800 |
| N09 | 6335603 | 6335602 | 6337518 | 6335470 | 6337517 | 6338800 |
| N10 | 6337509 | 6337503 | 6335450 | 6335697 | 6335696 | 6357510 |

| | | | | | | |
|---|---|---|---|---|---|---|
| N11 | 6335560 | 6337509 | 6335830 | 6338250 | 6342970 | 6338100 |
| N12 | 6335710 | 6340218 | 6335351 | 6337310 | 6335010 | 6335290 |
| N13 | 6337502 | 6335697 | 6335167 | 6335570 | 6337250 | 6342130 |
| N14 | 6337310 | 6337100 | 6335696 | 6337500 | 6336930 | 6335620 |
| N15 | 6335175 | 6335565 | 6337310 | 6335155 | 6337500 | 6337504 |
| N16 | 6342820 | 6335621 | 6337510 | 6337550 | 6335117 | 6335620 |
| N17 | 6340366 | 6335360 | 6342105 | 6335160 | 6342120 | 6337512 |
| N18 | 6337510 | 6335310 | 6335510 | 6335155 | 6338140 | 6338161 |
| N19 | 6337508 | 6342081 | 6335156 | 6335696 | 6335725 | 6335570 |
| N20 | 6335117 | 6337520 | 6335032 | 6338800 | 6335075 | 6335301 |
| N21 | 6337502 | 6335032 | 6342980 | 6337550 | 6337511 | 6342070 |
| N22 | 6337514 | 6338160 | 6357510 | 6337500 | 6340350 | 6335125 |
| N23 | 6337518 | 6337513 | 6342060 | 6338110 | 6337503 | 6338161 |
| N24 | 6340350 | 6338250 | 6340335 | 6335810 | 6335116 | 6342960 |
| N25 | 6342120 | 6335302 | 6340320 | 6335031 | 6335600 | 6338130 |
| N26 | 6335820 | 6340810 | 6335603 | 6335465 | 6337610 | 6335031 |
| N27 | 6335560 | 6335485 | 6335160 | 6335640 | 6335676 | 6335700 |
| N28 | 6337501 | 6335677 | 6337506 | 6335820 | 6342050 | 6335810 |
| N29 | 6337330 | 6335621 | 6342980 | 6342960 | 6335500 | 6337500 |
| N30 | 6340200 | 6335635 | 6335031 | 6337400 | 6337512 | 6336930 |
| N31 | 6338161 | 6340360 | 6340366 | 6340220 | 6335117 | 6337511 |
| N32 | 6335720 | 6335125 | 6340350 | 6337601 | 6337505 | 6335485 |
| N33 | 6335603 | 6335697 | 6335360 | 6335303 | 6338140 | 6342522 |
| N34 | 6338161 | 6335290 | 6335681 | 6342105 | 6337410 | 6335510 |
| N35 | 6335160 | 6342655 | 6338140 | 6337519 | 6342120 | 6338163 |
| N36 | 6337506 | 6342105 | 6335310 | 6335290 | 6335035 | 6342810 |
| N37 | 6335635 | 6335700 | 6335303 | 6338100 | 6337514 | 6340210 |
| N38 | 6335190 | 6337518 | 6335820 | 6335167 | 6335530 | 6340440 |
| N39 | 6337541 | 6335510 | 6335010 | 6342571 | 6338140 | 6335450 |
| N40 | 6342520 | 6335601 | 6335540 | 6335660 | 6337530 | 6335116 |
| N41 | 6337320 | 6336930 | 6335696 | 6335600 | 6335620 | 6335032 |
| N42 | 6335117 | 6335302 | 6337340 | 6335500 | 6340220 | 6342100 |
| N43 | 6342521 | 6335302 | 6340220 | 6337560 | 6335570 | 6337350 |
| N44 | 6335540 | 6335035 | 6337501 | 6335350 | 6337512 | 6335650 |
| N45 | 6335302 | 6337550 | 6335500 | 6338800 | 6342060 | 6335690 |
| N46 | 6337504 | 6335565 | 6335650 | 6340360 | 6337501 | 6337570 |
| N47 | 6340400 | 6340365 | 6335165 | 6335485 | 6335570 | 6335640 |
| N48 | 6335290 | 6342521 | 6335500 | 6335550 | 6338110 | 6337501 |
| N49 | 6343537 | 6340365 | 6337050 | 6335660 | 6335082 | 6335310 |
| N50 | 6342960 | 6335048 | 6335190 | 6337500 | 6340320 | 6335082 |
| N51 | 6335117 | 6335601 | 6335291 | 6335600 | 6337516 | 6335175 |
| N52 | 6337600 | 6337508 | 6337330 | 6335035 | 6336930 | 6342130 |
| N53 | 6335045 | 6335540 | 6337518 | 6338140 | 6335695 | 6335301 |
| N54 | 6335530 | 6335521 | 6335730 | 6335550 | 6335700 | 6335676 |
| N55 | 6335540 | 6335460 | 6335651 | 6335465 | 6335175 | 6335810 |
| N56 | 6335635 | 6335730 | 6340315 | 6337512 | 6342571 | 6342675 |
| N57 | 6335032 | 6335031 | 6335725 | 6335681 | 6335115 | 6335800 |
| N58 | 6340700 | 6335116 | 6335725 | 6342521 | 6335083 | 6340050 |
| N59 | 6335076 | 6338150 | 6335695 | 6335675 | 6340216 | 6335603 |
| N60 | 6335620 | 6337310 | 6335697 | 6335680 | 6342980 | 6337542 |
| N61 | 6340335 | 6335046 | 6335032 | 6335081 | 6357020 | 6338163 |
| N62 | 6342120 | 6335470 | 6340218 | 6337340 | 6335697 | 6335720 |
| N63 | 6335117 | 6338130 | 6338161 | 6335076 | 6335465 | 6337508 |
| N64 | 6335695 | 6335540 | 6335730 | 6335610 | 6337514 | 6337542 |
| N65 | 6342120 | 6338130 | 6340200 | 6335540 | 6335030 | 6335165 |
| N66 | 6338140 | 6335030 | 6338260 | 6335470 | 6340220 | 6342125 |
| N67 | 6337511 | 6340070 | 6337509 | 6337503 | 6342960 | 6335820 |
| N68 | 6342130 | 6340315 | 6340700 | 6335650 | 6342980 | 6338110 |
| N69 | 6340200 | 6342520 | 6337512 | 6335735 | 6337590 | 6335450 |
| N70 | 6335800 | 6335710 | 6335082 | 6335510 | 6337507 | 6342105 |
| N71 | 6335075 | 6335710 | 6336510 | 6335570 | 6335076 | 6340330 |
| N72 | 6335640 | 6335675 | 6335076 | 6337542 | 6342130 | 6338800 |
| N73 | 6335175 | 6335035 | 6337550 | 6337512 | 6335820 | 6338161 |
| N74 | 6337507 | 6337516 | 6337519 | 6340365 | 6337530 | 6337060 |
| N75 | 6342970 | 6340366 | 6338120 | 6340070 | 6335510 | 6340340 |
| N76 | 6335310 | 6335600 | 6335510 | 6337501 | 6335301 | 6335640 |
| N77 | 6335155 | 6342670 | 6335075 | 6337250 | 6342502 | 6342125 |
| N78 | 6335720 | 6335640 | 6338100 | 6337504 | 6335465 | 6340340 |
| N79 | 6335155 | 6335075 | 6335540 | 6335620 | 6335635 | 6342050 |
| N80 | 6338100 | 6335031 | 6335830 | 6342502 | 6335735 | 6335650 |
| N81 | 6335156 | 6335730 | 6335602 | 6335190 | 6340200 | 6340810 |
| N82 | 6340400 | 6342105 | 6336510 | 6337510 | 6342502 | 6337410 |
| N83 | 6335076 | 6337580 | 6335470 | 6337520 | 6335601 | 6337505 |
| N84 | 6340335 | 6338260 | 6335671 | 6342230 | 6335031 | 6335710 |
| N85 | 6337520 | 6335046 | 6337503 | 6335450 | 6342081 | 6338260 |
| N86 | 6342525 | 6335675 | 6340225 | 6335465 | 6340350 | 6335800 |

| | | | | | |
|---|---|---|---|---|---|
| N87 | 6337350 | 6341500 | 6337550 | 6335160 | 6340216 | 6342960 |
| N88 | 6335570 | 6335510 | 6335310 | 6335671 | 6335290 | 6335500 |
| N89 | 6337502 | 6335690 | 6335360 | 6335155 | 6342675 | 6342810 |
| N90 | 6337580 | 6338161 | 6336510 | 6335030 | 6335671 | 6340365 |
| N91 | 6335303 | 6335560 | 6335115 | 6335083 | 6335621 | 6335031 |
| N92 | 6335540 | 6335651 | 6335290 | 6342655 | 6335676 | 6357020 |
| N93 | 6335700 | 6335820 | 6342640 | 6335520 | 6342670 | 6340365 |
| N94 | 6335165 | 6335117 | 6335651 | 6340400 | 6335470 | 6340315 |
| N95 | 6337503 | 6340400 | 6335485 | 6337580 | 6335030 | 6338130 |
| N96 | 6335621 | 6335651 | 6338800 | 6338163 | 6335800 | 6337513 |
| N97 | 6338100 | 6335550 | 6335115 | 6335190 | 6335677 | 6337250 |
| N98 | 6335660 | 6342130 | 6342070 | 6335640 | 6340366 | 6335117 |
| N99 | 6357510 | 6342820 | 6342520 | 6335155 | 6342060 | 6342081 |
| N100 | 6335620 | 6337550 | 6335083 | 6337509 | 6335640 | 6337330 |
| N101 | 6342125 | 6337560 | 6335602 | 6342810 | 6337506 | 6335450 |
| N102 | 6335710 | 6341500 | 6335610 | 6340340 | 6335677 | 6342810 |
| N103 | 6335565 | 6335570 | 6337610 | 6335690 | 6335301 | 6335725 |
| N104 | 6335602 | 6335046 | 6342960 | 6335621 | 6335570 | 6335031 |
| N105 | 6340210 | 6338163 | 6335485 | 6340050 | 6335696 | 6335450 |
| N106 | 6335035 | 6338161 | 6342960 | 6337509 | 6335485 | 6335610 |
| N107 | 6335165 | 6335351 | 6337516 | 6335635 | 6335291 | 6335620 |
| N108 | 6335048 | 6342525 | 6338110 | 6340216 | 6337507 | 6335083 |
| N109 | 6335602 | 6342230 | 6337501 | 6335640 | 6335350 | 6335735 |
| N110 | 6335602 | 6342230 | 6340366 | 6337504 | 6337507 | 6335083 |
| N111 | 6337400 | 6335190 | 6357510 | 6335117 | 6335303 | 6342640 |
| N112 | 6337542 | 6342960 | 6342125 | 6335602 | 6335601 | 6336510 |
| N113 | 6337590 | 6335290 | 6340220 | 6335175 | 6337508 | 6336930 |
| N114 | 6335010 | 6335671 | 6335081 | 6337516 | 6338260 | 6342521 |
| N115 | 6338100 | 6335665 | 6335465 | 6335115 | 6342655 | 6335725 |
| N116 | 6335670 | 6335665 | 6342521 | 6342081 | 6335565 | 6342060 |
| N117 | 6337400 | 6335681 | 6337507 | 6335291 | 6337310 | 6335521 |
| N118 | 6337250 | 6337502 | 6335565 | 6335167 | 6335010 | 6342960 |
| N119 | 6335302 | 6335540 | 6335710 | 6341500 | 6337511 | 6337505 |
| N120 | 6335620 | 6337516 | 6335500 | 6337501 | 6335075 | 6342105 |
| N121 | 6340315 | 6335175 | 6337514 | 6340440 | 6335081 | 6335035 |
| N122 | 6337050 | 6337060 | 6342230 | 6337100 | 6337310 | 6342100 |
| N123 | 6340225 | 6337400 | 6342130 | 6342655 | 6335675 | 6340335 |
| N124 | 6337550 | 6337600 | 6342520 | 6337509 | 6335165 | 6335310 |
| N125 | 6335520 | 6335081 | 6335603 | 6335680 | 6335125 | 6342230 |
| N126 | 6335160 | 6338130 | 6342105 | 6337320 | 6341500 | 6340225 |
| N127 | 6342520 | 6337350 | 6335603 | 6335810 | 6338100 | 6337509 |
| N128 | 6337541 | 6337590 | 6335601 | 6335675 | 6343537 | 6335680 |
| N129 | 6337530 | 6335303 | 6335602 | 6335696 | 6337060 | 6337600 |
| N130 | 6335083 | 6337517 | 6341500 | 6337513 | 6342571 | 6335810 |
| N131 | 6335603 | 6335635 | 6342670 | 6335697 | 6337501 | 6338140 |
| N132 | 6335115 | 6337506 | 6337520 | 6335165 | 6335530 | 6337518 |
| N133 | 6335030 | 6335165 | 6357020 | 6335302 | 6335520 | 6340400 |
| N134 | 6337590 | 6335290 | 6342640 | 6335032 | 6335450 | 6335650 |
| N135 | 6335360 | 6340320 | 6342520 | 6335601 | 6357020 | 6337601 |
| N136 | 6340200 | 6338100 | 6340220 | 6337310 | 6335116 | 6342230 |
| N137 | 6337520 | 6335301 | 6335117 | 6335671 | 6335500 | 6337400 |
| N138 | 6335600 | 6335075 | 6335082 | 6335116 | 6338150 | 6342081 |
| N139 | 6342640 | 6335671 | 6336930 | 6338160 | 6337505 | 6340335 |
| N140 | 6357020 | 6335302 | 6342522 | 6335665 | 6338150 | 6340360 |
| N141 | 6335465 | 6338160 | 6342675 | 6337509 | 6340340 | 6335720 |
| N142 | 6340366 | 6335830 | 6337610 | 6337502 | 6335610 | 6338150 |
| N143 | 6335083 | 6337060 | 6342655 | 6335690 | 6335450 | 6335670 |
| N144 | 6337310 | 6335290 | 6335160 | 6335048 | 6335310 | 6337503 |
| N145 | 6337506 | 6335635 | 6342060 | 6343537 | 6340050 | 6335621 |
| N146 | 6335600 | 6338163 | 6335116 | 6335076 | 6335540 | 6335310 |
| N147 | 6337505 | 6337506 | 6335675 | 6335310 | 6337503 | 6341500 |
| N148 | 6340700 | 6342970 | 6335048 | 6337505 | 6335520 | 6338140 |
| N149 | 6342081 | 6340400 | 6342100 | 6335125 | 6335675 | 6336930 |
| N150 | 6335010 | 6340070 | 6335048 | 6337508 | 6335610 | 6335671 |
| N151 | 6335565 | 6337509 | 6335735 | 6338161 | 6338110 | 6335601 |
| N152 | 6342521 | 6335670 | 6338800 | 6335301 | 6335530 | 6335155 |
| N153 | 6342070 | 6335650 | 6337504 | 6335560 | 6335820 | 6337590 |
| N154 | 6337530 | 6335681 | 6337060 | 6335603 | 6335601 | 6338120 |
| N155 | 6342125 | 6340070 | 6335031 | 6335485 | 6335603 | 6335465 |
| N156 | 6335115 | 6335735 | 6335450 | 6340700 | 6335076 | 6340810 |
| N157 | 6335601 | 6335603 | 6335076 | 6335635 | 6335820 | 6335697 |
| N158 | 6335830 | 6335115 | 6342980 | 6335485 | 6337320 | 6337506 |
| N159 | 6337050 | 6336930 | 6335075 | 6340320 | 6342960 | 6342120 |
| N160 | 6337504 | 6340216 | 6335032 | 6337516 | 6342125 | 6335125 |
| N161 | 6335600 | 6335083 | 6335350 | 6335048 | 6338161 | 6335530 |
| N162 | 6335621 | 6335820 | 6335610 | 6342070 | 6335156 | 6342655 |

| N163 | 6335695 | 6338130 | 6338260 | 6342640 | 6335520 | 6342050 |
| N164 | 6335046 | 6335601 | 6342110 | 6337500 | 6335350 | 6335610 |
| N165 | 6335697 | 6337503 | 6335725 | 6337590 | 6340366 | 6337513 |
| N166 | 6337517 | 6335680 | 6340360 | 6340216 | 6342502 | 6357510 |
| N167 | 6337514 | 6343537 | 6335710 | 6342125 | 6337410 | 6335302 |
| N168 | 6335676 | 6335680 | 6342810 | 6335303 | 6342130 | 6337550 |
| N169 | 6335081 | 6335530 | 6336510 | 6335601 | 6337410 | 6342130 |
| N170 | 6342110 | 6335670 | 6335696 | 6340210 | 6335800 | 6337513 |
| N171 | 6337580 | 6337507 | 6335470 | 6335115 | 6335570 | 6335510 |
| N172 | 6337570 | 6337530 | 6337503 | 6342070 | 6343537 | 6335725 |
| N173 | 6342120 | 6337610 | 6342660 | 6335565 | 6343537 | 6340320 |
| N174 | 6335635 | 6335031 | 6340400 | 6335165 | 6337350 | 6335800 |
| N175 | 6335010 | 6335621 | 6335675 | 6340050 | 6335046 | 6340366 |
| N176 | 6340200 | 6335720 | 6342810 | 6342125 | 6340330 | 6335710 |
| N177 | 6342640 | 6342810 | 6337508 | 6337507 | 6337400 | 6335302 |
| N178 | 6342521 | 6340220 | 6335010 | 6342655 | 6337320 | 6337504 |
| N179 | 6337320 | 6342670 | 6338160 | 6338161 | 6337542 | 6342521 |
| N180 | 6342670 | 6335600 | 6337512 | 6342522 | 6335290 | 6335695 |
| N181 | 6335560 | 6335665 | 6340330 | 6337513 | 6335125 | 6335690 |
| N182 | 6335680 | 6337500 | 6335030 | 6337519 | 6342060 | 6337601 |
| N183 | 6357020 | 6335720 | 6335082 | 6338163 | 6342970 | 6337516 |
| N184 | 6340350 | 6337330 | 6337560 | 6340320 | 6337500 | 6335115 |
| N185 | 6357020 | 6338110 | 6337505 | 6338120 | 6338150 | 6337511 |
| N186 | 6335610 | 6335720 | 6335290 | 6337510 | 6335046 | 6335650 |
| N187 | 6340216 | 6335082 | 6342105 | 6335550 | 6335351 | 6337560 |
| N188 | 6335301 | 6342960 | 6337330 | 6335360 | 6335520 | 6335735 |
| N189 | 6337508 | 6335155 | 6337516 | 6337512 | 6337511 | 6342675 |
| N190 | 6340440 | 6337580 | 6335470 | 6336930 | 6338150 | 6342060 |
| N191 | 6340810 | 6342655 | 6335310 | 6342670 | 6337508 | 6342070 |
| N192 | 6335010 | 6342980 | 6335680 | 6335600 | 6335621 | 6335565 |
| N193 | 6335680 | 6342130 | 6337060 | 6338250 | 6343537 | 6335500 |
| N194 | 6335650 | 6337560 | 6335600 | 6335725 | 6342110 | 6335160 |
| N195 | 6338160 | 6335651 | 6337508 | 6338161 | 6340366 | 6337510 |
| N196 | 6337518 | 6335076 | 6340218 | 6338150 | 6335650 | 6335697 |
| N197 | 6342060 | 6337507 | 6335681 | 6337600 | 6335165 | 6335730 |
| N198 | 6335650 | 6335160 | 6340440 | 6357020 | 6335167 | 6340200 |
| N199 | 6343537 | 6342100 | 6338250 | 6342640 | 6337510 | 6338260 |
| N200 | 6335303 | 6338120 | 6337250 | 6338260 | 6337610 | 6335485 |

---

## Author Comment (AC2)

*Original reviewer comments are in italics,* **authors' response is in bold**.

**Anonymous Referee #2:**
*Review of "High-resolution drought simulations and comparison to soil moisture observations in Germany" This manuscripts analyses the relationship between soil moisture observations and estimations by models in Germany with focus on drought monitoring. The manuscript is well written and organised. Nevertheless, I would like to include some caveats related to the limitations of the validation approach and the usefulness of the new high spatial resolution data base in order to assess drought severity. I include specific details related to these issues (and others) below (numbers refer to the specific lines of the manuscript):*

**Authors' response #1: We thank the Reviewer for the assessments of our work. We paid detailed attention to all comments and we have addressed all of them below accordingly.**

*11- What is "vegetation period"? Is maybe "vegetative active period"?*

**Authors' response #2: We agree to the suggestion and will change terms in the manuscript.**

*Table 1- I would like to ask for a technical question. Do you think if the quality of the globcover map is sufficient for the modelling. How is considered the uncertainty of land cover information in the model? I find very high detail of information related to the improvement of the soil maps, map I have the impression that the land cover data is not considered so carefully and it can be strongly relevant to model soil moisture given different water consumption by ecosystem types (even at the scale of species), the role of root structure, root depth, etc.*

**Authors' response #3: The hydrological simulations of German drought monitor operate at the nation-wide scale with large-scale available information. One of the research questions was to evaluate whether it possible to provide higher resolved information at a satisfying quality.**
**The increase of model resolution in the second version of the drought monitor was motivated both by the release of a new German-wide soil map [1] and increased user need to higher resolution simulations as extensively described in the main manuscript. This resulted in $\approx 1.2 \times 1.2\,\mathrm{km}^2$ model resolution in the GDM-v2-2021 setup as a compromise between scientific/model perspective (limited by data availability and process representation) and stakeholder/user perspective (see also conclusion lines 438-441). Changes in landuse data (also the change of geology data and projection to WGS-84) in the new drought monitor version (GDM-v2-2021) on the other hand were driven by current efforts increasing the applicability and comparability of mHM to regions other than Germany and outside Europe. Change in these landuse datasets have minor implications compared to the change in the soil dataset. Currently, mHM takes relatively raw landuse classes. Species specific landcover is currently not accounted for. The difference in the resolution of GLOBCOVER and**

CORINE landuse dataset are in sub grid scale that influences the sub-grid variability (GLOBCOVER resolution: 300 meters, CORINE < 100 m). Differences between the land cover datasets reduce if the land cover data is aggregated to the spatial resolution of the model. For example, at the spatial resolution of 1.2km, over 85 % of the grid cells both datasets agree on the dominant landcover. This shows that differences stem from differences at high spatial resolution and do not have a large impact on the simulation.

We will include these aspects in the main manuscript to point out the limitations of the study. We propose to add the following sentence in the main manuscript in line 147: "The changes in landuse and geology dataset can influence the simulations, yet play a minor role for the soil moisture simulations compared to the change in the soil dataset because changes of landuse data are in subgrid scale (resolution GLOBCOVER 300m, CORINE <100m) and no direct feedback of from saturated "groundwater" storage to soil moisture storage is implemented in mHM."

*150- I find very few information related to the meteorological data. There is not information on the number of stations used for each variable, the quality of the data, quality control processes, data gap filling, temporal homogeneity, etc., but also information related to the quality of resulting gridded data (e.g., cross-validation statistics would be useful). Meteorological data can be also an important source of uncertainty in the model outputs...*

Authors' response #4: The meteorological input station data that is used for interpolation is provided by the German Weather Service (DWD) through the Climate Data Center (`ftp://opendata.dwd.de/climate_environment/CDC/`). It is subject to extensive quality controls [2]. Additionally, quality controls are implemented in the preprocessing steps of the interpolation routine e.g. checking plausible variable range. In [5] describing the mHM simulations underlying the GDM version 1, the interpolation method for interpolating the meteorological data is described and validated in detail. Different approaches to calculate theoretical semi-variograms were tested and evaluated. A cross-validation (Jackknife method) was performed to test the ability of the External Drift Kriging (EDK) to estimate meteorological variables at the measurement locations. According to [5] the average and the standard deviation for the different errors assessments over all stations were 0.01 and 0.15 mm d−1 for the bias, 0.64 and 5.60% for the relative bias, 0.93 and 0.03 for the Pearson correlation coefficient, and 1.75 and 0.48mm d−1 for the root mean square error. Additionally, a comparison of the EDK interpolations conducted by [5] to the REGNIE gridded precipitation data [4] provided by DWD showed satisfactory results with spatially averaged bias of the daily fields of 0 with a standard deviation of 0.11 mmd−1 within the period 1951–2010.

*151-154- What about uncertainty of the Hargreaves-Samani equation to estimate Potential Evapotranspiration? It is widely known that temperature based methods show uncertainties related to physically based models like the Penman-Monteith equation. For example, wind speed and relative humidity may have large importance on PET, even more in non-stationary scenarios charecterised by decreased relative humidity over land and wind speed reduction.*

Authors' response #5: The actual ET is the important water balance component being the reduction term of the potential Evaporation. Comparisons of actual ET estimated with mHM were conducted in [5] comparing to remote sensing data (MODIS) and FLUXNET towers and by [3] over Europe using in situ observations and a gridded product from FLUXNET showing a good overall fit.

We certainly agree on the superiority of Penman-Monteith methods to estimate PET at the field scale if high quality field-scale data is available. The conclusion in Line 436 refers to this comment stating that "we may achieve a more precise estimation of potential evapotranspiration through implementing the Penman-Monteith methods.". Regionalized estimates of physical based PET are still however largely limited by spatial data availability in terms of number of measurement stations and temporal data availability in terms of record lengths. No reliable high quality daily gridded estimates for both wind and global radiation are currently available for full time period (1950-2020) to allow Penman-Monteith ET estimation at the spatial modelling scales used in the study. A longer simulation time period is prioritized for the German Drought Monitor to obtain a long statistical database for the SMI estimation instead of cutting the simulation period.

*172- Figure 1 > Figure 2. 231-235- The validation procedure is exclusively based on correlations. Nevertheless, if the main purpose of the manuscript is related to drought monitoring, I think more relevant to assess model outputs during periods of water deficits. For example, it would be useful to check the capability of models to identify duration and magnitude of the dry periods. High correlation could mask a poor goodness between observations and models during dry periods. I would suggest to include statistics focusing on the drought periods in addition to the non-parametric correlations.*

Authors' response #6: The soil moisture index (SMI) is estimated for every grid cell and every day of the year. Hence, the number of data points to estimate the histogram and percentiles to classify drought is equal to the number of years with observational data. Due to the limited observational data time series lengths, it is not possible to estimate drought characteristics as intensities and duration. Therefore, we decided to use the time span 1951-2015 initially in setting up the first version of the drought monitor to ensure statistical stability of the system. Due to these limitations, we came up with our study design: comparison of observed and simulated soil moisture in a first step and comparison of simulated drought intensities in the following step between the two model setups. The correlation statistics are calculated on deseasonalized anomalies that removes the seasonal mean cycle. From a mathematical point of view, having a constant bias between the observations and the simulated soil moisture would have no effect on the drought classification. The percentile based approach of the SMI would remove the bias.

*170-210- The length of the observation series is not indicated in this section. This information is relevant to assess robustness of the relationship between observations and models. Have the series the same length? How is this considered in the assessment of the signification of the relationships? I think this issue is affecting the*

*validation of the results over the entire section 3.1 since the length of the series affect the degrees of freedom of the correlation analysis. I see in table 3 that the length of the series is between 2 and 5 years, which is too low to provide a robust validation of the model outputs.*

**Authors' response #7: We generally agree that longer observational time series would support a more robust validation. Nevertheless, in our study we compiled the best possible observational soil moisture data base on the national scale for Germany. We suggest to describe the time series lengths more clearly in the manuscript by adding following sentence "Time series lengths of the observations are between 2.8 and 17.8 years with a median (mean) of 6.5 (6.7) years."**
**In order to investigate the consequences of different time series lengths, Figure R1 shows correlations against the length of time series. No systematic relation between correlations and the time series length can be detected.**

[Figure]

Figure R1: Correlations for the soil moisture observations against simulations (GDM-v2-2021 setup, 0-25cm depth) dependent of number of years of observations.

*Figures 6 and 7. Under my opinion, I do not think that this information is providing an useful output to determine the goodness of providing additional spatial resolution to assess drought severity. Large scale statistics are aggregating the information, being normal that both databases at 4km and 1 km of spatial resolution provide similar results. I think the relevant information of the 1 km modelling approach is not the general large spatial pattern but the local differences that could emerge given higher spatial resolution. This is something interesting to be analysed (e.g. using spatial statistics: the variance between grid cells, the differences between areas characterised by diversity of land cover/soil characteristics) to determine if higher spatial resolution is providing relevant information for drought monitoring and management. Observing Figures 6 and 7 I would say that the higher spatial resolution is really not needed as it basically identifies the same patterns that 4 km grids.*

**Authors' response #8: Thank you very much for the suggestions to include spatial statistics to show the regional differences between the model setups.**

**Drought intensities that are shown in Fig. 6 and 7. can reach a maximum value of 0.2. Fig 7 shows that in the absolute differences up to 0.1 occur between both drought monitor versions on the grid scale. This clearly shows a large impact of the new study setup on simulated drought characteristics.**

**Following the reviewers suggestions, we conducted an additional analysis that complement the analysis of the drought clusters. In Figure R2 the variance between grid cells for drought intensities during vegetation active period are shown as semi-variograms. In general, the spatial variance is larger in the total soil than top soil. The GDM-v2-2021 setup shows a general larger spatial variance between grid cells in the top soil and larger increase with distance (see Figure R2 a)). The spatial variance in the total soil is lower at smaller distances in the GDM-v1-2016 setup, but slightly higher at larger distances. Figure R2 b) showing semi-variance normalized by distance (and log scaled x-axis to to improve visibility of smaller distances) demonstrates that in the GDM-v2-2021 setup the distance-normalized variance of drought intensities is increased especially at small spatial scale in both the top and total soil, indicating larger local differences in response to drought intensities. We suggest to include Figure R2 in the main manuscript in section 3.2 and add a paragraph based on the findings above..**

[Figure]

Figure R2: Empirical semi-variograms for drought intensities during vegetation active period in upper Soil for GDM-v1-2016 and GDM-v2-2021 setup. The bin size was set to 5 km (nearest larger even km bin size relative to the GDM-v2-2016 modelling resolution). The len scale and nugget of the fitted exponential theoretical semi-variograms are noted in the legend. In Subplot b) the y and x axis are log scaled.

**References**

[1] BGR. Digital soil map of Germany 1 : 200,000 (BUEK 200) v0.5, 2020.

[2] F. Kaspar, G. Müller-Westermeier, E. Penda, H. Mächel, K. Zimmermann, A. Kaiser-Weiss, and T. Deutschländer. Monitoring of climate change in Germany – data, products and services of Germany's National Climate Data Centre. *Advances in Science and Research*, 10(1):99–106, Aug. 2013.

[3] O. Rakovec, R. Kumar, J. Mai, M. Cuntz, S. Thober, M. Zink, S. Attinger, D. Schäfer, M. Schrön, and L. Samaniego. Multiscale and Multivariate Evaluation of Water Fluxes and States over European River Basins. *Journal of Hydrometeorology*, 2016.

[4] M. Rauthe, H. Steiner, U. Riediger, A. Mazurkiewicz, and A. Gratzki. A Central European precipitation climatology – Part I: Generation and validation of a high-resolution gridded daily data set (HYRAS). *Meteorologische Zeitschrift*, pages 235–256, July 2013.

[5] M. Zink, R. Kumar, M. Cuntz, and L. Samaniego. A high-resolution dataset of water fluxes and states for Germany accounting for parametric uncertainty. *Hydrology and Earth System Sciences*, 21(3):1769–1790, 2017. ISBN: 1607-7938.

---

## Author Response (AR1)

Helmholtz Centre for Environmental Research – UFZ Permoserstr.15 • 04318 Leipzig • Germany

> Contact person: Friedrich Boeing Department Computational Hydrosystems friedrich.boeing@ufz.de

Leipzig, April 2022

**Re-submission HESS-2021-402**

Dear Dr. Teuling,

we would like to thank you and the reviewers again for the comments which helped to increase the overall quality of the manuscript.

Please find the newly uploaded revised manuscript, incorporating the proposed changes and additions (including the proposed changes and additions to the figures, the discussions and language edits), as well as a marked-up manuscript version (latexdiff file) showing the differences between the originally submitted and the revised manuscript below, as well as the detailed responses to the reviewers' comments. Additionally to the reviewers requested changes, we updated and extended the sections "Code and data availability" (including DOI links) and "Acknowledgments".

Kind regards,

**Friedrich Boeing**

**Helmholtz Centre for Environmental Research – UFZ**

Company domicile: Leipzig

Permoserstr. 15, 04318 Leipzig, Germany or PF 500136, 04301 Leipzig, Germany phone +49 341 235-0

info@ufz.de www.ufz.de

Registration court: Leipzig district court Commercial register No. B 4703

Chairman of the Supervisory Board: MinDirig'in Oda Keppler

Scientific Director: Prof. Dr. Georg Teutsch

Administrative Director: Dr. Sabine König

Bank details: HypoVereinsbank Leipzig Sort code 860 200 86 Account No. 5080 186 136 Swift (BIC) code HYVEDEMM495 IBAN No. DE12860200865080186136 VAT No. DE 141 507 065 Tax No. 232/124/00416

Original reviewer comments are in italics, authors' response is in bold.

**Referee #1:**

**SUMMARY:**

**A recommendation: moderate revision**

Using observed SM data collected from 40 locations and in 4 different measurement methods in Germany, the authors evaluate the performance of second generation operational German Drought Monitor in simulating soil moisture (SM). Two major research questions within this paper have been adequately addressed and can be summarized as follows: 1. how well the GDM capture the SM dynamics; 2. will GDM with higher spatial resolution produce SM estimates with higher quality compared with the GDM of former edition. Through the research, it was found that 1. SM dynamics simulations could be moderately improved; 2. higher resolution drought information at the one-kilometer scale can be met.

This research is a report of the improvement in the model performance which is evaluated in the perspective of comparisons between the model simulations and the observations. The article conforms to the journal-specific instructions and is relevant to HESS. The work is appropriate to be published in this journal after some revisions. In the following part, I will state the major arguments in detail.

**Authors' response #1: We thank the Reviewer for an overall positive assessments of our work. We paid detailed attention to all comments and have addressed all of them below accordingly.**

The comparison between the observations and the model simulations are mainly indicated in the form of Spearman rank correlation coefficient. Nevertheless, the results of the comparison are in fact not so ideal in the perspective of solely the Spearman correlation coefficient, not to mention that significance values have been neglected when the observations and model results are compared. For example, when the observations from all the sites are included, the coefficients are normally lower. In addition, p-value has only been mentioned when both versions of the mHM and observations are compared (table 2). No mentioning of significance values also makes the comparison results not so validated. The significance value should be mentioned in the research to make results more reliable

Authors' response #2: Thank you for this comment. We agree with the reviewer that the significance value is the essential information. Therefore we added the significance values into Figure 4 (see exemplary Figure R1 below) and Figure 5 by indicating locations with p-values

---

## Author Response (AR2)

**Manuscript ID: hess-2021-402 Response to the Reviewers**

**Referee #3:**

Review of Boeing et al., HESS-2021-402 "High-resolution drought simulations and comparison to soil moisture observations in Germany"

This study investigates the performance of the land surface modelling system underlying the German Drought monitor. The authors compare two model versions with different spatial resolution against a comprehensive set of soil moisture measurements obtained through different methods and from different regions across Germany. They find overall similar performance with slight improvements in the cold season, while additionally regional changes in the results can be observed, probably following changes in the underlying soil type quantification.

Recommendation: I think the paper requires major revisions.

The topic of the study is interesting and timely. Drought monitoring in Germany is of increasing relevance given the recent droughts, and their impacts on agriculture, forestry and water resources. Accurate drought monitoring, as well as respective analyses and prediction, requires accurate land surface models which can represent relevant processes. In this context, the current study is a relevant contribution as it compares the modelling system of the German drought monitor with a comprehensive set of soil moisture measurements obtained through different techniques, and from different regions throughout the country. Further, the study illustrates the advantages of performing simulations at higher spatial resolution which is increasingly relevant for regional- scale drought assessments. I think that the manuscript is a good match match for the readership of HESS, and appreciate the efforts made by the authors to improve the manuscript in response to the previous comments of both reviewers.

However, before it is ready for publication some concerns particularly related to the understanding of differences between modelled and observed soil moisture dynamics should be resolved, as detailed below.

Authors' response #1: We really appreciate your time and efforts in providing detailed and relevant comments to improve the quality of our work. During the revision, we paid detailed attention to all critical comments and we have addressed them with our best efforts. In the following you find all corresponding answers.

General comments:

(1) I agree with both reviewers that additional explanations/analyses are necessary for understanding the differences between the modelled and observed soil moisture dynamics, and feel that the author's responses in this context fall short in several aspects.

(i) Why does the updated model version compare better with observations in fall and winter, but not so much during the growing season?

Authors' response #2: Thank you for this comment. We recognise that several of the changes in the model setup may provide explanations for the improved model performance in the fall and winter. The higher modelling resolution of the 1 km runs may better resolve the sub-grid variability of cold season related processes such as snow accumulation that improves

the simulated SM dynamics. Additionally, the finer spatial soil texture representation possibly contribute to an improved model representation of soil wetting/drying e.g. especially during saturated conditions in the cold season. Nevertheless, we would like to stress that the changes in these seasons are significant in statistical terms, but the differences in absolute terms are still rather small ( $\Delta + 0.07$  for fall and  $\Delta + 0.12$  for winter over all locations). We added these clarifications to the revised manuscript.

(ii) Following the main purpose of the German drought monitor, the validation of the modelled soil moisture against observations needs to be performed also with an exclusive focus on drought periods in addition to the overall agreement already assessed in the paper. Even though there is less data available during drought periods, I feel it should still be sufficient for a meaningful analysis, particularly given the relatively low threshold used in the definition of droughts, and the fact that the time series length does not systematically affect the determined agreement between modelled and observed time series (Figure R1).

Authors' response #3: We thank the Reviewer for this suggestion. To address this comment, Figure R1 shows the dry anomaly spectrum based on the Spearman rank correlations between simulated and observed deseasonalized SM anomalies that fall below the 20th percentile in the observed SM time series. It is important to emphasize that we do not aim to estimate drought periods here, as its solid calculation requires much longer time series. The estimation of drought is performed using histograms for every grid cell and day of the year (see method section 2.4.1 in the manuscript). Consequently, estimating robust percentiles requires time series lengths of minimum 30 years – this means that the time series length of the observational data is considered insufficient. Figure R1 a) shows a median correlation of 0.61 over all observations in the GDM-v2-2021 (1-km setup). The performance in the two model setups remains similar. However, the comparison separated between the measurements with larger spatial footprint (SDM, CRNS) and point scale measurements (SPM, LYSI) shows that the agreement between simulations and the larger footprint observations increased towards the high resolution setup, but the median agreement to the point scale SM measurement decreased. In general, the measurements with larger spatial footprint display higher agreement to the simulations. Due to the varying day-to-day variability of SM between the SM observation types and the simulations, in Figure R1 b) additionally a statistical smoothing was applied by calculating a running 30 day mean on the daily SM time series before removing the seasonal cycle. This approach is similar to the SM pre-processing for the SMI as proposed in Zink et al. [2016]. Figure R1 b) shows when smoothing is applied, the agreement between observations and simulations during dry periods can be substantially improved to a median correlation of 0.7 over all observations in the GDM-v2-2021 setup ( $\approx$  1-km resolution). Especially the agreement between the point scale measurements and simulations is increased to a median correlation of 0.63 in both model setups. We added this paragraph and Figure R1 to the section 3.2 of the revised manuscript.

Figure R1: Correlations of deseasonalized soil moisture below 20th percentile (based on the observed SM timeseries) between simulations and observations. In b) additionally a statistical smoothing was applied by calculating a running 30 day mean on the daily SM timeseries before subtraction of the seasonal cycle. The correlations are shown for all observations (n=46) and separated between observations with larger spatial footprint (n=20) including Cosmic Ray Neutron Sensing (CRNS) and spatially distributed measurements (SDM) as well as point measurements (n=26) including single profile measurements (SPM) and lysimeters (LYSI).

(iii) To better understand the small-scale differences between both model versions it would be helpful to relate the obtained differences (e.g. in Figure 8) to the underlying differences between the soil type maps and land use classifications to establish some cause-effect relations.

(iv) The role of updates in the land use vs. soil datasets, as raised by both reviewers, is not really clear even in the updated version of the manuscript. While the authors state that changes in the land use dataset do not affect the results, it remains unclear how they arrived to this conclusion.

Authors' response #4: We would like to thank the Reviewer for his suggestion. We combined question (iii) and (iv) due to their similarity. The study design was conceptualized to present a comparison of the high resolution 1 km setup to soil moisture observations and secondly a comparison to the previous coarser resolution operational modelling setup. The major motivation to test a newer version of the hydrological modelling system was the availability of the BUEK200 soil dataset, resulting in a 25-fold spatial resolution increase. Also other changes were implemented (e.g. increased grid cell resolution of the hydrological simulations, change of landuse and geology dataset). Additionally, the calibration parameters of both setups were estimated independently. We would like to clarify that our study does not aim to explain methodological differences between model setups (e.g. the effect of solely changing the soil maps on soil moisture), but to provide a comparison of two operational modelling setups used for drought monitoring. To address these important and interesting questions, a different study setup would be required.

However, to eliminate any reader's confusion, we have added these limitations more clearly to the revised manuscript: Figure R2 demonstrates the different roles of the change in SM dynamics between the model setups related to the specific soil and land use datasets by showing temporal correlations between SM from separated model runs fixing all model settings (L1  $\approx 1.2 \times 1.2 \text{ km}^2$  resolution, default mHM parameters) only changing the soil dataset (BUEK200 - BUEK1000) and in a separate step only changing the land use dataset (CORINE – GLOBCOVER). Figure R2 suggests that the change of the soil dataset has a much larger impact on the SM simulations compared to the change of the land use dataset in these specific model setups. The CORINE and GLOBCOVER land use datasets both have already high horizontal resolutions ( $\approx 100 \,\mathrm{m}$ and 300 m, respectively). The differences between the land use datasets mostly lie in the sub-grid scale of the mHM hydrological modelling resolution and have a minor effect on the upscaled hydrological response at the L1 level (here  $\approx 1.2 \times 1.2 \,\mathrm{km^2}$ ). This paragraph and Figure R2 has been added to section 3.2 and the appendix in the revised manuscript, respectively.

---

## Author Response (AR3)

Contact person:
Friedrich Boeing
Department Computational
Hydrosystems
friedrich.boeing@ufz.de

**Helmholtz Centre for Environmental Research – UFZ**
Permoserstr.15 ⬚ 04318 Leipzig ⬚ Germany

Leipzig, August 2022

**Minor revision HESS-2021-402**

Dear Dr Teuling,

we would like to thank you and the reviewers again for the comments, which helped to increase the overall quality of the manuscript.

Please find the newly uploaded revised manuscript, incorporating the last referees' suggestion to include the main findings of Figure 7 and A4 into the abstract and conclusions.

Also, we updated Figure A4 to a colorblindless friendly version.

Kind regards,

Friedrich Boeing

**Helmholtz Centre for Environmental Research – UFZ**

Company domicile: Leipzig

Permoserstr. 15, 04318 Leipzig, Germany
or
PF 500136, 04301 Leipzig, Germany
phone +49 341 235-0

info@ufz.de
www.ufz.de

Registration court: Leipzig district court
Commercial register No. B 4703

Chairman of the Supervisory Board:
MinDirig'in Oda Keppler

Scientific Director:
Prof. Dr. Georg Teutsch

Administrative Director:
Dr. Sabine König

Bank details:
HypoVereinsbank Leipzig
Sort code 860 200 86
Account No. 5080 186 136
Swift (BIC) code HYVEDEMM495
IBAN No. DE12860200865080186136
VAT No. DE 141 507 065
Tax No. 232/124/00416